 

## Registered report

cognition/psychology/developmental biology

word learning, cognitive development, infant cognition, language acquisition, word comprehension

**Author for correspondence:**
Natalia Kartushina
e-mail: natalia.kartushina@psykologi.uio.no

# Word knowledge in six- to nine-month-old Norwegian infants? Not without additional frequency cues

## Natalia Kartushina and Julien Mayor

Department of Psychology, Faculty of Social Sciences, University of Oslo, Forskningsveien 3A, 0373 Oslo, Norway

NK, 0000-0003-4650-5832

The past 5 years have witnessed claims that infants as young as six months of age understand the meaning of several words. To reach this conclusion, researchers presented infants with pairs of pictures from distinct semantic domains and observed longer looks at an object upon hearing its name as compared with the name of the other object. However, these gaze patterns might indicate infants' sensibility to the word frequency and/or its contextual relatedness to the object regardless of a firm semantic understanding of this word. The current study attempted, first, to replicate, in Norwegian language, the results of recent studies showing that six- to nine-month-old English-learning infants understand the meaning of many common words. Second, it assessed the robustness of a 'comprehension' interpretation by dissociating semantic knowledge from confounded extra-linguistic cues via the manipulation of the contingency between words and objects. Our planned analyses revealed that Norwegian six- to nine-month-old infants did not understand the meaning of the words used in the study. Our exploratory analyses showed evidence of word comprehension at eight to nine months of age—rather than from six to seven months of age for English-learning infants—suggesting that there are cross-linguistic differences in the onset of word comprehension. In addition, our study revealed that eight- to nine-month-old infants cannot rely exclusively on single extra-linguistic cues to disambiguate between two items, thus suggesting the existence of early word-object mappings. However, these mappings are weak, as infants need additional cues (such as an imbalance in frequency of word use) to reveal word recognition. Our results suggest that the very onset of word comprehension is not based on the infants' knowledge of words *per se*. Rather, infants use a converging set of cues to identify referents, among which

frequency is a robust (pre-semantic) cue that infants exploit to guide object disambiguation and, in turn, learn new words.

# 1. Introduction

## 1.1. Emergence of word comprehension in infants

Recent research on infants' early word comprehension has considerably changed our view of language development. While for decades, researchers believed that infants wait until they are approximately eight months of age to understand their first words, recent studies suggest that children know the meaning of several common words as early as from six months of age [1–5].

The first signs of word recognition can already be seen as early as at 4.5 months of age. In particular, it was found that 4.5-month-olds recognize their own names among foils having similar prosodic patterns, suggesting that at this age infants are able to represent and recognize sound patterns frequently used in their environment [6]. Later, at the age of six months, infants show comprehension of words referring to salient social figures in their lives, i.e. 'mummy' and 'daddy', as indicated by greater proportion of looks to the video of their mother when compared with the video of their father while hearing the label 'mummy', and vice versa [7]. Therefore, by the age of six months, infants are considered to have established one-to-one associations between proper names and the corresponding parent. Moreover, from the same age, infants start showing comprehension for some concrete objects, for example, body parts 'hand' and 'feet' [5] or food-related items [2], and, some months later (at 9–10 months), their comprehension extends to more abstract words (e.g. all-gone and hi [3]), as revealed by longer looks to the target picture upon hearing its label when compared with the distractor [3,4].

However, recent research suggests that word comprehension in 6–12-month-old infants is fragile. For instance, Bergelson & Swingley [2] (hereafter BS12) have shown that when presented with two pictures of objects sampled from two different semantic categories (for example, banana and hair), six- to nine-month-old infants looked longer at the picture of a banana when they heard the word 'banana' when compared with when they heard the word 'hair' [2]. Yet, other studies have shown that when picture-pairs refer to words from the same semantic category, e.g. a book and a ball (toys) or a dog and a cat (animals), or when a label is semantically related to the target, infants failed to reveal significant increase in looking times at the target compared with the distractor at 6 [1] and 12 [8] months of age. Taken together, these results may suggest that early word categories are semantically underspecified [1,8]: 6–12-month-old infants rely on semantic cues to disambiguate between two objects; however, if the objects belong to the same semantic category (e.g. a cat and a dog both represent animals), infants fail to disambiguate between them, even though they discriminate both objects visually.

Early word categories get progressively semantically refined over the second year of life. While 12-month-old infants look at the target (e.g. cookie) to the same degree whether hearing a matching label (e.g. cookie) or hearing a semantically related label of an object not presented on the picture (e.g. banana), 18-month-olds show stronger preference for the target when they hear the matching label over the related one [8]. These data suggest that infants' word-referent mappings are semantically refined by the age of 18 months, although limited to typical exemplars of the semantical categories until the age of 24 months [9]. Yet another study [10] has shown that perceptual similarity between two objects suffice to destabilize word comprehension of semantically related items at this age. Arias-Trejo & Plunkett [10] have shown that 18–24-month-old infants fail to disambiguate between two perceptually similar objects if they belong to the same semantic category (e.g. apple and orange), as compared with objects belonging to different semantic categories (e.g. apple and ball). In sum, although early word categories are progressively refined over the second year of life, they are not yet fully differentiated even in 24-month-old infants: infants may use different cues (e.g. semantic and perceptual) to disambiguate between items; however, if multiple cues (i.e. semantic and perceptual) are congruent with both items, infants fail to disambiguate them.

To sum up, recent studies suggest that infants' early word categories are semantically coarse: young infants rely on available linguistic (semantic category) and perceptual (e.g. shape) cues to disambiguate between two objects, rather than on a firm semantic mapping between a word and a referent. Although word categories gain in semantic specificity with age, they are not fully specified even in 24-month-old infants, who fail to disambiguate between two objects when the available cues are congruent with both of them [10]. In the current study, we tested whether six- to nine-month-old infants might also use other,

extra-linguistic, cues to disambiguate between items: namely, the context in which words are typically uttered, and the frequency of word use—two of the strongest factors helping infant learn words as suggested by recent large-scale corpus analyses that we briefly review next.

## 1.2. Two alternative accounts for infants' behaviour in word recognition studies

A recent large-scale corpus analysis of data from more than 38 000 children across 10 languages revealed that a noun's concreteness and frequency are the first two strongest factors in predicting the emergence of its comprehension [11]. Empirical data support this result: most experimental studies showing word comprehension in young infants used concrete and frequent words [1,2,4]. Moreover, a recent study has shown a relationship between infants' (concrete) noun comprehension in the laboratory and the quantity and, in particular, the quality of use of the (concrete) nouns at home [1]. Six-month-old infants who heard concrete nouns more frequently in co-occurrence with the referred objects tended to show better word recognition in the laboratory, suggesting that frequent word-object co-occurrence facilitates word learning. Therefore, since word frequency and word-object co-occurrence are related to word comprehension in young infants, an imbalance between words' frequencies in a pair may be exploited by infants to disambiguate between objects, such that a frequently heard label will trigger more looks to a frequently seen object, whereas an infrequently heard label will trigger more looks to an infrequently seen object, despite a lack of a firm semantic label-object mapping.

Interestingly, the analysis of an ultra-dense corpus collected from one child demonstrated that factors related to contexts in which a word is used (linguistic, temporal and spatial) are even stronger predictors of the age of word learning than frequency of use [12]. In particular, the study has shown that words that the child hears consistently and repeatedly in a given time, space and linguistic context (referred to as highly temporally, spatially and linguistically distinctive words) are produced earlier than less distinctive words. For instance, a word 'spoon' is a highly contextually distinct word [13], as it is used consistently in one spatial location, i.e. the kitchen, and within a homogeneous linguistic set of words, for example, when eating (a yogurt, soup, etc.). Hence, the word 'spoon' is predicted to be acquired (here, produced) earlier than, say, the word 'hand', which is heard in less spatially and linguistically distinct contexts, for example, when eating, washing, playing, dressing [13]. Indeed, the first production of the word 'spoon' appears at 16.4 months, whereas 'hand' is first produced at 18.1 months [14]. In the light of these findings, it is important to ensure that studies addressing word comprehension in young infants match within-pair items for frequency, as well as for their temporal, spatial and linguistic context, as such cues might bias infants' looking behaviour.

Given the results of these corpus-based studies, we argue that there might be two alternative explanations for the findings of word comprehension previously reported in six- to nine-month-old infants. First, it is possible that young infants rely on frequency to disambiguate between objects. For instance, in the BS12 study [2], whereby six- to seven-month-olds displayed comprehension of common words for body parts and food items, six out of the eight item pairs were unbalanced in terms of the frequency of occurrence in child-directed speech [15]. For example, in the pairs 'hair–banana' and 'hand–yogurt' the former words (in each pair) were considerably (four to six times) more frequent (3699 and 3899) than the latter words (907 and 630) (extracted from all corpora in English from https://github.com/mikabr/aoa-prediction/, see [11]).[1] Since infants are sensitive to frequency of words and objects [1], it is possible that they use it to disambiguate between objects: when hearing a high-frequency label 'hand', infants might look at a high-frequency encountered object, 'hand', but also to other high-frequency encountered objects, as, for example, 'hair' and 'milk'. However, when hearing a low-frequency label, for example, 'yogurt', infants might look at objects that they encountered less frequently in their environment, such as 'yogurt', but also 'ear'. In fact, a detailed re-analysis of word pairs used in the BS12 data revealed a strong correlation between their effect-sizes and the frequency mismatch between the two words in a pair (Pearson $r = 0.71$, $p = 0.048$, $n = 8$), suggesting that frequency differences might be an important additional cue that infants use to disambiguate between items.

Second, longer looks to a given item might not reflect comprehension of the referent word *per se*, but its recognition in relation to the context in which it is used [12]. For example, the word 'spoon' is consistently used in the same spatial, linguistic and temporal context, i.e. kitchen and food at the

---

[1]Although, frequencies retrieved from infant-directed speech corpuses do not reflect the exact frequency with which each individual infant encounters specific words, they are considered reliable indicators of their relative frequency of occurrence. In the present study, we compute a frequency ratio for each word pair. See Methods for details.

mealtime, in a combination with other contextually related words (e.g. yogurt and cup). Hence, in the laboratory, if a child hears 'spoon', she might think of a general kitchen–food–meal-type context and then look at a picture of a spoon; analogously, she could look at a picture of a spoon after hearing words used in the same context, for example, 'yogurt' or 'cup'. Note that, however, although 'spoon' is contextually related to the word 'yogurt', these two words belong to different semantic categories, i.e. utensils and food items, respectively. Previous research [1] has already provided evidence that infants' early word comprehension of semantically related items is fragile: six-month-old infants fail to disambiguate between items when the semantic cues are congruent with both items (e.g. a book and a ball, both are toys), suggesting that they rely on semantic cues to disambiguate between items. Is it possible that six- to nine-month-old infants use other available cues to disambiguate between items, in particular, cues that have been shown to help infants learn words; i.e. the context (spatial, linguistic or temporal) of word use and its frequency? In the current study, we tested these two alternative explanations for the finding that infants know the meaning of multiple words, as reported in BS12: we examined whether six- to nine-month-old infants can rely on non-semantic, extra-linguistic cues, i.e. frequency and context of use, to disambiguate between objects.

## 1.3. The current study

The aims of the current study were twofold. First, we attempted to replicate, in a new language, Norwegian, the results of the BS12 study [2], in which six- to nine-month-old English infants were shown to disambiguate between items sampled from distinct semantic categories (and contexts). Due to some methodological differences between our experiment and the BS12 study (e.g. the present study investigated word comprehension in Norwegian infants, see Methods for further details), the current study should be considered as a conceptual replication of Bergelson and Swingley's study, rather than a direct replication. A comparison of the number of words understood by Norwegian- and American-English-speaking infants, as reported by their parents in the Communicative Development Inventories (CDIs; retrieved from wordbank.stanford.edu, see [13,16,17]), suggests that language acquisition proceeds at a similar pace across both languages: at the age of eight months, the median receptive vocabulary size is nine words for Norwegian and eight words for English; at the age of nine months, it is 19 for English and 22 for Norwegian. Also, a cross-language comparison of the size of the receptive vocabulary across ages and different grammatical categories reveals similar developmental trajectories in English- and Norwegian-speaking infants (retrieved from [18], http://mikabr.github.io/vocab-comp/). Thus, we expected that Norwegian six- to nine-month-old infants, tested in the current study, would have shown similar behaviour, compared to their English-speaking peers, tested in the BS12 study. The second aim was to assess the interpretation that infants understand the meaning of the words by examining two alternative explanations: namely, that infants may differentiate items capitalizing on differences in (i) their frequencies of occurrence or (ii) in their contexts of use.

To examine word comprehension in six- to nine-month-old infants, we adopted an infant preferential looking (IPL) paradigm, as in the BS12 study. In each trial, the infant saw on a screen two pictures of objects sampled from different semantic categories, and contexts, and heard the following sentence: 'Look at the ⟨target⟩!' (similar to [2]). For instance, the pair 'cookie–belly' (i.e. kitchen-related versus bathroom-related contexts) was presented once with the carrier sentence: 'Look at the cookie!' and another time with: 'Look at the belly!'. These matching trials acted as a replication of the BS12 study.

To evaluate two alternative interpretations, we created two additional types of trials: frequency- and context-related trials. In these two types of trials, we used the same pictures as the ones used in the matching trials. Yet, in the context-related trials, matching labels were substituted by contextually related words, while in the frequency-related trials, they were substituted by frequency-related words. In the context-related trials, the items were selected in such a way that infants would not be able to use frequency to disambiguate between the objects, whereas in the frequency-related trials, infants would not be able to use context to disambiguate between objects. Crucially, in both frequency- and context-related trials, infants would not be able to rely on semantic cues to disambiguate between objects, since the matching labels and their related words belonged to different semantic categories (e.g. 'spoon' and 'cookie' belong to 'utensils' and 'food' categories, yet both are contextually related to the spatial location 'kitchen'; similarly, 'belly' and 'bathtub' belong to different categories, but are encountered in the spatial location of bathroom during bath/cleaning routines). This specificity was one of the key manipulations in the current study, since previous research on word comprehension and semantic relatedness in young infants [1,2,8] did not aim at disentangling semantic from contextual overlap between the objects in a pair.

We expected that, on matching trials, infants would look longer at the target picture than at the distractor when hearing the target's label, similarly to the BS12 study [2]. However, in order to conclude that this looking behaviour reflects infants' true semantic knowledge and not their reliance on extra-linguistic cues when disambiguating between two objects, infants should fail to show preference for an object when its corresponding label is substituted by a semantically unrelated word that is either similar in frequency or sampled from the same context. If, on the other hand, we observed that infants looked to the same degree at the target picture whether hearing the name of a contextually and/or frequency-related object, as compared with the matching word (the label), then we would conclude that infants rely on extra-linguistic cues (contextual and/or frequency) to disambiguate between two objects despite a lack of word knowledge at a semantic level. A third scenario was that six- to nine-month-old infants rely on multiple available cues (e.g. semantic, contextual, frequency) to learn and recognize words; thus, infants would display a more robust recognition when more cues are congruent with one of the items, as is the case for matching labels (semantic, contextual and frequency congruence), than when they can only rely on a single cue, as is the case for related words (only context or frequency congruence). In that case, infants would look longer at the target object when hearing the target's label, compared with when they hear the name of contextually or/and frequency-related objects; yet, their performance on related words would be significantly above chance.

# 2. Methods

## 2.1. Participants

Data collection strictly followed the procedures announced in the preregistered report (see https://osf.io/gj8u9/?view_only=a4f61a751c4b478a814db3a54ac51ead). Data collection stopped when we reached the required sample size of 50 (with the parameters *effect size*, *alpha (sig.level)* and *test type* set to Cohen's *d* = 0.41 [1,5],[2] 0.05 and one-sample two-sided, respectively, this sample size should lead to 0.80 power). Data from 50 monolingual, healthy, full-term infants between six and nine months of age (mean = 7.07 months, range from 6.05 to 8.77 months, girls *n* = 29) living in Oslo and its surroundings were retained for the analyses.[3] Additional 34 infants were recruited, but their data were not included in the final sample (see Data pre-processing section). The eligibility criteria were the following: (i) the child was born full term (gestational weeks greater than 37) and was between six and nine months of age (184–274 days, similar to [19]); (ii) the child is exposed to 80% Norwegian (Oslo dialect) or more at home; (iii) both parents speak Norwegian to the child and (iv) the child has no developmental delays and no history of chronic ear infections. The participants were recruited from the National Registry (Folkeregister). Informed consent was obtained from the parents before the test. The study has been approved by the Norwegian Centre for Research Data (NSD).

## 2.2. Stimuli

### 2.2.1. Pictures

Sixteen pictures presenting familiar objects from various categories (e.g. cookie, belly, cat, keys) were selected for the study. Given previously reported effects of typicality on word recognition [9], the pictures were assessed for typicality by five native Norwegian speakers on a scale from 1-low to

[2]Given that the original study [1] does not report sufficient information required to calculate the standardized effect size, we rely on the results of similar studies [2,5], which provide it. The first study examined whether infants manifested looking preference for the target upon hearing its name if the target and the distractor images were contextually related (e.g. juice and milk) or not (e.g. juice and foot). The study revealed that, on unrelated pairs of images, infants looked significantly more than chance at the labelled image (*M* = 0.044, s.d. = 0.108), whereas on related pairs of images, they performed at chance (*M* = −0.013, s.d. = 0.15), suggesting that infants disambiguate between two words, if these are sampled from distinct categories; yet they show no preference if the two items belong to the same semantic category. The second study [5] examined word comprehension in six-month-old infants by presenting them with the videos of a hand and a foot and asking infants to look at either of the videos. The authors reported that infants looked significantly more than chance at the matching label (*M* = 0.036, s.d. = 0.08).

[3]Data for 12 of these participants were collected before the date of the in principle acceptance (IPA) of the Stage 1 of the preregistered report. We certify that (1) the experimental protocol was absolutely identical for the data already collected before the IPA Stage 1 date and the data that was collected after the IPA date and (2) we had not inspected any of the already-collected data in any way prior to the date of Stage 1 IPA.

(a)

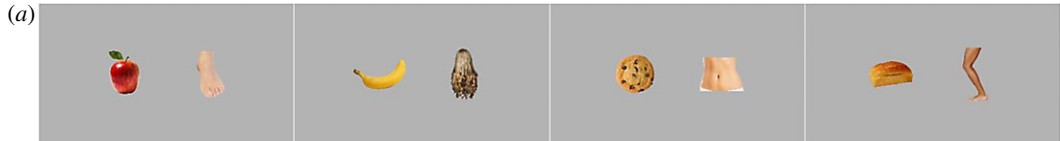

matching words: apple–foot, banana–hair, cookie–belly, bread-leg
context–related words: cup–pants, bottle–toothbrush, spoon–bathtub, table–diaper

(b)

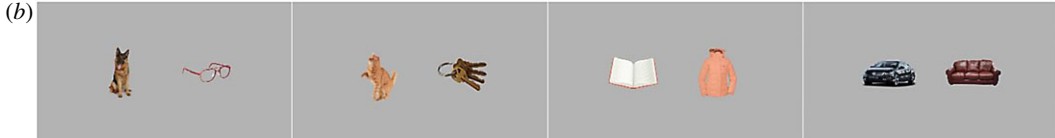

matching words: dog–glasses, cat–keys, book–jacket, car–couch
frequency–related words: pacifier–pillow, ball–sun, phone–moon, water–carpet

**Figure 1.** Picture-pairs and matching and related words used in context (*a*) and frequency (*b*) conditions. In the context condition, the images on the left and on the right side of the picture-pair show food/kitchen-related and bathroom/cleaning-routine-related objects, respectively. In the frequency condition, the images on the left and on the right side of the picture-pair represent high- and low-frequency words, respectively (see table 1 for details). In each condition, every object appeared once on the left and once on the right side of the screen.

5-high: all 16 pictures were unambiguously recognized as representing the words they were meant to represent (all scored 5). Each object was depicted on a light-grey background 16.9 by 12.7 cm (as in the BS12 study [2]). Pictures were assembled in eight picture-pairs (figure 1). Four of them, featuring food and bathroom/cleaning routine-related objects (cookie–belly, banana–hair, apple–foot, bread–leg), were used in the context condition; while the other four, featuring objects, whose labels differ in frequency of occurrence, were sampled from distinct semantic and contextual categories (dog–glasses, cat–keys, car–couch, jacket–book) and were used in the frequency condition (figure 1). In the context condition, one object in each picture-pair was related to the (spatial) context of kitchen and food and the other one was related to the context of body (washing) and/or to the bathroom routines. In the frequency condition, the two objects within each picture-pair belonged to distinct contextual and semantic categories. Crucially, their corresponding labels differed on their frequency of occurrence in child-directed speech (see Words section for details). The objects within each pair were edited so that their relative brightness and size were approximately the same. The eight picture-pairs were laid out on a light-grey background 51 by 28 cm, i.e. the size of the experimental screen used for the study (as shown in figure 1). In order to counterbalance the side of object presentation, an additional set of eight picture-pairs was created by switching the side of the objects within each pair.

### 2.2.2. Words

Thirty-two words were used in the study. Sixteen of them are the labels of the 16 objects used in the context (*n* = 8: cookie–belly, banana–hair, apple–foot, bread–leg) and in the frequency (*n* = 8: dog–glasses, cat–keys, car–couch, jacket–book) conditions. These words were used on matching trials in their respective conditions. Among the other 16 words, eight of them (spoon–bathtub, bottle–toothbrush, cup–pants, table–diaper), referring to food and bathroom/body-cleaning objects, were used on related trials in the context condition. The other eight words (pacifier–pillow, ball–sun, phone–moon, water–carpet) are frequency related to the words used in the frequency condition; thus, they were used on related trials in the frequency condition (figure 1). To prevent partial disambiguation based on word phonology, words within each pair (the labels) and the two related words either had different onset phonemes (e.g. /bɑnɑːn/, /hoːr/, /flɑske/, /tɑnbœʃtə/ for banana, hair, bottle and toothbrush) or all shared the same onset (as it is the case for one pair only: /brøː/, /bæjn/, /buːr/, /blæjə/, for bread, leg, table and diaper). See table 1 for a full description of the word pairs used in each condition.

In the context condition, the words referring to items within each picture-pair (as well as the contextually related words) were selected so that the objects could not be disambiguated by relying on an unbalance in the frequency of occurrence of the matching labels in child-directed speech. Importantly, contextually related words were selected to match the spatial context of the target words (see table 1 for further details). In order to validate our context assignments, 14 parents, having children of ages from six months to 6 years, filled in a 5-force-choice[4] questionnaire in which they had

**Table 1.** Features of the word pairs used in the context and frequency conditions. Note: The measure 'frequency' reflects the frequency of word occurrence in the analysed Norwegian corpus of child-directed speech CHILDES from [10,20,21], available in htttps://github.com/mikabr.

**context condition**

*referent words: words referring to the displayed objects*

| | cookie–belly | | banana–hair | | apple–foot | | bread–leg | |
|---|---|---|---|---|---|---|---|---|
| English translation | cookie–belly | | banana–hair | | apple–foot | | bread–leg | |
| Norwegian words | kjeks-mage | | banan-hår | | eple-fot | | brød-bein | |
| phonetic transc. | çeks | mɑːgə | bɑnɑːn | hɔːr | eplə | fuːt | brøː | bæjn |
| frequency | 20,0 | 6,0 | 107,0 | 47,0 | 40,0 | 11,0 | 26,0 | 16,0 |

*related words: words related contextually to the displayed objects*

| | spoon–bathtub | | bottle–toothbrush | | cup–pants | | table–diaper | |
|---|---|---|---|---|---|---|---|---|
| English translation | spoon–bathtub | | bottle–toothbrush | | cup–pants | | table–diaper | |
| Norwegian words | skje-badekar | | flaske-tannbørste | | kopp-bukse | | bord-bleie | |
| phonetic transc. | ʃeː | bɑːdəkɑr | flaskə | tɑnbœʃtə | kɔp | buksə | buːr | blæjə |
| frequency | 6,0 | n.a. | 6,0 | 8,0 | 17,0 | 18,0 | 4,0 | 34,0 |

**frequency condition**

*referent words: words referring to the displayed objects*

| | dog–glasses | | cat–keys | | book–jacket | | car–couch | |
|---|---|---|---|---|---|---|---|---|
| English translation | dog–glasses | | cat–keys | | book–jacket | | car–couch | |
| Norwegian words | hund-briller | | katt-nøkler | | bok-jakke | | bil-sofa | |
| phonetic transc. | hʉn | brilːər | kɑtː | nøkler | buːk | jɑkːe | biːl | suːfɑ |
| frequency | 52,0 | 1,0 | 35,0 | 1,0 | 107,0 | 15,0 | 137,0 | 2,0 |

*related words: words having the frequencies similar to the names of the displayed objects*

| | pacifier–pillow | | ball–sun | | phone–moon | | water–carpet | |
|---|---|---|---|---|---|---|---|---|
| English translation | pacifier–pillow | | ball–sun | | phone–moon | | water–carpet | |
| Norwegian words | smokk-pute | | ball-sol | | telefon-måne | | vann-teppe | |
| phonetic transc. | smuk | pʉːtə | bal | suːl | teləˈuːn | moːnə | vɑn | tepə |
| frequency | 14,0 | 2,0 | 96,0 | 14,0 | 41,0 | 2 | 255,0 | 14,0 |

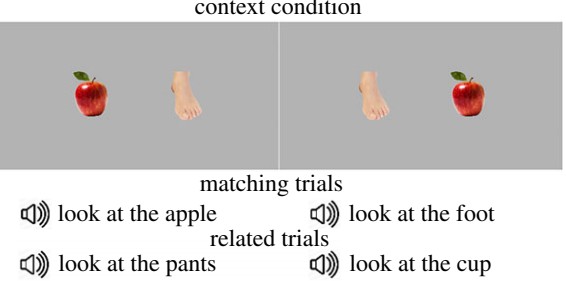

**Figure 2.** Visual stimuli and audio sentences used for matching and context-related trials for the picture-pair 'apple–foot' used in the context condition. On each trial, only one picture-pair was shown on the screen.

to choose, for each word ($n = 16$), the context in which they believed a 0–12-month-old infant was most likely to encounter it. The results confirmed our initial assignments for both contexts, with 96 and 83% of parents assigning the test words to the expected respective kitchen/food and bathroom/cleaning-related routines contexts. Crucially, none of the tested words from the kitchen-related set of items was assigned to the bath/cleaning-related routines, and vice versa. Similarly, using the University of South Florida word association database we assessed each pair of matching and related words [22]. Within each word pair, none of the words was listed as associated with the other word (e.g. for a pair apple–cup, cup was not listed as being associated with the word apple, and vice versa).

In the frequency condition, words referring to items (and the frequency-related words) within each picture-pair were contrasted on their frequency of occurrence in Norwegian child-directed speech (retrieved from https://github.com/mikabr/aoa-prediction, see also [11,20,21]), with a minimal frequency ratio between the two words of 1 : 7; i.e. within each pair, the high-frequency word is heard at least seven times more frequently than the low-frequency word. Crucially, within each picture-pair, referent words and their frequency-related words referred to objects from different semantic categories and different contexts, consequently removing confounding biases at semantic and/or contextual levels.

### 2.2.3. Audio stimuli

In order to minimize the number of errors due to parental mistakes, but also to control timing across infants, we used recordings of a Norwegian female speaker to prompt infants' search (note that BS12 study used parental speech). A native Norwegian female speaker was recorded while reading at a slow speed and in a child-directed fashion four types of sentences: 'Can you find the ⟨target⟩?', 'Where is the ⟨target⟩?', 'Do you see the ⟨target⟩?' and 'Look at the ⟨target⟩!'. The same sentences were used in the BS12 study [2]. Each of the sentences was repeated eight times, for a total of 32 sentences. These sentence-frames were used to prompt infants' search. The same sentence-frame was used for the two words within a pair. For instance, in the context condition, for the picture-pair 'apple–foot', infants heard, on matching trials, 'Look at the apple!' and 'Look at the foot!', and on related trials they heard 'Look at the cup!' and 'Look at the pants!' (figure 2). The following parameters were used for recordings: 16 bits, two channels, 44 kHz. All 32 sentences were subsequently normalized for amplitude: the mean amplitude of each utterance was set to 70 dB.

In order to approach our experimental procedure as closely as possible to the BS12 study, we inserted a 1.5 s period of silence at the beginning of each trial so that infants would have the same exposure to the visual stimuli before the onset of the sentence, as in the BS12 study. Trials ended 3.5 s after the target word onset (see figure 3 for an illustration). For the 'bread–leg' picture-pair, whose matching and related words share the same onset (/**b**rø:/, /**b**æjn/, /**b**u:r/, /**b**læjə/), the word onset was fixed to the onset of the second (disambiguating) phoneme. The length of the sentences varied from 5.4 to 6.2 s ($m = 5.7$ s).

### 2.2.4. Video files

The 32 audio files were combined with the eight picture-pairs to create 32 video files, which were used in the experiment. The duration of the videos was aligned to the duration of the audio files. Additional set of 32 video files was created in order to counterbalance the side of the object presentation.

---

[4]The available options were: kitchen/food-related context, indoor activities, outdoor activities, bathroom/cleaning routines, good-night routines.

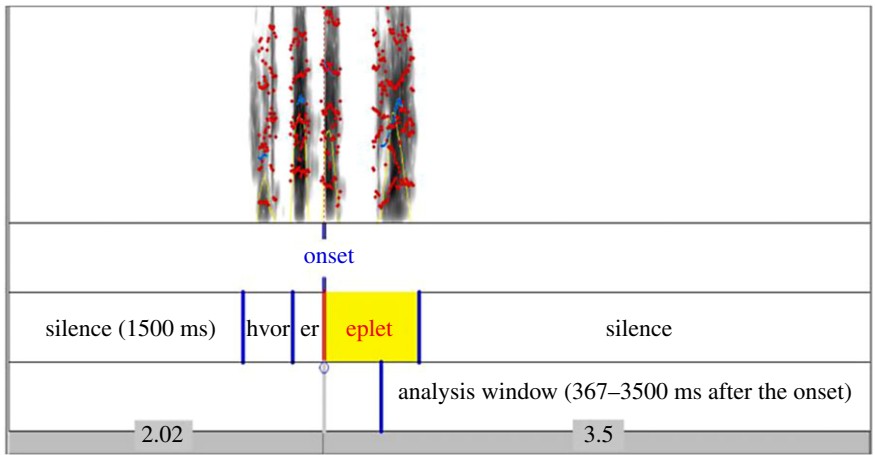

**Figure 3.** Illustration of an audio file used for the picture-pair 'apple–hair' on a matching trial for 'apple'. The sentence is 'Where is the apple?' – 'Hvor er eplet?'.

### 2.2.5. Control stimuli

Using the same procedure as above, we created four control video files. Two of them displayed a picture of a tree either on the left or on the right side of the screen, accompanied by the sentence prompt 'Look at the tree!'. The other two displayed a picture of a house, again, either on the left or on the right side of the screen, accompanied by a sentence prompt 'Where is the house?'.

### 2.2.6. Attention getters

A video of a spinning colourful flower displayed in the centre of a light-grey screen (the same shade as for the experimental stimuli) and accompanied by a tinkling attractive sound was used to get infants' attention and to centre their gaze between each trial. The length of the video is 10 s.

## 2.3. Procedure

First, the experimenter, a female Norwegian native speaker, received the families in a large laboratory room and briefly explained the study to them. Then, she obtained the consent form and accompanied the parent with the child to the experimental, dimly lit, room. The child was sitting on a car seat facing the experimental computer screen fitted with an eye-tracker base. Parents were sitting behind their child and wore a visor and sound-attenuating headphones through which they heard masking music (jazz music with overlaid spoken human conversations). The parents were asked not to point to the screen, shift their bodies, or move their chairs during the experiment. The experimenter was sitting in the same room, next to the experimental screen; however, a thick curtain separated the experimenter from the family. Therefore, neither the child nor the parent was able to see the experimenter. We collected infants' gaze using a Tobii TX300 eye-tracker, which has a sampling rate (binocular) of 300 Hz and a screen resolution of 1920 × 1080 pixels. The stimuli were presented at the average amplitude of 65 dB through two speakers, positioned at the left and right sides of the screen. The experimenter was able to monitor the infants' looking behaviour via the Tobii Live Viewer tool operating on the control screen, situated in front of the experimenter.

The experiment started with a 5-point calibration procedure, which was followed by the experiment. There were two experimental blocks, i.e. one for the context and one for the frequency condition, with 16 trials in each. The order of conditions was randomized across participants and there were no pauses between the conditions.

On each trial, infants saw two pictures displayed on the right and left sides of the screen and heard, after 1.5 s, a target sentence prompting them to look at either of the objects presented on the screen or to a related object (which is not presented on the screen). The pictures remained on the screen for 3.5 s after the target-word onset. Before each trial, an attention-getter video was displayed on the centre of the screen. The length of the attention-getter video was controlled manually: the experimenter stopped it after an infant had continuously fixed the attention getter for one second.

In the context condition, in a half of the trials ($n = 8$), infants saw, twice, each of the four picture-pairs from the context condition (figure 1) accompanied either by the name of one of the object or the other. These were the matching trials. In the other half of the trials ($n = 8$), infants saw, twice, the same four pairs of pictures used for matching trials, while hearing contextually related words either to one object in the pair, or to the other (e.g. 'spoon' for 'cookie', as both are likely heard in the kitchen; 'bathtub' for 'belly', as both are typically heard in the context of body-cleaning routines; see figure 2 for an illustration). In the frequency condition, infants saw the four picture-pairs from the frequency condition, accompanied, similarly, either by their matching labels (eight trials) or by their frequency-related words (eight trials); for example, on the related trials, they heard 'water' for 'car' as both are high-frequency words and 'pillow' for 'glasses', as both words are low-frequency words in infant-directed speech (figure 1).

Each picture-pair, therefore, was seen by the infant four times: two on the matching trials and two on the related trials; however, within each matching and related pair, the sides of the object presentation were counterbalanced (figure 2). So, in each condition, every object appeared once on the left and once on the right side of the screen. To counterbalance the side of the object presentation across trials and conditions, two presentation lists were created. In each list, the target object appeared twice on each side (once on matching and once on related trial). The order of trials within each block was pseudo-randomized. Also, the order of different sentences-frames was pseudo-randomized across trials. Infants were randomly assigned to one of the two experimental lists. The design was within-subject: each infant saw all pictures from both conditions.

In order to make sure that infants are fully engaged in the task and that their performance arises from the experimental manipulation(s), we included four control trials, in which infants saw familiar objects (e.g. a house) which appeared either on the left or on the right sides of the screen, preceded by a prompting phrase produced in a child-directed register. These stimuli were functionally similar to attention getters, which are commonly used in infant research due to their efficacy in capturing infants' attention and gaze: infants reliably fix the objects when they appear on the screen. At the beginning of each control trial, infants' attention was centred on the middle of the screen using an attention-getter (as during the experimental trials). There were two control trials in each block; they were displayed at regular intervals, after every eighth trial. Presentation sides were counterbalanced. During the experiment, the experimenter took notes and reported, on a trial-per-trial basis, potential experimental or other types of errors: for example, if parents or a third person interfered with the test, if there was a technical problem, or if the child cried (see Data exclusion section).

After the experiment, parents were asked to fill in three questionnaires, which allowed us to collect infants' data on (i) their linguistic and general background (sex, maternal education, age, number of siblings, language exposure), (ii) their receptive vocabulary using the McArthur CDI for Norwegian [17] and (iii) their familiarity with the words used in the experiment. The word familiarity questionnaire asked, for each of the 32 words used in the experiment, how frequently parents have used it (on a scale from 0-never to 5-very frequently) while interacting with the child or in his presence, since their baby was born. The results of this questionnaire were used in the analyses of the gaze data (see Data exclusion section for detail). At the end of the experiment, parents were able to choose a small gift for their infant (e.g. a body, a T-shirt) and were reimbursed for travel costs. The experimental protocols and the materials were uploaded to the Open Science Framework depository https://osf.io/gj8u9/?view_only=a4f61a751c4b478a814db3a54ac51ead.

## 2.4. Data exclusion

The data were collected from the eye-tracker and pre-processed. The following criteria were used to exclude infants based on their behaviour on the experimental trials: (i) failed calibration of the eye-tracker; (ii) software problem (e.g. technical reasons: software stops displaying images or playing sounds for more than 50% of the trials); and (iii) the child did not contribute to at least 20% of experimental trials due to fussiness (less than 0.5 s of fixation on the screen at post-naming period) and/or audible crying that led to experiment termination. Then, we performed a quality check by looking at the data on the control trials for each infant. A failure to fixate the target in the control trials implied that an infant was not at all engaged in the task, hence, his/her performance in other conditions is not reliable or informative. In this case, the data from the child was excluded.

After the quality check, the following criteria were used to exclude single trials: (i) no looking at either image for at least 0.5 s in the post-naming period, and (ii) no looking was recorded in the pre-naming period (as in [1]). Also, we excluded trials in which the experimenter had reported that (iii) the parent interfered (e.g. pointed to the screen, shifted his/her body or moved his/her chair), or (iv) the trial

was interfered by a third person or due to a technical error. Finally, similar to [5], individual picture-pair trials were removed from individual child data if the parents reported that they did not use one (or both) of the produced words with the child (or in his presence), since the child was born. For example, if a parent did not use the word 'apple', then all 'apple–foot' picture trials were removed from his/her child analyses.

# 3. Results

## 3.1. Data pre-processing

Data pre-processing and analyses strictly followed the procedures announced in the preregistered report (see https://osf.io/gj8u9/?view_only=a4f61a751c4b478a814db3a54ac51ead). In total, data from 50 infants of six to nine months of age (mean = 7.07 months, range from 6.05 to 8.77 months, girls $n = 29$) were retained for the analyses. Additional 34 infants were recruited, but their data were not included in the final sample, for the following reasons. Twelve of them did not pass the infant inclusion criteria: the child was born preterm ($n = 4$), had a neuro-disease ($n = 1$), was bilingual ($n = 1$), did not complete the study ($n = 3$), or the experimenter was unable to calibrate the eye-tracker for the child ($n = 3$). Twenty other infants did not pass the trial inclusion criteria, i.e. they were not able to contribute to at least 20% (6/32) of the experimental trials, either because they were not looking at either image in the pre-naming period or they were not looking at either image for at least 0.5 s in the post-naming period, as in [1]. In addition, two infants did not pass the trial inclusion criteria, because they were unfamiliar with the words depicted in six out of eight picture-pairs; their parents reported to never have used them with the child (or in his/her presence), since the child was born. All infants in the final sample attended to the targets on the control trials (see Control quality check below), suggesting that they were engaged in the task. Note that, in the final sample, there were fewer eight- to nine-month-old infants ($n = 13$) than six- to seven-month-olds ($n = 37$).

In the final sample of 50 infants, we included only experimental trials that followed the preregistered inclusion criteria: first, that for each trial infants looked at either image in the pre-naming period, and at least 0.5 s in the post-naming period; and, second, that all target (heard) words were familiar to infants, as reported by their parents, similar to [4]. If parents reported that they have never used one (or both) of the target matching words with the child (or in his presence) since the child was born, then we excluded all trials (matching and related), that were associated with this target word. For example, if a parent did not use the word 'apple', then all 'apple–foot' picture trials (matching and related) were removed from his/her child analyses. However, if parents reported never using a related word, as, for example, 'pants', then we removed only condition-related trials, that is, 'cup' and 'pants', but kept the matching trials 'apple' and 'foot'. Raw and processed data, digital materials/in-home scripts and the laboratory log can be found in https://osf.io/xbjyt/?view_only=26ba5854a8134967af8509eb5f2a2a43.

## 3.2. Planned statistical analyses

### 3.2.1. Dependent variables

To align our analyses to those reported in BS12 [2], to evaluate each of our hypotheses and to compare the effect sizes with those obtained in previous research, as preregistered, we adopted the same analysis strategy (e.g. dependent variables, type of statistical tests, transformation of data when needed) used in BS12 and in more recent studies [1,8] having similar experimental designs and infants of the same age [1]. Two naming windows were identified on each trial (similar to [1,2]: pre-naming (from the start of the trial to the target word onset, around 2000 ms) and post-naming (367–3500 ms after target onset), see figure 3 for a schematic illustration. The target or the distractor areas of interest were limited to an invisible 800 × 680-pixel rectangle around each object.

The dependent measure *difference in (target) looking proportion* was computed at the post-naming window for each picture-pair (e.g. objects A and B) as the difference in the proportion of looks at object A between the trials when A was a target and when A was the distractor. Then, for each condition and trial type, the differences in proportion scores were averaged over item pairs to obtain *average differences in looking proportion* for the by-subject analysis and over subjects for the by-item analysis. Given that the dependent measure was computed for picture-pairs, we discarded picture-pairs for which we did not have data in one of the trials. For example, if a child did not provide

sufficient looking time during a trial where the word 'apple' was the target, then all 'apple–foot' picture trials were removed from the child's analyses. Such data analysis, used in BS12 [2], resulted in a substantial data loss in our sample (a total of around 400 trials); in total, we obtained average scores for 49 participants for the matching trials and for 45 participants for the related trials. The distribution of the average proportion scores was tested for normality; the results of Shapiro–Wilk test revealed that the data were not normally distributed ($p < 0.0006$), hence, non-parametric tests were used for the statistical analyses. All the analyses were performed with the R software [23].

### 3.2.2. Control quality check

We computed, for each participant, an average fixation time at the target picture over the four trials in the control condition (pictures 'house' and 'tree'). The results revealed that all participants attended the targets upon their appearance on the screen, suggesting that they were engaged in the task and fixated the target when it appeared on the screen (average data can be seen in 'Participants_info_70sbj.xlsx' file https://osf.io/xbjyt/?view_only=26ba5854a8134967af8509eb5f2a2a43). On average, participants fixated the target for 1.36 s, that represented 25% of the total trial duration (approx. 5.5 s). Second, as announced in the preregistered report, we computed, for each participant, an *average proportion of looks at the target at post-naming window*.[5] One-sample one-tailed *t*-test against zero on the average proportion measure revealed a significant effect ($n = 50$, $\mu = 0.90$, s.d. = 0.13, 95% CI [0.87, 0.94], $V = 1275$, $p < 0.000001$, $BF_{10} = 7.1$, *Cohen d* = 6.92)[6], indicating that infants were engaged in the task and that their performance was driven by their analysis of the visual stimuli used in the task.

### 3.2.3. Paired tests

As preregistered, to examine whether six- to nine-month-old Norwegian infants looked significantly above chance at the target when hearing the matching label as compared with when hearing the name of the distractor, we performed, for each condition, one-sample Wilcoxon signed rank one-tailed tests against chance on the *average differences in looking proportion* for the matching trials[7] [2]. The results of the by-subject analyses revealed that the average difference in looking proportion was not significantly different from zero in the context ($n = 37$, $\mu = -0.03$, 95% CI [−0.159, 0.096], $V = 262.5$, $p = 0.27$, *Cohen d* = 0.08, $BF_{10} = 0.20$) nor in the frequency ($n = 42$, $\mu = -0.04$, 95% CI [−0.164, 0.093], $V = 434$, $p = 0.83$, *Cohen d* = 0.08, $BF_{10} = 0.19$) conditions (figure 4). The analysis on the matching trials over the two conditions (as in [2]), revealed similar non-significant result ($n = 49$, $\mu = -0.005$, 95% CI [−0.107, 0.096], $V = 580.5$, $p = 0.94$, *Cohen d* = 0.014, $BF_{10} = 0.15$). By-item analysis also revealed non-significant results in the context ($n = 4$, $\mu = -0.087$, 95% CI [−0.280, 0.106], $V = 2$, $p = 0.37$, *Cohen d* = 0.72, $BF_{10} = 0.81$) and in the frequency ($n = 4$, $\mu = -0.016$, 95% CI [−0.049, 0.017], $V = 2$, $p = 0.37$, *Cohen d* = 0.75, $BF_{10} = 0.85$) conditions. These results suggest that Norwegian six- to nine-month-old infants do not recognize the familiar concrete words used in the study, contrarily to the results reported in BS12 [2] for American-English-learning infants. Note, that we carefully selected only words that parents reportedly 'used' when interacting with the child.[8]

Similar planned analyses were performed to assess whether six- to nine-month-old infants looked significantly above chance at the target when hearing contextually and frequency-related labels. The results of the by-subject analyses revealed that the average difference in looking proportion at a target while hearing a related word was not significantly different from zero neither in the context ($n = 39$, $\mu = 0.06$, 95% CI [−0.049, 0.176], $V = 397$, $p = 0.09$, *Cohen d* = 0.18, $BF_{10} = 0.54$) nor in the frequency ($n = 35$, $\mu = -0.08$, 95% CI [−0.226, 0.057], $V = 192.5$, $p = 0.18$, *Cohen d* = 0.20, $BF_{10} = 0.62$) conditions. Analogous result was found on the related trials over the two conditions ($n = 45$, $\mu = 0.01$, 95% CI [−0.088, 0.108], $V = 512$, $p = 0.32$, *Cohen d* = 0.029, $BF_{10} = 0.19$). By-item analysis also revealed non-significant results for related trials in the context ($n = 4$, $\mu = 0.075$, 95% CI [−0.167, 0.318], $V = 7$, $p = 0.31$, *Cohen d* = 0.49, $BF_{10} = 0.95$) and in the frequency ($n = 4$, $\mu = -0.055$, 95% CI [−0.276, 0.164], $V = 3$, $p = 0.62$, *Cohen d* = 0.40, $BF_{10} = 0.54$) conditions. These results suggest that Norwegian six- to nine-

---

[5]Trials with no looks at the screen during the whole trial duration were discarded from the analyses.

[6]Here and elsewhere, the packages *BayesFactor* [24] and *rcompanion* [25] were used to compute the Bayes factor and the confidence intervals, respectively.

[7]Note that when the average looking time was negative (which was not expected); therefore, two-tailed (rather than one-tailed) *t*-tests were performed.

[8]Similar results were obtained when we included words that parents reported as *not* used with the child.

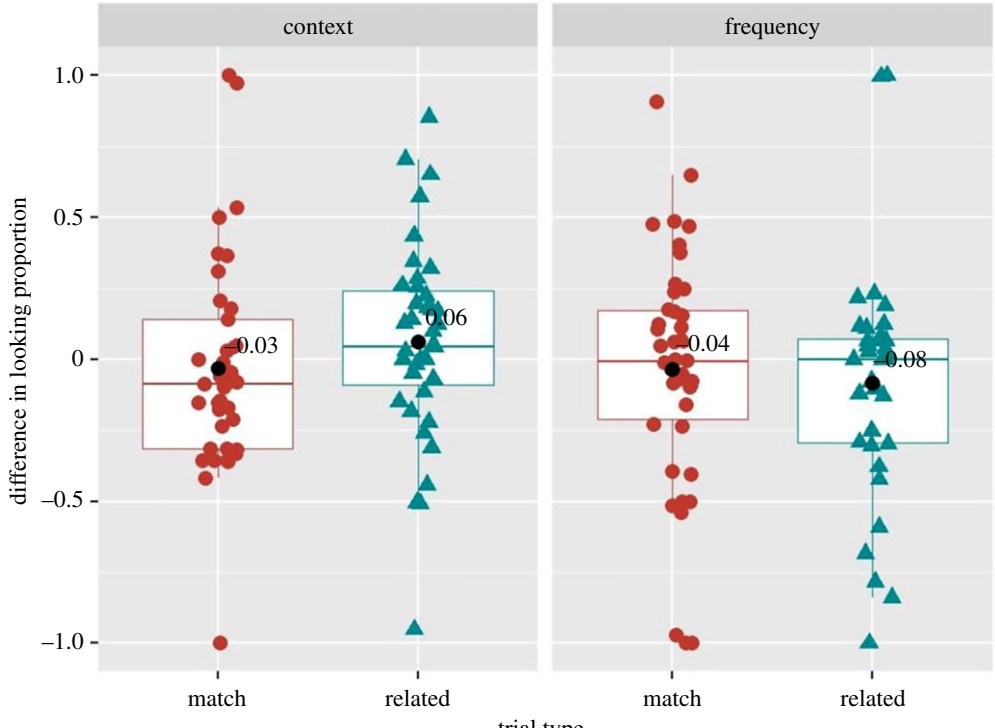

**Figure 4.** Difference in looking proportion on match and related trials in the context and frequency conditions in six- to nine-month-old Norwegian infants. Coloured shapes (red and blue) represent individual child data averaged over item pairs. Black circles indicate average values over all subjects in a given trial type and condition.

month-old infants do not rely solely on the context and frequency of word use to disambiguate between objects.

Given the absence of word comprehension for matching and related trials in both conditions, we do not report the planned analyses—between condition and multi-level regression—in the main manuscript (they can be found at https://osf.io/gj8u9/?view_only=a4f61a751c4b478a814db3a54ac51ead, though). However, we proceed with the cluster permutation analysis that allows to take into account the time course of the infant looking pattern and to identify potential differences in the dynamics of the looking pattern between trials and conditions.

### 3.2.4. Cluster permutation analysis

As announced in the preregistered report, we performed cluster permutation analyses [26] that can provide important insights into the dynamics of word recognition in young infants and, potentially, reveal the time windows in which infants looked longer at the target as compared with the distractor on matching and related trials. In addition, they can reveal the time windows during which there were significant differences between matching and related trials. A new dependent measure, the *target proportion* looking metric, was computed, in 50 ms time bins, as the proportion of target looking [target/(target + distractor)] from the beginning of the trial to its end. Given that this measure is calculated on each trial, i.e. for each target word (as compared with the *increase in target looking*, that is computed for word *pairs*), considerably more trials per subject were included to compute the average by-subject measure ($n = 315$ in the previous analyses and $n = 895$ in the current analyses). Similar to the previous analyses, we removed those words that parents have reported as never used with their child.

For each condition, we conducted three cluster-based permutation analyses, similar to [27]. We compared the average proportion of looks at the target object to 50% (the chance) on matching and related trials, and we compared the looking proportions between the matching and related trials. All *target_prop* were transformed via the arcsin square function to align with the *t*-test assumptions. Time bins with significant effect ($t > 2$, $p < 0.05$) were grouped into a cluster. Then, we computed the size of the cluster, as the sum of all *t*-values within this cluster, and tested the probability of observing a cluster of the same size by chance. For that, 1000 simulation analyses were performed, where the type

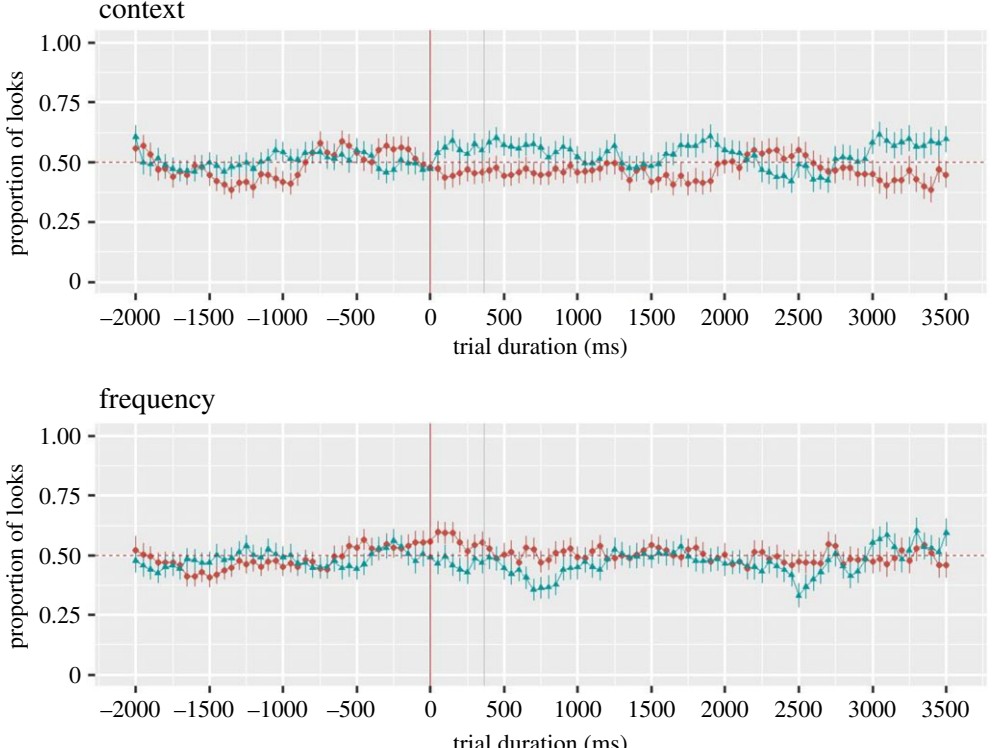

**Figure 5.** Time course of looking behaviour (in 50 ms bins) in the context and frequency conditions. Matching trials are red circles and related trials are blue triangles. Dots represent the mean and the bars represent the standard error of the mean (averaged over item pairs). The red vertical line indicates word onset, the grey vertical line indicates 367 ms from word onset.

of word (matching versus chance, related versus chance, matching versus related) were assigned randomly for each speaker. Then, for each simulation, we computed the size of the biggest cluster using the same procedure as the one that we used to compute the size for the real data. If the probability of observing a cluster of this size or bigger was smaller than 5% ($p < 0.05$), we concluded that the differences (matching versus chance, related versus chance, matching versus related) were significant. The analysis window was restricted to a window from −2000 ms before word onset to the end of the trial. Since no previous study has analysed the time course of word recognition in six- to nine-month-old infants, our expectations were based on a visual analysis of the time course plots provided in some recent studies on word comprehension in six- to nine-month-old infants: they suggested that the significant window for matching/related versus chance would be approximately between 367 and 2000 ms after the target onset [2]. The same temporal window is expected for matching versus related trials, as suggested by the results in 12–14-month-old infants [8].

For each condition and trial type, we identified those cluster sizes of $t$-values that were superior to 2; yet, the results of the cluster permutation analyses did not reveal that the probability of observing a cluster of a bigger size was smaller than 5% for any of them (figure 5). The biggest clusters were obtained in the frequency-related condition (−7.81 [2450–2600 ms] and −11.33 [650–850 ms], albeit with $p > 0.1$).

### 3.2.5. Interim discussion of the planned analyses

The planned paired tests and cluster permutation analyses revealed that, overall, 50 six- to nine-month-old Norwegian infants did not recognize the 16 familiar words tested in the current study: their proportion of looks at the target on matching trials was not different from chance in both frequency and context conditions. These results showed that, unlike American-English speaking six- to nine-month-old infants, six- to nine-month-old Norwegian infants, tested in our study, failed to disambiguate between two items sampled from distinct semantic categories (and contexts), suggesting that they have not yet established mappings between familiar objects and referent words. Analyses of the related trials revealed that Norwegian six- to nine-month-old infants cannot rely solely on contextual or frequency cues to disambiguate between objects; thus, our results provide no evidence

for the hypothesis that six- to nine-month-old Norwegian infants rely exclusively on extra-linguistic cues to disambiguate between objects.

Note, however, that there were only 13 eight- to nine-month-old infants in the final sample of 50 infants and our sample was slightly younger than in BS12 (mean ages were 7.07 and 7.45 months, respectively); is it possible that the absence of word recognition was driven by poor performance of the more numerous six- to seven-month-old infants? We examined this possibility in additional exploratory statistical analyses and assessed the developmental trajectory of word comprehension in six- to nine-month-old Norwegian infants (ranging from 184 to 267 days) by including age (in days) as a covariate, similar to [8]. Other studies (and in a different English dialect, i.e. British English) have shown that infants recognize familiar words at nine months of age [4]. As far as Norwegian language is concerned, parental reports suggest that infants understand, on average, 20 words at eight months of age and 32 words at nine months of age [16,17]. We hypothesized that word comprehension would be modulated by age, with older infants (close to their 9th-month 'birthday') showing significant increase in target looking. This hypothesis was tested in additional exploratory analyses on the original sample of 50 infants and on a larger sample with additional 20 infants (see section Statistical analyses on the original sample size with additional 20 infants ($n$ total = 70)).

## 3.3. Exploratory statistical analyses

### 3.3.1. Mixed-effect regression analyses with age as covariate

To examine the developmental trajectory of word comprehension in six- to nine-month-old Norwegian infants, we performed mixed-effect regression analyses that take into account variability between infants and word pairs; we used R's lme4 [28] and lmerTest [29] packages. The fixed factors were trial (match versus related), condition (context versus frequency) and their interaction; age (in days, mean-centred) was included as a covariate. By-infant and by-word-pair intercepts were included as random factors. The dependent variable *difference in looking proportion* that was used in the paired analyses (but not averaged over subjects here, so we had $n = 315$ trials) was log transformed for the mixed-effect regression analysis. In order to handle the negative values, a constant value 2 was added to the data prior to applying the log transform. The anova function was used to assess the model simple effects. The results revealed a significant effect of age ($F_{1,307} = 5.72$, $p = 0.017$, $R^2 = 0.27$)[9], indicating that older infants showed an increase in target looking when compared with younger infants (figure 6). All other effects were not significant ($p > 0.1$, see section 'Exploratory analyses' in the supplementary document analyses_results_code.pdf for exact $p$-values and statistical details https://osf.io/gj8u9/?view_only=a4f61a751c4b478a814db3a54ac51ead).

### 3.3.2. Statistical analyses on the original sample size with additional 20 infants ($n$ total = 70)

#### 3.3.2.1. Mixed-effect regression analyses with age as covariate

Additional 29 six- to nine-month-old infants were tested; however, their data were not included in the planned analyses, because they took part in the present study after having completed a task for a different study.[10] Their data were pre-processed strictly using the criteria applied for the 'main' sample. As a result, data from six infants (two bilinguals, two preterm, one too old (296 days) and one refused to participate) were excluded based on the infant exclusion criteria, while data from the other three infants were discarded based on the trial exclusion criteria (they were not able to contribute to at least 20% (6/32) of the experimental trials). Furthermore, we excluded those picture-pairs for which we did not have data for one of the trials or the child was not familiar with the item ($n = 2$). All remaining data validated the control quality check, i.e. participants attended targets on

---

[9]The effect size estimate, $R^2$, was computed on a simpler model with the fixed factor of interest (age) as the only predictor, using the r.squared LR function from the MuMIn package [30].

[10]After we had handed the first revision of our preregistered manuscript, we started data collection for a different study, i.e. MANYBABIES1, an international collaborative research project on infant preference for infant-directed speech over adult-directed speech [19]. Given that the infant age range and the recruitment/exclusion criteria in the MANYBABIES1 study were identical to those announced in our preregistered study, we decided to use this opportunity and to test 'MANYBABIES1' infants also in our study. All 29 infants performed MANYBABIES1 test first and, after a pause, took part in our preregistered word comprehension test. We confirm that the experimental protocol for the additional 29 infants was absolutely identical to the one used in the current study and that we had not observed any of the already-collected data in any way prior to the date of Stage 1 of the in-principle acceptance of the preregistered report.

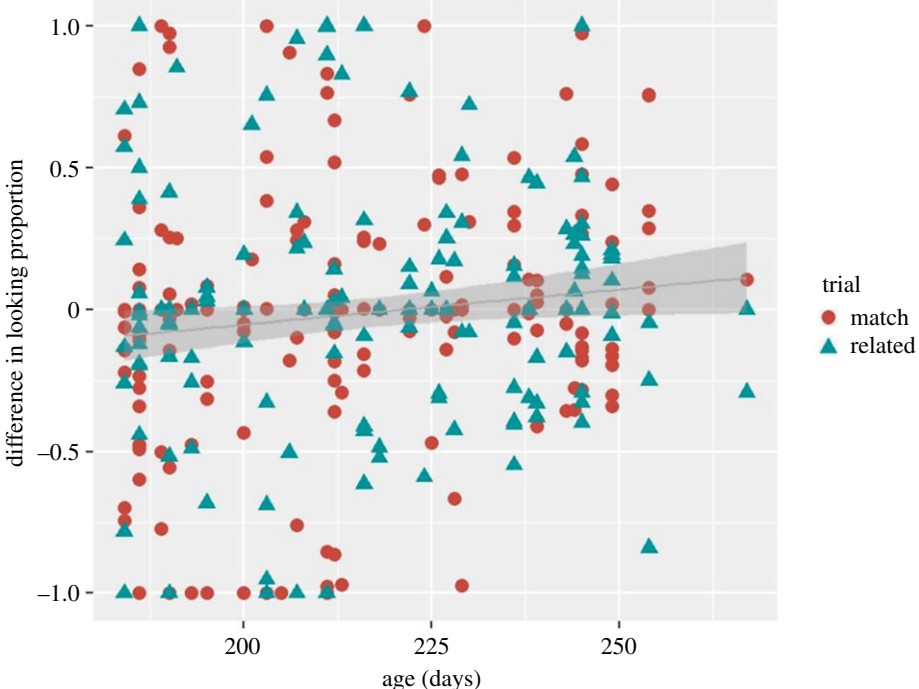

**Figure 6.** Subjects' ($n = 50$) raw difference scores on matching and related trials as a function of their age. The age of 225 days corresponds to 7.5 months. The grey line represents the regression line and the shaded area depicts 95% confidence region.

control trials. For the remaining data from 20 infants ($\mu = 7.36$, range from 6.05 to 9.04 months), we computed the dependent measure *difference in looking proportion* using exactly the same formula as for the main data analyses. There were 128 trails in total. Note, that 11 out of the 20 participants were eight- to nine-month-old infants (more than 230 days).[11] We combined these data with the original sample size and obtained a new sample size of 70 six- to nine-month-old infants ($\mu = 7.30$, range from 6.05 to 9.04 months). Identical to the above mixed-effect regression analyses were performed on the *difference in looking proportion* measure ($n$ pairs $= 443$). The results revealed a significant trial by age interaction ($F_{1,435} = 6.77$, $p = 0.0095$, $R^2_{adj} = 0.43$; figure 7), indicating that looking time at the target picture increased with the child's age on matching trials ($F_{1,231} = 7.92$, $p = 0.0053$, $R^2_{adj} = 0.31$), whereas it was stable on related trials (as revealed by follow-up mixed-effect regression analyses run on the related trials only, $F_{1,205} = 0.89$, $p = 0.34$, $R^2 = 0.0$). All other effects were not significant ($p > 0.1$, see section 'Exploratory analyses' in the supplementary document analyses_results_code.pdf for details on the analyses https://osf.io/gj8u9/?view_only=a4f61a751c4b478a814db3a54ac51ead). There was a marginally significant trial by condition interaction ($F_{1,435} = 3.53$, $p = 0.060$, $R^2_{adj} = 0.18$), but follow-up simple tests run with the lsmeans function [31] did not reveal any significant difference between the matching and related trials for both context and frequency conditions ($\beta = -0.039$, s.e. $= 0.033$, $t = -1.184$, $p = 0.236$ and $\beta = 0.048$, s.e. $= 0.033$, $t = 1.478$, $p = 0.140$).

### 3.3.2.2. Paired analyses

To understand the age by trial interaction, we split 70 infants into two age groups based on the median age of 230 days: six- to seven-month-olds, below the median age (184–229 days, $n = 46$) and eight- to nine-month-olds, above the median age (230–275 days, $n = 24$), and performed, separately for each group, one-sample Wilcoxon sign rank analyses on the *average difference in looking proportion*, similar to BS12 study and to our first paired analyses. Note, that these analyses were similar to those announced in the preregistered report, but performed separately for each age group. The results in six- to seven-month-old infants revealed that the average difference in looking proportion was not significantly different from zero on matching ($n = 44$, $\mu = -0.03$, 95% CI [$-0.132$, $0.067$], $V = 409$, $p = 0.45$, *Cohen d* $= 0.09$, $BF_{10} =$

[11]Note that given the age ranges of the infants recruited in our study, between 184 and 274 days or 6.05 and 9.04 months (similar to the MANYBABIES consortium study, our inclusion criterion for ages ranged between 184 and 274 days, see [19]), in the current study, we used the median of 230 days (7.5 months) to divide participants into two age groups, six- to seven-month-olds and eight- to nine-month-olds.

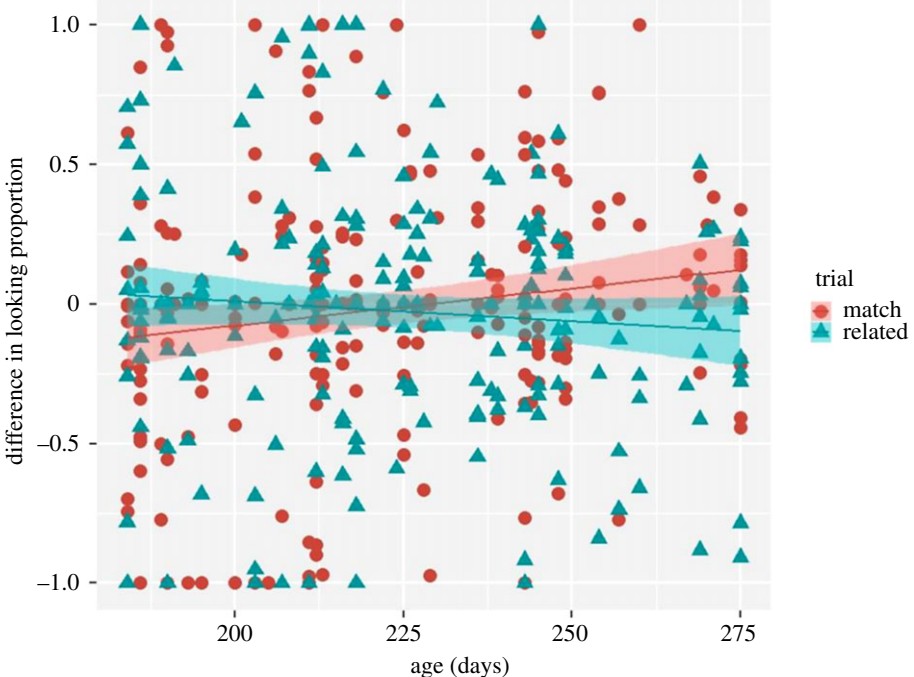

**Figure 7.** Subjects' ($n = 70$) raw difference scores on matching and related trials as a function of age. The age of 225 days corresponds to 7.5 months. The grey line represents the regression line and the shaded area depicts 95% confidence region.

0.19) and related ($n = 41, \mu = 0.01, 95\%$ CI $[-0.085, 0.113], V = 408, p = 0.40, Cohen\ d = 0.04, BF_{10} = 0.21$) trials. However, the results in eight- to nine-month-old infants revealed that the average difference in looking proportion was significantly different from zero on matching ($n = 23, \mu = 0.11, 95\%$ CI $[0.003, 0.220], V = 207, p = 0.024, Cohen\ d = 0.44, BF_{10} = 2.83$), but not on related ($n = 23, \mu = -0.06, 95\%$ CI $[-0.162, 0.041], V = 84, p = 0.28, Cohen\ d = 0.25, BF_{10} = 0.42$) trials (figure 8a). Bayes factor analysis [24] on the matching trials in eight- to nine-month-old group indicates that data presented moderate evidence for the alternative hypothesis; that is, Norwegian eight- to nine-month-old infants showed word comprehension for the 16 familiar words used in the current study. Given our *a priori* hypotheses, we examined which condition (context or frequency, or both) drove the overall effect on word comprehension in eight- to nine-month-old infants. One-sample Wilcoxon sign rank analyses revealed a significant effect in the frequency condition ($n = 20, \mu = 0.14, 95\%$ CI $[0.059, 0.211], V = 178, p = 0.0024, Cohen\ d = 0.84, BF_{10} = 56.32$), but not in the context ($n = 20, \mu = 0.05, 95\%$ CI $[-0.128, 0.234], V = 108, p = 0.305, Cohen\ d = 0.14, BF_{10} = 0.39$), suggesting that word comprehension in Norwegian eight- to nine-month-old infants is driven by accurate disambiguation in the frequency condition only (figure 8b). Item-pair analysis of the matching trials, however, did not reveal that the average difference in looking proportion was significantly different from zero either in the context ($n = 4, \mu = 0.037, 95\%$ CI $[-0.243, 0.319], V = 5, p = 0.56, Cohen\ d = 0.21, BF_{10} = 0.59$) or in the frequency ($n = 4, \mu = 0.084, 95\%$ CI $[-0.135, 0.304], V = 8, p = 0.18, Cohen\ d = 0.61, BF_{10} = 1.16$) condition (see figure 9 for the results for each item pair).

### 3.3.2.3. Relationship between frequency imbalance and effect size

Our analyses of word pairs used in BS12 study revealed a strong and significant correlation between the imbalance in frequencies between the two words in a pair and the word-pair effect size, suggesting that six- to seven-month-old English-learning infants exploit frequency imbalance to disambiguate between two words (see the Introduction section). In the current study, we also examined the relationship between the imbalance in frequencies between the two words in a pair and their effect size, by applying strictly the same analyses as those for the BS12 study. That is, we performed, separately for six- to seven- and eight- to nine-month-old age group, Pearson correlation analysis between the average difference in looking proportion,[12] for eight word pairs, and the absolute difference in their

---

[12]To align our analyses and dependent measures to those used in BS12 study, for the correlation analysis, we did not exclude those words that parents reported as not familiar to their child (similar to BS12 study). Note, that the direction and the strength of the correlation remained the same when these words were included (see §5.2 in the electronic supplementary material).

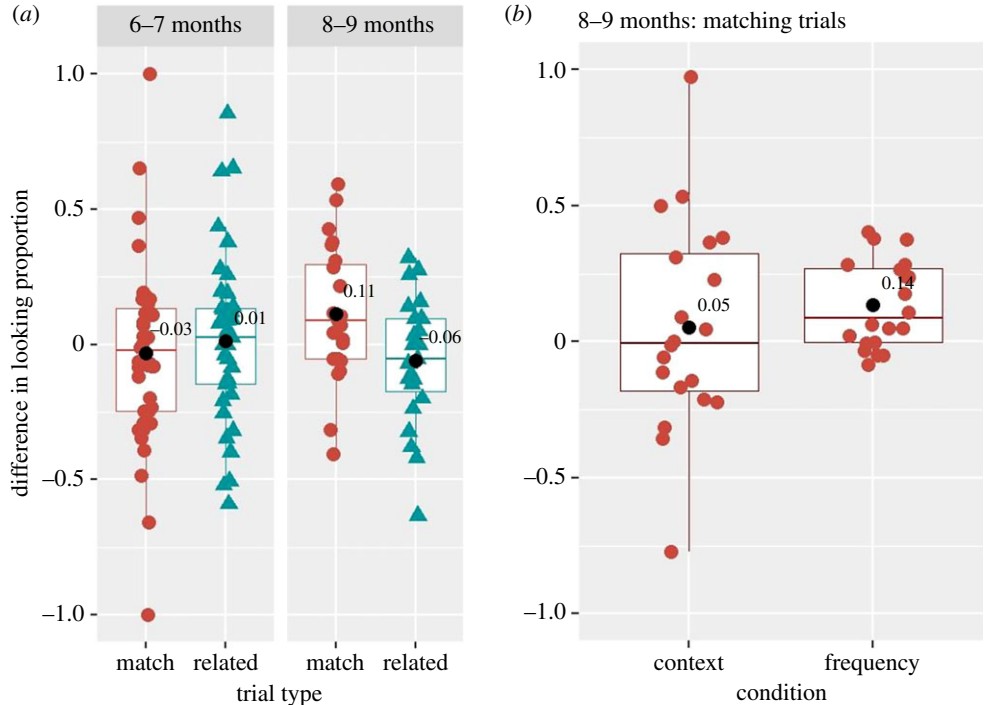

**Figure 8.** Boxplot showing the distribution of the difference in looking proportion for six- to seven- and eight- to nine-month-old infants on matching and related trials (*a*) and for eight- to nine-month-old infants on matching trials only in context and frequency conditions (*b*). Data were averaged across item pairs. Black circles indicate average values over all subjects in a given trial type and condition.

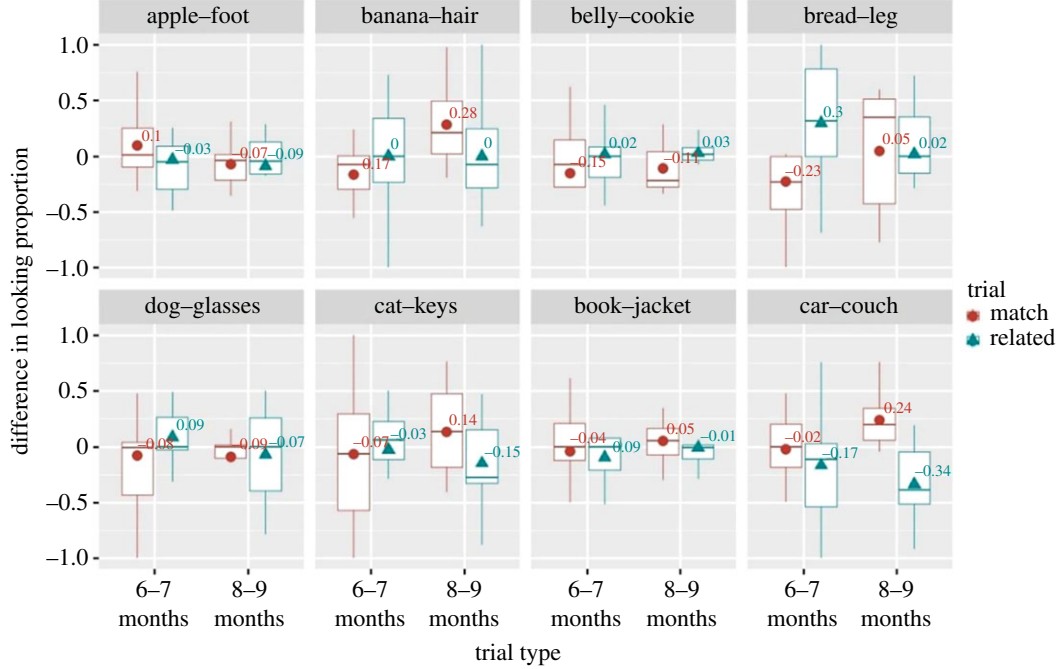

**Figure 9.** Boxplot showing, for each item pair, the distribution of the difference in looking proportion for six- to seven- and eight- to nine-month-old infants on matching and related trials.

frequencies of word occurrence in Norwegian child-directed speech [11,20,21]. The results revealed a positive relationship in both age groups, $r = 0.27$ ($t = 0.69$, d.f. $= 6$, $p = 0.25$, $BF_{10} = 1.05$) and $r = 0.67$ ($t = 2.21$, d.f. $= 6$, $p = 0.034$, $BF_{10} = 3.17$), although, only in the older group the correlation reached significance (figure 5). This result suggests that, in eight- to nine-month-old Norwegian infants, word pairs with

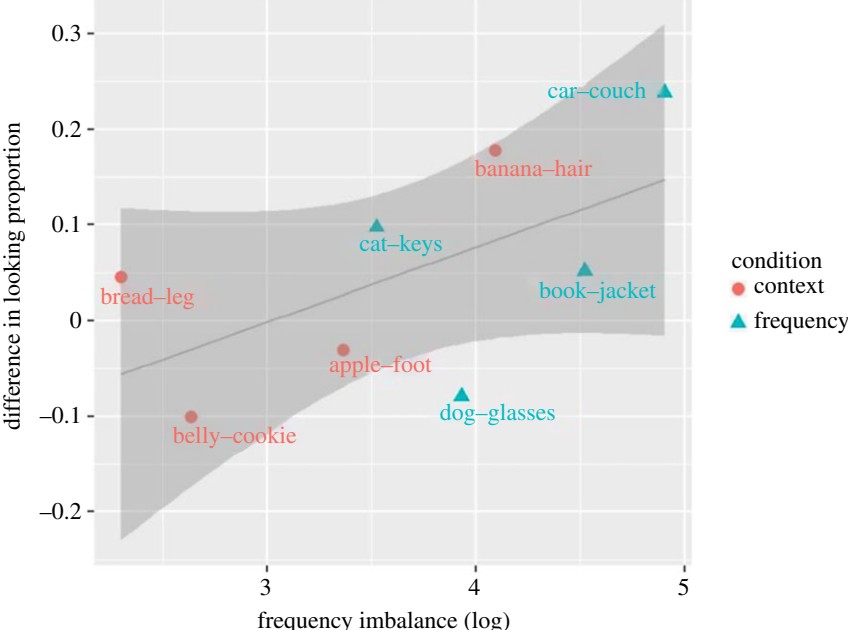

**Figure 10.** Relationship between the imbalance in frequencies between the two words in a pair and their effect size in word comprehension task in eight- to nine-month-old infants.

larger imbalance in frequencies led to larger effect sizes; the Bayes factor analysis indicated that the data presented moderate evidence for this relationship (figure 10).

Finally, similar to previous research [4], we explored, in addition, whether eight- to nine-month-old infants showed better word recognition on those words that parents reported as 'understood', as compared with the 'non-understood' words (as revealed by their answers in the CDI questionnaire). For that, in line with previous research [1,2], we computed a *baseline-corrected looking time* measure on each matching trial,[13] as a difference in the proportion of looks to the target as compared with the distractor between pre- and post-naming periods. We performed mixed-effect regression analysis on a log-transformed baseline-corrected looking time measure with the fixed factors of condition, comprehension (yes, no) and their interaction, and with by-subject and by-item intercepts. The results revealed a significant effect of condition ($F_{1,248} = 7.55$, $p = 0.0064$, between condition *Cohen d* = 0.75), with larger baseline-corrected looking times in the frequency ($n = 21$, $\mu = 0.07$, 95% CI [0.001, 0.13]), than in the context ($n = 22$, $\mu = -0.08$, 95% CI [$-0.17$, 0.024]) condition (figure 11a). Note that, analogously to our previous analyses with the dependent measure 'difference in looking proportion', the current (additional) analysis with a baseline-corrected looking measure revealed that eight- to nine-month-old infants showed word comprehension in the frequency condition only ($p = 0.041$, *Cohen d* = 0.47, $BF_{10} = 2.89$).[14] Furthermore, the results of the mixed-effect regression analysis revealed a non-significant effect of comprehension ($F_{1,248} = 0.099$, $p = 0.75$, between group *Cohen d* = 0.39), suggesting that there were no differences in looking times between the words reportedly 'understood' ($n = 8$, $\mu = 0.048$, 95% CI [$-0.138$, 0.235]) and 'non-understood' ($n = 22$, $\mu = -0.018$, 95% CI [$-0.085$, 0.048]), figure 11b. Finally, there was a significant condition by comprehension interaction ($F_{1,248} = 5.79$, $p = 0.016$), figure 11. Separate Wilcox one-sample two-tailed test showed that looking times were significantly different from chance only for the words reported as 'understood' in the frequency condition ($n = 22$,[15] $\mu = 0.19$, $V = 21$, $p = 0.036$, *Cohen's d* = 1.42, $BF_{10} = 7.07$, see §6.4 in analyses_results_codes.pdf for statistical detail in other conditions https://osf.io/gj8u9/?view_only= a4f61a751c4b478a814db3a54ac51ead). These results suggest that among words reportedly 'understood', infants showed word comprehension only in the frequency condition, i.e. when item pairs were imbalanced in their frequency.

---

[13]Note that, in line with previous research, for this analysis, we did not remove words that parents reported as not used with the child.

[14]The effect size in the context condition was *Cohen d* = 0.33, $BF_{10} = 1.97$ ($p = 0.13$, two-tailed one-sample test).

[15]Note that 1 out of 24 parents did not return the CDI questionnaire, and one parent returned it six months after the test was taken, so we excluded their infants from these analyses.

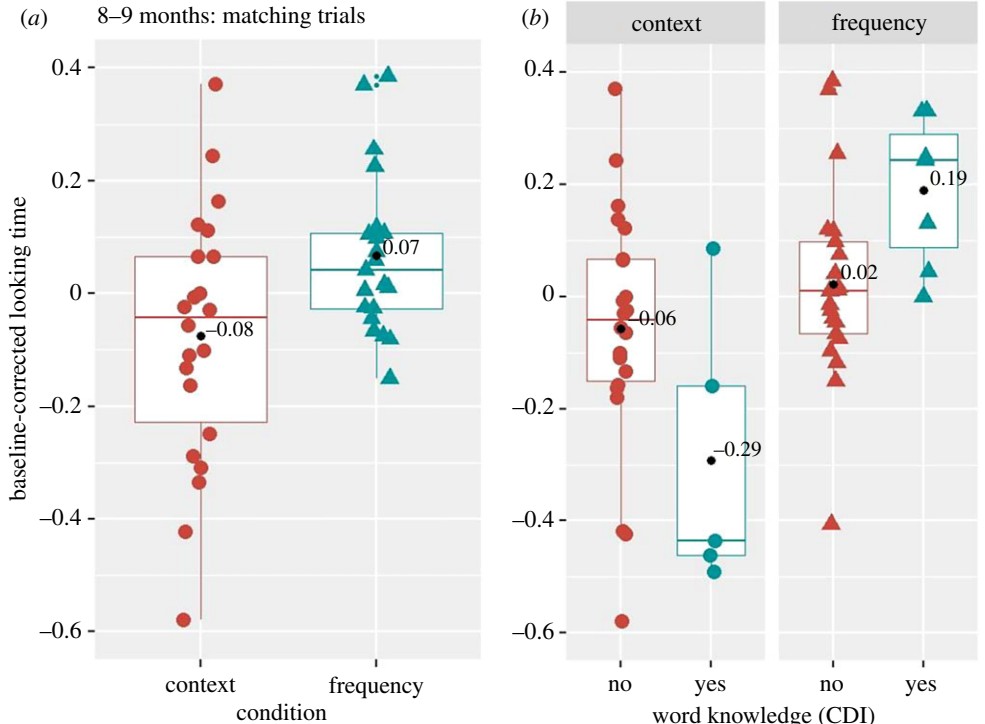

**Figure 11.** Baseline corrected looking time in context and frequency conditions in eight- to nine-month-old infants averaged over the words used in the task (*a*) and separated for words reported as 'understood' and 'not understood' by parents in the CDI (*b*). Coloured shapes (red and blue) represent individual child data averaged over item pairs.

## 4. Discussion

The current study had two aims. First, it aimed to replicate conceptually, in Norwegian, the results of BS12 study [2], in which six- to nine-month-old American-English-learning infants were shown to understand words sampled from distinct semantic categories (and contexts). Second, it aimed to examine two alternative explanations for the interpretation that infants understand the meaning of several words: namely, that infants may differentiate items capitalizing on differences in their frequencies of occurrence and/or in their contexts of use. Our planned (preregistered) analyses with 50 participants revealed that Norwegian six- to nine-month-old infants have not yet established a mapping between familiar objects and referent words: they showed no disambiguation between two objects sampled from different semantic categories. Note, however, that the group consisted dominantly from six- to seven-month-old infants. When, in the additional analyses, we included age as a covariate, the results indicated an age-related increase in looking times, suggesting that older infants might have started establishing word-object mappings. Our exploratory analyses with additional 20 participants confirmed this observation. They revealed that, (i) while eight- to nine-month-old Norwegian infants looked longer at the target when they heard the target label as compared with the distractor, six- to seven-month-old infants' looks at the target were not different from chance and (ii) eight- to nine-month-old Norwegian infants failed to rely exclusively on the frequency or context of word use to disambiguate between two items; yet, they only disambiguated item pairs that were imbalanced in their frequency of word use.

In contrast to previous studies with American-English-learning infants [1,2], our results revealed that six- to seven-month-old Norwegian infants did not show word comprehension for familiar words used in the study (e.g. body parts, food items, animals). The proportion of looks at the target was not different from chance, suggesting that six- to seven-month-old Norwegian infants have not yet established a mapping between familiar objects and their referent words. Similar results were observed in cluster permutation analyses that allow to take into account the time course of infant looking pattern. These differences with the results of BS12 study cannot be attributed to differences in the statistical analyses between the studies, given that we computed the same dependent measures and applied the same statistical analyses as in the BS12 study. Instead, these differences can potentially be accounted for by linguistic dissimilarities between the Norwegian and English languages. Norwegian is phonologically more complex than English. For example, while English uses formants (concentrations of acoustic

energy around a particular frequency) to distinguish vowel contrastive words ('bed', low first formant versus 'bad', high first formant), Norwegian, in addition, uses pitch (single pitch tone in *hender* 'hands' versus double in *hender* 'happens' pitch tone) and lengthening (short vowel *takk* 'thank you' versus long vowel *tak* 'roof') [32]. That is, in order to establish word-form representations, a Norwegian infant has to discover that three distinct acoustic cues (formants, pitch and length) allow to distinguish meaning in Norwegian (e.g. vowel minimal-word pairs), whereas an English-learning infant has to tune, *a priori*, to formant differences only. We speculate, that the high number of lexically relevant cues in Norwegian might require more time for Norwegian infants to tune to the language-specific phonology and to establish word-form representations. However, more data are required in order to be able to claim that there are developmental differences across the Norwegian and American-English-learning infants.

Older Norwegian infants, i.e. eight- to nine-month-olds, on the other hand, showed word comprehension for familiar words used in the study: their proportion of looks at the target was larger when they heard the target word label, as compared with when they heard the distractor. The effect size was moderate (*Cohen d* = 0.44), yet, positive confidence intervals and Bayes factor analysis suggest that our data present evidence for an emergence of word comprehension in Norwegian infants by eight- to nine months of age.[16] To the best of our knowledge, this study is the first showing word comprehension in eight- to nine-month-old infants learning a different language than English, suggesting that there may be a language-dependent onset for word comprehension. Yet, more data from other languages and with more complex phonologies (e.g. Danish) are required to confirm this hypothesis. For instance, an analysis of the CDI reports revealed that the vocabulary size in the Danish children was the lowest across 17 languages from the age 1 onwards [33], suggesting that the complexity of a language's sound system might influence early lexical development. Together with the fact that Danish is considered to be less intelligible than Norwegian (as judged by native Scandinavians, see [34]), we can only speculate that word comprehension would be delayed in infants learning a phonologically more complex language (Danish). Future research needs to shed light on this issue.

Exploratory analyses revealed that word comprehension observed in eight- to nine-month-old infants was driven by their larger proportion of looks at the target in the frequency condition, in particular; infants' proportion of looks in the context condition was not different from chance. Large effect size (*Cohen d* = 0.84), positive confidence intervals and, importantly, high Bayes factor ($BF_{10}$ = 56.32) in the frequency condition provide strong evidence in favour of this interpretation [35]. Recall that, in the frequency condition, words were imbalanced in terms of their frequency of occurrence in Norwegian child-directed speech. In the context condition, on the other hand, words had comparable frequencies of use. Crucially, in both conditions, within each picture-pair, referent words were sampled from different semantic categories and different contexts. Thus, these results suggest that eight- to nine-month-old infants rely on multiple available cues (e.g. semantic, contextual, frequency) to recognize words and that semantic cues alone do not trigger word comprehension; infants need additional cues, among which frequency appears to be a good candidate that they readily use to disambiguate between items. Similar results were obtained when we controlled for word knowledge as revealed by parental reports in the CDI [17]. Among words reportedly 'understood', infants showed word comprehension only for words used in the frequency condition, suggesting that infants needed additional cues (as, for example, frequency imbalance) to reveal word comprehension. Other research has shown that object-related factors, such as salience and typicality may also influence word comprehension in young infants, as 12-month-old infants do not recognize atypical objects, suggesting that their early word categories are conceptually limited to typical word-referent mappings [9,36]. Note that, in a difference to the results of a recent study [4], our study did not provide evidence that parents are good assessors of their infant's early word knowledge, as, overall, there was no difference in looking time between the words reportedly 'understood' and 'non-understood'.[17] However, given that only a few of the stimuli used in the current study (and sometimes none) were reported by parents as 'understood' by their child, these results should be taken with caution and confirmed with larger samples and more items. In sum, our results are in line with a recent study in six-month-old infants showing a relationship between infants' noun comprehension in the

---

[16]Note that the median vocabulary sizes of 8.5 and 15 words, reported in our study by parents of eight- and nine-month-old infants, respectively, are very similar to the median vocabulary sizes of 9 and 19 words previously reported in Norwegian eight- and nine-month-old infants (CDIs; retrieved from wordbank.stanford.edu, see [13,16,17]), suggesting that the vocabulary size revealed in our sample represents the vocabulary size of Norwegian eight- to nine-month-old infants (at least as far as the CDI reports are concerned).

[17]Although note that the average looking time for the 'understood' words was positive and higher than the average looking time for the 'non-understood' words (which was negative).

laboratory and the frequency of object-noun co-occurrence use at home [1], suggesting that frequency of word-object co-occurrence modulates word learning.

Finally, similar to the analyses performed for BS12 study, we examined the relationship between the imbalance in frequencies between the two words in a pair and their effect size in six- to seven- and eight- to nine-month-old infants. The results revealed a positive relationship—its magnitude was similar to our re-analysis of BS12 six- to seven-month-old infants—showing that, in eight- to nine-month-old Norwegian infants, word pairs with larger imbalance in frequencies had larger effect sizes. Bayes factor analysis indicates that our data provide moderate evidence in favour of this relationship [35]. Our data provided no support for an existence of a similar relationship in six- to seven-month-old Norwegian infants, though. Given that infants failed to show preference for an object when its corresponding label was substituted by a semantically unrelated word that was either similar in frequency or sampled from the same context, our results suggest that young infants use cumulative evidence from a number of converging cues to disambiguate between two items. Infants display a more robust recognition when more cues are congruent with one of the items, as is the case for matching labels (semantic, contextual and frequency congruence), than when they can rely only on a single cue, as is the case for related words (only context or frequency congruence); yet, an imbalance in frequencies between two items provides additional cue that triggers word recognition. Previous research has already shown that infants use semantic, conceptual and perceptual cues to disambiguate between two items and when these cues are congruent with both items, infants fail to disambiguate [1,8,10], suggesting that early semantic categories are underspecified and associative in nature [37].[18] The current study provides evidence that eight- to nine-month-old infants use frequency of word use as an additional cue to recognize words. A comparison with the item-pair effect sizes in BS12 study suggests that the onset of the frequency effect differs between languages and it coincides with the early onset of word recognition, given that our data provided no evidence for a relationship between the imbalance in frequencies and word-pair effect size in six- to seven-month-old Norwegian infants, who, overall, did not show word recognition. The absence of such a relationship in American-English eight- to nine-month-old infants (as revealed by our re-analysis of the BS12 study) potentially suggests the short longevity of the effect; yet, a proper study carefully controlling for the frequencies of word use is needed to shed further light on this issue.

In sum, our exploratory results have shown, first, that Norwegian infants show first evidence for word comprehension at eight to nine months of age—rather than from six to seven months of age for English-learning infants, suggesting that there are cross-linguistic differences in the onset of word comprehension. Yet more cross-language studies are needed to confirm this hypothesis. Second, our study revealed that infants at that age cannot rely exclusively on single extra-linguistic cues to disambiguate between two items, thus suggesting that they have established some/a mapping between a referent word and an item; however, this mapping is weak, as infants need additional cues (such as an imbalance in frequency of word use) to reveal word recognition. These results align with previous research showing that word frequency is one of the most important factors in predicting the emergence of its word comprehension [11,12] and that infants readily use frequencies of word-object co-occurrence to learn words [1].

Ethics. The study has been approved by the Norwegian Centre for Research Data (NSD) no. 56312.
Data accessibility. Data available from https://osf.io/gj8u9/?view_only=a4f61a751c4b478a814db3a54ac51ead.
Authors' contributions. N.K. participated in the design of the study, carried out the statistical analyses and drafted the manuscript. J.M. participated in the design of the study, coordinated the study and helped draft the manuscript.
Competing interests. We declare we have no competing interests.
Funding. Part of this work was supported by the HUP funding (Department of Psychology, University of Oslo).
Acknowledgements. We are very thankful to Anna Stigum Trøan for her help with participant recruitment and data collection. We are thankful to all infants and their parents who took part in the study.

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
