## [Reviewer comments · Royal Society Open Science]

Review History

RSOS-172200.R0 (Original submission)

Review form: Reviewer 1

Is the language acceptable?

Yes

Do you have any ethical concerns with this paper?

No

Have you any concerns about statistical analyses in this paper?

Yes

Recommendation?

Major revision

Comments to the Author(s)

I had the pleasure of reading the Stage 1 registered report on "Word knowledge in 6-9-month-old infants. Recognition without comprehension?" for Royal Society Open Science.

Overall, I agree with the authors that early word knowledge requires further investigation, especially in a new language and with added exploratory conditions such a study is a worthwhile undertaking. My subsequent comments all have in mind the improvement of the study - it's a privilege to be able to do this before data collection.

1. Framing

I do think the overall research question is very important and as the authors point out, there are still too few studies on the topic and (because of inherent difficulty with small lexica) stimuli are not always controlled for dimensions of interest.

Nonetheless, I am not sure I follow the reasoning for the two added conditions; in other words: the logic of the hypotheses was not clear to me.

Note that it is important to have a completed and polished introduction and hypothesis section as this section is "frozen" after acceptance according to the rules put forward by the journal.

I will try to reiterate how I understood the authors think the two conditions are useful and add my comments, if I have misunderstood, it should thus become clear now and all following criticisms should be considered with this in mind (and then the introduction might need to be expanded correspondingly, I think in any case it would be better to take the reader through the reasoning more carefully - apologies if the authors worked with a strict word limit).

The authors argue that in previous studies on word recognition (i.e. cross-modal matching of words with one of two possible referents) only showed a significant looking preference for the correct item because infants could associate a term with a general context of use or differences in frequency.

So in the lab, if infants hear "apple" and think of a general kitchen-food type of situation, they will look more to the apple (or in this experiment a related item like a cookie) versus the toy truck. Similarly for frequency, if they hear a word that they encountered a lot before they will look to the object more frequently present in their environment (the latter part is in my opinion completely missing from the paper, I infer it).

My first question is now, how this explains that 6-month-olds seem to do fine when dealing with hands and feet (ref 5).

I also wonder: Would it also be possible to make predictions about effect sizes based on the difference in frequency / context or both? Bergelson & Swingley provide very interesting item-pair analyses.

Further, I am not sure I agree with the presentation and interpretation of reference (1), especially in light of reference (15), which did look at semantic relatedness versus nonce words in the labels. Bergelson & Aslin in (15) do find that infants look to the related item (eg milk when juice is mentioned), but they do less so than when label and objects match. In fact, such an outcome is not covered by the present hypotheses. Either infants look both to the same object or a semantically related one, or they only look when object and label match. This binary prediction thus does in my view not work with the existing literature; I will discuss the proposed analyses in more detail below.

The home-lab links found in (1) also point towards a frequency effect, though. Observing words and their referents together becomes more likely with higher frequency. This could be discussed in more detail in the paper.

Finally, (1) provides a very useful discussion of the key results that this paper seems to be built on and offers an interesting alternative that the authors here do not bring up as far as I can see, namely competition effects - and a lot of evidence points towards semantic relatedness predicting word acquisition in typically developing children. The present paper is not able to disentangle the two options, if I see correctly. It doesn't have to, but it should acknowledge the competing possibilities that end up making the same coarse predictions, i.e. less looking when two semantically related objects are present.

Further comments on the Introduction:

In footnote 3, the authors mention mutual exclusivity, but as far as I know a corresponding ability has not been attested before late in infants' second year (see eg here:

https://figshare.com/articles/Disambiguation_Meta_Analysis/1348836), I am not sure how this point applies to infants before their first birthday - see also results in (15). If I understand the authors correctly, a similar (or precursor) process is what drives infants' looking to less frequent items upon hearing their label. I think this is an important assumption that actually should be tested - either by more closely inspecting the data from previous studies or in a dedicated analysis of the data to be collected.

The possibilities are that (1) infants look more to target when it is very frequent and equally to both items when the same pair is presented with the less frequent label or (2) infants look to the "correct" label independent of frequency as long as there is a frequency mismatch.

I think disentangling those possibilities would really push knowledge on infants' early lexical abilities much further.

I would not conclude from the discussion in paragraph 2 of the introduction that research is inconclusive, but that under different conditions infants do different things - which is to be expected and actually studied in this paper.

On page 2, the authors end the top paragraph with a rather general statement, it would have been useful to reiterate how to choose stimuli and why this is (not) surprising.

The authors discuss babiness in quite some detail, but I see no definition nor does this come back in the actual experiment.

On page 2, also carefully distinguish production and comprehension, both require different skills both in the infant and in the parent filling in a questionnaire.

Why are familiarity and novelty effect introduced as terms on page 2 if they do not come back?

Finally, do you expect any differences since you are (finally!) testing infants learning a language that is not North American English? Looking at <http://wordbank.stanford.edu> the overall trajectories look similar enough. It would be good to at least mention the language difference and whether this influences the predictions or not.

2. Statistical analysis

The authors propose to conduct three t-tests to assess what are essentially interaction effects (and I would much prefer to see more graded predictions). This analysis is not able to address the research questions, as null results are essentially uninterpretable in the way authors wish to do so (see below).

In addition, I was surprised to see two-tailed tests, while the predictions are clearly directional (looking more at target than distractor upon labelling). A one-tailed test would overall be more useful.

Finally, in accordance with the journal's guidelines for good statistical practice, include assumption checks before conducting any statistical analyses. For possible data transformations, see also:

Csibra, G., Hernik, M., Mascaro, O., Tatone, D., & Lengyel, M. (2016). Statistical treatment of looking-time data. *Developmental Psychology*, 52(4), 521-536.

In any case, to address the (now implicit) question whether infants' responses differ across conditions, a regression-type analysis is necessary. Simply comparing p-values as the authors plan to do now is not sufficient, they cannot even speak to a possible difference between conditions.

(But even this analysis does not do the rich eye tracking data justice - I would like to point the authors to this excellent R package: <http://www.eyetracking-r.com/>).

An updated manuscript should specify the according analyses and move away from p-values as sole basis for any conclusions.

3. Power / the meaning of null results

The authors conduct a power analysis as carefully as possible given the state of the literature (and - to my shared frustration - lacking reporting of all information needed to compute effect sizes). I think the sample size is suitable for the planned replication, especially given the additional constraints imposed by infant research (it would, for example, be highly unethical to run severely over-powered studies with a population that cannot consent themselves).

However, there are some issues that I must mention.

- The authors wish to run an extension that essentially requires looking at an interaction effect (see point 2 for more on that). Effect sizes of interactions can be expected to be smaller than main effects.

- Because first studies on a topic tend to over-estimate effect sizes some have argued that replications need 2.5 times as many participants.

As a consequence of both, the resulting effect size might just be much smaller, meaning that any null result is due to lack of power and not support for a "true" null (which cannot be concluded from null-hypothesis significance tests).

This does not mean that the authors should drastically increase their sample size, but procedures to be better able to interpret null results should be implemented to distinguish low-powered results from support for the null hypothesis: the authors should report Bayes Factors and effect sizes (as required by the APA since the 5th edition). Both are straightforward for t-tests and not impossible to obtain for ANOVAs: <https://sites.google.com/site/lakens2/effect-sizes> and <https://jasp-stats.org/>

Any conclusions should be based on a consideration of all of those. It would of course also be possible to completely move to Bayes Factors, I simply assume the authors might be more comfortable with null hypothesis significance tests.

4. Degree of detail needed to replicate the study

I would recommend a number of additions to ensure replicability:

- In Table 1,
- Provide a lab visit walk-through video. In the ManyBabies project for example such videos are required and they highlight things that are difficult to describe, like seating, light, how rooms look like and what the participant experience would be. Such videos can be made with a student or colleague as stand in for parent and baby, and then shared on platforms like Databrary.
- List all demographic data you are collecting (infant sex, age; parental education or similar)
- Detail whether (and if yes, how) you plan to share your protocols and materials
- Add a mock screen of what the infant sees and hears to better illustrate the conditions
- Provide a list of all picture-pairs and replacement words per conditions (I must admit I was only confused by Table 1).
- Add a schematic of the lab set up

5. Piloting

I strongly recommend piloting, especially when working with a new method. I see no specification of pilot participants, and thus there should be no pilots planned. It might be useful to reconsider this, if only to get the procedure straight and ensure that the target images are not dramatically different in terms of salience (this is possible through analyzing pre-naming periods).

6. "Quality checks" / item/trial/participant exclusion criteria

I have a number of concerns regarding the quality checks and exclusion criteria.

First, everything hinges on a successful replication of the original study (see above comments regarding power, this is not a foregone conclusion, and even with 80% power every 5th result is a null result).

The quality check does not ensure that the stimuli were chosen to ensure a successful replication (see above point about picture salience, in addition quality checks for the auditory stimuli (adult ratings) and chosen words (are they familiar to the child? The authors make a strong point about this in the introduction) seems advisable.

In other words, try to set yourselves up for success and maximal effect size, also to guarantee that the two manipulations yield interpretable results.

I also wonder whether 64 trials is realistic for this age group and would expect a lot of infants to not complete the study (see again above point on piloting). Even 50% of the trials (ie 32) which also fulfill the additional requirements (looks at both objects before naming, no looks for 1/3 of post-naming window, parent errors [which could be avoided with recordings, I am not aware of a strong argument for parent-produced stimuli, if I am not mistaken most cited studies rely on recordings]). Note that you cannot adjust these criteria post hoc.

Finally, the definition of fussiness is insufficient, it would be possible to use looking times to have strict exclusion criteria (eg failure to look at the screen for 5 [or what you deem reasonable] consecutive seconds instead of "shows no interest" etc or a decline of overall looking time to the controls by more than a certain percentage (60%?)).

7. Further comments on the method

- I take table 1 to contain the actual words, but then I do not see how items are matched for frequency in the corresponding condition. The differences in frequency in fact seem to be similar to the other conditions. This needs a lot more explanation or different stimuli.

- A white background (p5, l 35) is not to be recommended for eye tracking research (the pupil is too small for most eye trackers to obtain a reliable signal), grey would be a much better choice, as the authors themselves say on page 6 (l 13)

- Some pairs share onsets, how do you account for the to be expected timing difference in infant looks to target (if they know both words, the first phoneme usually disambiguates and lets infants initiate saccades towards or decide to remain fixated on target).

- Why no attention getter between trials? Will you ensure infants look to the screen before a trial starts? Right now it does not look this way, which might lead to even greater data loss.

Review form: Reviewer 2

Is the language acceptable?

Yes

Do you have any ethical concerns with this paper?

No

Have you any concerns about statistical analyses in this paper?

No

Recommendation?

Major revision

Comments to the Author(s)

The goals of this study are two-fold: 1. a replication of the finding of word comprehension in 6-9 month olds in the form of longer looking toward a visual display of a labeled object in preferential looking/eye-tracking experiments. 2. an examination of two alternative explanations for the infants' performance, namely a) the object is associated contextually rather than semantically with the auditory label (e.g. experiences in the kitchen) and/or b) the object is associated by frequency with the auditory label (i.e., high frequency words and high frequency labels).

The topic is an important and timely one. These findings of early word-meaning associations in young infants have generated much discussion. The overall theoretical question of what it means to have lexical knowledge of a word-meaning pairing has been a topic of discussion for a long time but is far from resolved – this study would add to that discussion. However, I have a number of concerns about the study as currently articulated. These concerns centre mainly around a) a clearer articulation of the distinction between semantic/conceptual/contextual relationships as they relate to word learning and b) concerns about the frequency analysis. I also have some questions of clarification around the method.

The authors propose to replicate the general finding of an association between the target words and the visual display. Two additional conditions will test the alternative explanations for the infants' behavior. In the context condition, infants will not hear either of the presented images labeled, but will instead hear a contextually related word to one of the images. In the frequency condition, the label will be frequency-matched to one of the two presented visual displays. In

principle, the proposed hypotheses and the manner of testing are logical, and have the potential to provide some important insight into how to interpret the findings of word comprehension in infants.

However, a number of questions and concerns emerged regarding the details of the method and stimuli:

1. The testing age range is wide (6-9 month olds). Although this is consistent with widely cited research from one particular research laboratory (the “original study”, see below), and therefore appropriate, the authors should take care in their wording in some places regarding whether this is examining lexical knowledge in 6-month-olds (which is the bottom of the testing range).
2. There was some lack of clarity in the description of the study, I think in part because the authors may have been expecting the reader to be familiar with the details of Bergelson & Swingley. For example, it took me a while to sort out (even with Table 1) how many trials of each type would be presented to each infant. I think I have it figured out, but there is still some confusing language – for example while there appear to be 16 trials repeated twice each (for a total of 32, as with B&S), in one place the authors refer to the “8 picture-pairs”. I can interpret this as meaning that “there are 8 picture-pairs for the replication” and an additional 8 for the two tests of alternative explanations (4 each), but I don’t see any matching of pairs between the context-relatedness condition and the associated comprehension condition. So will all pairs be presented to all infants? If not, how are they organized?
3. I don’t see any information for what audio target will be given in the context and frequency conditions – this seems crucial for evaluating any potential confounds in the stimuli, and also assessing subtleties regarding the extent to which “contextually-related” and “conceptually-related” are truly differentiable ideas.
4. Ideally, I would want to have access to at least some examples of the audio and videos, e.g. over OSF (I don’t see this necessarily as a requirement to evaluate the integrity of the study, but it seems like something that one might expect to see in this case, and might help to clarify some of my above questions).
5. Given that the authors are setting this up as a direct replication of Bergelson & Swingley rather than a more conceptual replication, it would be helpful to have some more explicit information about the ways in which the stimuli/design were the same or different. Note that while the authors state in a footnote that certain information is unavailable from that study, I am confident that Bergelson and Swingley would be responsive to requests for further information about their methods and results, and perhaps even access to their stimuli. (If an attempt was made and they were not, this is worth stating outright.)
6. The authors state that “the specificities of the eye-tracking machine and the apparatus will be provided at later stages of the pre-registration submission”. (p. 6, line 8) From the perspective of “Whether the authors provide a sufficiently clear and detailed description of the methods to prevent undisclosed flexibility in the experimental procedures or analysis pipeline”, these details are very important to tie down prior to data collection. It is not clear to me whether there will be an additional opportunity to submit them prior to IPA, but they should be committed to in advance. Again, it would be helpful to know if there are significant details that differ from Bergelson & Swingley in this aspect of the method, particularly in case there is a failure to replicate.
7. I am concerned about the problem identified by the authors in footnote 5. If the CHILDES frequencies are not indicative of their “true” frequencies in the child’s experience, and I think they are right that this is likely a problem, this becomes problematic both for testing Hypothesis 3 and for matching frequencies in the context-related condition. I don’t see a way around this (I didn’t parse the sentence “Therefore, frequency-matching of words within pairs of food- and bath-related objects is indicative”, so I’m not sure what is meant by this or whether it might help at all.)
8. It’s not clear to me what will be used as the “distractor” for the proportional (baseline-corrected) measure in the control trials.

9. The “cookie-belly” pairing seems problematic as separate categories, since one could imagine talking about eating a cookie and it ending up in the belly?
10. It would be helpful to have some articulation of how the authors went about frequency-matching and/or what was considered “high” vs. “low” frequency. In the Table, there is great diversity in frequency (as would be expected given the constraints of a natural language), and it’s not at all clear to me what is being considered matched/different and why. This seems critical to being able to address Hypothesis 3 (see below).

In addition, I had a number of wording concerns in the introduction. Although to a certain extent these are relatively minor wording issues, given that the goal of this study is to split some rather fine (if important) hairs regarding what it means for infants to have word-knowledge as a potential critique of prior findings, I think it is incumbent on the authors to take special care.

1. The authors need to better differentiate in some places between studies that identify word comprehension AT 6 MONTHS versus those finding evidence from 6-9 months, particularly given the historical alternative they are differentiating this from is “approximately 8 months of age” (p1, line 20), which is within the 6-9 months range.
2. It’s not clear what is meant by “Research on common words between 6 and 9 months, however, is less conclusive”. What is the comparison group here, since the same references are being cited as in the prior paragraph?
3. I feel like to a certain extent the introduction is setting up a straw man. Don’t get me wrong, I agree that it is very important to understand how fragile infants’ representations are and how and when they break down. But I don’t think anyone is arguing that infants have anything like adult-like understanding of these words. Certainly, if frequency effects are driving some of the prior results, that would put those prior findings in a very different light. But it’s not clear to me that finding a contextual effect is completely inconsistent with the field’s current interpretation of the existing literature. This comes down to needing a more nuanced articulation as noted above/below regarding what is meant by “contextual” and how it differs from other kinds of information that infants might use to form word category boundaries.
4. Relatedly, while the fact that our ability to detect the infants’ knowledge is fragile is presumably in large part due to the fragility of their knowledge, it is equally likely that some of the fragility is due to the difficulty in testing infant knowledge. While we need to be careful about over-interpreting infant knowledge, we also need to acknowledge the limits of our own measurements of that knowledge.
5. Again relatedly, if infants fail to differentiate between dog and cat, this may not mean that they don’t know the word “dog”, but rather that their category of dog is not fully differentiated (i.e. a conceptual/category issue rather than a lexical one). One might differentiate (though the authors don’t do so cleanly in their introduction if that is indeed their intention) between conceptually related categories of items like “dog” and “cat”, and contextually related items like “items encountered during mealtime”. But even in the latter case, it is possible to argue that this is part of the earliest stage of category formation – not a failure of the word learning system, but simply a consequence of an emerging conceptual repertoire of meanings to which to attach labels. (The labels are attached to the available meanings, but the meanings are still being refined.) I am not saying that the authors don’t have an important distinction to make here, but just that greater clarity in the fine-grained details about what they are actually arguing is needed.
6. The authors refer to reference 2 as “the original study”. There are two possible meanings, one that this is the first study to demonstrate word knowledge in young infants (the way I first interpreted it on page 2, and note that it is not the first such study), and two that it is the study being replicated (which I believe is the intended meaning). This should be made more clear.

With these in mind, I had some specific concerns about the extent to which the analyses can address the hypotheses:

1. Given that the infants under study live in Oslo, it is safe to say they would not be English-speaking as the infants in the original studies were? Given this, and other methodological

differences, this should probably not be considered a direct replication, but more of a conceptual one. I don't think this is a problem, but just to be clear in how any "failure to replicate" might be interpreted.

2. For Hypothesis 2, I'm not sure that testing against chance performance is right given what they are trying to demonstrate. In addition to the more conceptual concerns expressed already, it's not clear to me that infants looking longer toward contextually-related words is necessarily a "wrong" behavior. When presented with a two-alternative forced choice, looking at the most related item by some measure makes sense as a response. One approach would be to directly compare looking in the comprehension condition to looking in the contextually-related condition. One might hope to see greater looking in the comprehension condition if indeed infants have "established object-name associations at a semantic level" (page 4, line 28). However, it is questionable whether there would be sufficient power to pull out such a difference, so a null result would be less interpretable. If testing against chance is used, the conclusion would be that contextual relatedness influences the infants' looking time, which is therefore a viable alternative explanation for their performance, but I would argue one could not go so far as to say that "infants rely on contextual... cues... without having established object-name associations at a semantic level". Here is why a more nuanced discussion of the semantic-conceptual-contextual distinction is really needed.

3. For Hypothesis 3, I think the argument for testing against chance is more sound – it's harder to come up with an explanation for the infants' behavior in that case that is positive for building lexical knowledge, and not simply a confound. However, again a positive statistical test would demonstrate that the confound exists, but would not directly rule out the possibility of semantically-relevant word knowledge. More importantly, the concerns outline above regarding the frequency matching/mis-matching need to be addressed in order to evaluate the validity of this test.

Despite these concerns, I do see value in the enterprise and look forward to hearing the authors' response.

Other/Smaller points:

1. "Recent large-scale corpus analysis..." (p. 2 line14) – is there a reference for this?
2. "babiness" is not word, but I understand the intended meaning. Perhaps "babyish-ness" would work? Or something more technical
3. The sentence "Note, however..." (p. 2, line 25) is difficult to parse.

Review form: Reviewer 3 (Kerstin Meints)

Is the language acceptable?

Yes

Do you have any ethical concerns with this paper?

No

Have you any concerns about statistical analyses in this paper?

Yes

Recommendation?

Major revision

Comments to the Author(s)

- The significance of the research question(s)

Good research idea- to differentiate between frequency and context or context familiarity is very useful indeed.

□ The logic, rationale, and plausibility of the proposed hypotheses

Overall the logic and rationale are good and very worthwhile, and the hypotheses are well worth exploring and would bring new knowledge to this area of research.

□ The soundness and feasibility of the methodology and analysis pipeline (including statistical power analysis where applicable)

Exclusion criteria are all appropriate.

Sample size is appropriate for the suggested power and effect size (GPower3 calculation comes to same result).

The sentence-framing is appropriate, too.

Items that I would like to hear more about are as follows:

- Attention getters: Attention-getters are usually displayed in IPI in the centre of the screen – also to avoid side-bias – why would they be displayed at the side? Why is separate analysis interesting? Do the authors worry that the children will drop out due to the repetition of the stimuli? The attention-getter stimuli are not described – if they are more interesting than the other stimuli, then they would obviously attract more attention, ie one could argue if they are made to fulfil their purpose, then they have to be more interesting – but then they cannot be compared to the other stimuli.
- Repetition: Why will stimuli be repeated? The design could also be tested between-subjects.
- Choice of stimuli: It is appreciated how very hard it is to find appropriate early word stimuli for this study, however, when it comes to the selection of stimuli, the authors need to make sure that they are not introducing accidental confounders. For example, comparing the comprehension condition (food-related and bath-related item) and the context-related conditions and frequency-related conditions with other comprehension raises some questions. Food-related and bath-related items under Comprehension condition are really food items and body parts. In the context-related condition, food-related and bath-related items are 1 food (drink “milk”) and 2 “kitchen cupboard items” and a piece of furniture (table). I wonder whether this could be problematic. While the 3 items are context-related, albeit from 2 different semantic/conceptual categories, milk is another food item. It might be useful to exchange the food item (milk) with “bowl” or “chair” which are early words, too, so at least the items are all non-foods.

The authors write: Crucially, words within each picture-pair and their frequency-matched substitute words will refer to objects from different categories, consequently minimizing any confounding bias at a semantic and/or contextual levels.

Do the authors mean different categories to food and bath-time? Water in the frequency-match is related to the bath-routine.

It will be vital to make sure the stimuli really only vary on the dimensions chosen – context and frequency. Introducing more variables – same category / different category) may make findings hard to interpret and has the potential to confound the study as the authors agree.

The authors could use the Norwegian CDI data to help with stimulus choice for the age range.

- Frequency: It is unclear how the items in the frequency-match condition are frequency matched (range from 2-255) and with which items – within the pair or between conditions?
- Sound stimuli: Pre-recorded native Norwegian female – then presented to parents over headphones and then repeat to child – why introduce this variation? What if fathers attend versus mothers? In addition, it will be impossible to match target onset with word onset between children. Why not simply use all pre-recorded stimuli and avoid this variation? Baseline and post-word onset times can then also be made identical – this helps with ease of analysis, too.
- Analysis – please explain more clearly what data is intended for comparison.

□ Whether the clarity and degree of methodological detail would be sufficient to replicate exactly the proposed experimental procedures and analysis pipeline

• Procedure & Stimuli:

- Detail and clarity is not enough yet to replicate the procedure or stimuli, eg the onset of the auditory input from the parents may vary from each parent to the next – how would this be replicated?

- It is unclear whether Norwegian CDIs are collected from the children to be tested to see if their individual word knowledge fits with the stimuli selected – this would be vital as children look at stimuli they have a name for over stimuli they cannot name yet.

- Visual stimuli are not described – as especially children’s early word-object-mappings depend on how typical these objects are, visual stimuli should be assessed for typicality in Norway and described for replication (size, background, colour or B&W, etc.). Ideally, and wherever possible, animate stimuli should not be shown with inanimate stimuli as animates tend to be more interesting to children than inanimates.

- Attention-getters are not described.

- The authors say “The pictures will remain on the screen for 3500 ms (after the target-word onset) – it is unclear how this will be controlled

□ Whether the authors provide a sufficiently clear and detailed description of the methods to prevent undisclosed flexibility in the experimental procedures or analysis pipeline

Not yet enough information given and also, the analysis needs to be clarified. Which data is compared to which other data in the t-tests? Also unclear why this is not tested between-subjects to avoid repetition (and drop-out)?

Whether the authors have considered sufficient outcome-neutral conditions (e.g. positive controls) for ensuring that the results obtained are able to test the stated hypotheses

Yes, but please see comments above that need to be addressed.

Other:

Introduction:

In the introduction, it may be more useful not to speak of concrete versus abstract words – but use the appropriate linguistic terms instead (eg verbs, etc.). “Abstract word” can be easily confused with abstract noun (love, belief, etc.).

Babiness is not a good word. Saliency seems to be meant, or infant-relatedness?

Spoon over nose: “‘nose’, which is heard in less spatially and linguistically distinct contexts.” This is unclear - children may only hear their nose mentioned in highly-specific contexts, eg when they hearing and playing baby rhymes when brought to bed, so this needs careful consideration.

Decision letter (RSOS-172200.R0)

16-Feb-2018

Dear Dr Kartushina,

The Editors assigned to your Stage 1 Registered Report (“Word knowledge in 6-9-month-old infants. Recognition without comprehension?”) have now received comments from reviewers. We would like you to revise your paper in accordance with the referee and editors suggestions which can be found below (not including confidential reports to the Editor). Please note this decision does not guarantee eventual acceptance.

Please submit a copy of your revised paper within three weeks (i.e. by the 10-Mar-2018). If we do not hear from you within this time then it will be assumed that the paper has been withdrawn. In

exceptional circumstances, extensions may be possible if agreed with the Editorial Office in advance. We do not allow multiple rounds of revision so we urge you to make every effort to fully address all of the comments at this stage. If deemed necessary by the Editors, your manuscript will be sent back to one or more of the original reviewers for assessment. If the original reviewers are not available we may invite new reviewers.

Please note that Royal Society Open Science will introduce article processing charges for all new submissions received from 1 January 2018. Registered Reports submitted and accepted after this date will ONLY be subject to a charge if they subsequently progress to and are accepted as Stage 2 Registered Reports. If your manuscript is submitted and accepted for publication after 1 January 2018 (i.e. as a full Stage 2 Registered Report), you will be asked to pay the article processing charge, unless you request a waiver and this is approved by Royal Society Publishing. You can find out more about the charges at <http://rsos.royalsocietypublishing.org/page/charges>. Should you have any queries, please contact openscience@royalsociety.org.

on behalf of Professor Chris Chambers (Registered Reports Editor, Royal Society Open Science)
openscience@royalsociety.org

Associate Editor Comments to Author (Professor Chris Chambers):

The manuscript has now been assessed by three expert reviewers who provide a wide range of detailed and very constructive recommendations. All reviewers agree that the research question being asked is important, satisfying the first criterion of Registered Reports. However, all reviewers also raise significant concerns about all other sections of the proposal, from matters concerning the theoretical framing (including the criticism that the introduction may be setting up a straw man proposition), validity of the experimental procedures and proposed analyses (including statistical power and sample planning), overall detail of the proposed methods (which is insufficient to enable replication in many places), and potential interpretation of outcomes. The depth and breadth of these concerns would be sufficient to trigger rejection of a standard paper,

but one of the main advantages of the Registered Reports process is that it enables authors to amend and optimise a design before such concerns become fatal flaws. Therefore, please attend thoroughly to all comments in any revised submission.

Comments to Author:

Reviewer: 1

Comments to the Author(s)

I had the pleasure of reading the Stage 1 registered report on "Word knowledge in 6-9-month-old infants. Recognition without comprehension?" for Royal Society Open Science.

Overall, I agree with the authors that early word knowledge requires further investigation, especially in a new language and with added exploratory conditions such a study is a worthwhile undertaking. My subsequent comments all have in mind the improvement of the study - it's a privilege to be able to do this before data collection.

1. Framing

I do think the overall research question is very important and as the authors point out, there are still too few studies on the topic and (because of inherent difficulty with small lexica) stimuli are not always controlled for dimensions of interest.

Nonetheless, I am not sure I follow the reasoning for the two added conditions; in other words: the logic of the hypotheses was not clear to me.

Note that it is important to have a completed and polished introduction and hypothesis section as this section is "frozen" after acceptance according to the rules put forward by the journal.

I will try to reiterate how I understood the authors think the two conditions are useful and add my comments, if I have misunderstood, it should thus become clear now and all following criticisms should be considered with this in mind (and then the introduction might need to be expanded correspondingly, I think in any case it would be better to take the reader through the reasoning more carefully - apologies if the authors worked with a strict word limit).

The authors argue that in previous studies on word recognition (i.e. cross-modal matching of words with one of two possible referents) only showed a significant looking preference for the correct item because infants could associate a term with a general context of use or differences in frequency.

So in the lab, if infants hear "apple" and think of a general kitchen-food type of situation, they will look more to the apple (or in this experiment a related item like a cookie) versus the toy truck. Similarly for frequency, if they hear a word that they encountered a lot before they will look to the object more frequently present in their environment (the latter part is in my opinion completely missing from the paper, I infer it).

My first question is now, how this explains that 6-month-olds seem to do fine when dealing with hands and feet (ref 5).

I also wonder: Would it also be possible to make predictions about effect sizes based on the difference in frequency / context or both? Bergelson & Swingley provide very interesting item-pair analyses.

Further, I am not sure I agree with the presentation and interpretation of reference (1), especially in light of reference (15), which did look at semantic relatedness versus nonce words in the labels. Bergelson & Aslin in (15) do find that infants look to the related item (eg milk when juice is mentioned), but they do less so than when label and objects match. In fact, such an outcome is not

covered by the present hypotheses. Either infants look both to the same object or a semantically related one, or they only look when object and label match. This binary prediction thus does in my view not work with the existing literature; I will discuss the proposed analyses in more detail below.

The home-lab links found in (1) also point towards a frequency effect, though. Observing words and their referents together becomes more likely with higher frequency. This could be discussed in more detail in the paper.

Finally, (1) provides a very useful discussion of the key results that this paper seems to be built on and offers an interesting alternative that the authors here do not bring up as far as I can see, namely competition effects - and a lot of evidence points towards semantic relatedness predicting word acquisition in typically developing children. The present paper is not able to disentangle the two options, if I see correctly. It doesn't have to, but it should acknowledge the competing possibilities that end up making the same coarse predictions, i.e. less looking when two semantically related objects are present.

Further comments on the Introduction:

In footnote 3, the authors mention mutual exclusivity, but as far as I know a corresponding ability has not been attested before late in infants' second year (see eg here:

https://figshare.com/articles/Disambiguation_Meta_Analysis/1348836), I am not sure how this point applies to infants before their first birthday - see also results in (15). If I understand the authors correctly, a similar (or precursor) process is what drives infants' looking to less frequent items upon hearing their label. I think this is an important assumption that actually should be tested - either by more closely inspecting the data from previous studies or in a dedicated analysis of the data to be collected.

The possibilities are that (1) infants look more to target when it is very frequent and equally to both items when the same pair is presented with the less frequent label or (2) infants look to the "correct" label independent of frequency as long as there is a frequency mismatch.

I think disentangling those possibilities would really push knowledge on infants' early lexical abilities much further.

I would not conclude from the discussion in paragraph 2 of the introduction that research is inconclusive, but that under different conditions infants do different things - which is to be expected and actually studied in this paper.

On page 2, the authors end the top paragraph with a rather general statement, it would have been useful to reiterate how to choose stimuli and why this is (not) surprising.

The authors discuss babiness in quite some detail, but I see no definition nor does this come back in the actual experiment.

On page 2, also carefully distinguish production and comprehension, both require different skills both in the infant and in the parent filling in a questionnaire.

Why are familiarity and novelty effect introduced as terms on page 2 if they do not come back?

Finally, do you expect any differences since you are (finally!) testing infants learning a language that is not North American English? Looking at <http://wordbank.stanford.edu> the overall trajectories look similar enough. It would be good to at least mention the language difference and whether this influences the predictions or not.

2. Statistical analysis

The authors propose to conduct three t-tests to assess what are essentially interaction effects (and I would much prefer to see more graded predictions). This analysis is not able to address the research questions, as null results are essentially uninterpretable in the way authors wish to do so (see below).

In addition, I was surprised to see two-tailed tests, while the predictions are clearly directional (looking more at target than distractor upon labelling). A one-tailed test would overall be more useful.

Finally, in accordance with the journal's guidelines for good statistical practice, include assumption checks before conducting any statistical analyses. For possible data transformations, see also:

Csibra, G., Hernik, M., Mascaro, O., Tatone, D., & Lengyel, M. (2016). Statistical treatment of looking-time data. *Developmental Psychology*, 52(4), 521-536.

In any case, to address the (now implicit) question whether infants' responses differ across conditions, a regression-type analysis is necessary. Simply comparing p-values as the authors plan to do now is not sufficient, they cannot even speak to a possible difference between conditions.

(But even this analysis does not do the rich eye tracking data justice - I would like to point the authors to this excellent R package: <http://www.eyetracking-r.com/>).

An updated manuscript should specify the according analyses and move away from p-values as sole basis for any conclusions.

3. Power / the meaning of null results

The authors conduct a power analysis as carefully as possible given the state of the literature (and - to my shared frustration - lacking reporting of all information needed to compute effect sizes). I think the sample size is suitable for the planned replication, especially given the additional constraints imposed by infant research (it would, for example, be highly unethical to run severely over-powered studies with a population that cannot consent themselves).

However, there are some issues that I must mention.

- The authors wish to run an extension that essentially requires looking at an interaction effect (see point 2 for more on that). Effect sizes of interactions can be expected to be smaller than main effects.

- Because first studies on a topic tend to over-estimate effect sizes some have argued that replications need 2.5 times as many participants.

As a consequence of both, the resulting effect size might just be much smaller, meaning that any null result is due to lack of power and not support for a "true" null (which cannot be concluded from null-hypothesis significance tests).

This does not mean that the authors should drastically increase their sample size, but procedures to be better able to interpret null results should be implemented to distinguish low-powered results from support for the null hypothesis: the authors should report Bayes Factors and effect sizes (as required by the APA since the 5th edition). Both are straightforward for t-tests and not impossible to obtain for ANOVAs: <https://sites.google.com/site/lakens2/effect-sizes> and <https://jasp-stats.org/>

Any conclusions should be based on a consideration of all of those. It would of course also be possible to completely move to Bayes Factors, I simply assume the authors might be more comfortable with null hypothesis significance tests.

4. Degree of detail needed to replicate the study

I would recommend a number of additions to ensure replicability:

- In Table 1,
- Provide a lab visit walk-through video. In the ManyBabies project for example such videos are required and they highlight things that are difficult to describe, like seating, light, how rooms look like and what the participant experience would be. Such videos can be made with a student or colleague as stand in for parent and baby, and then shared on platforms like Databrary.
- List all demographic data you are collecting (infant sex, age; parental education or similar)
- Detail whether (and if yes, how) you plan to share your protocols and materials
- Add a mock screen of what the infant sees and hears to better illustrate the conditions
- Provide a list of all picture-pairs and replacement words per conditions (I must admit I was only confused by Table 1).
- Add a schematic of the lab set up

5. Piloting

I strongly recommend piloting, especially when working with a new method. I see no specification of pilot participants, and thus there should be no pilots planned. It might be useful to reconsider this, if only to get the procedure straight and ensure that the target images are not dramatically different in terms of salience (this is possible through analyzing pre-naming periods).

6. "Quality checks" / item/trial/participant exclusion criteria

I have a number of concerns regarding the quality checks and exclusion criteria.

First, everything hinges on a successful replication of the original study (see above comments regarding power, this is not a foregone conclusion, and even with 80% power every 5th result is a null result).

The quality check does not ensure that the stimuli were chosen to ensure a successful replication (see above point about picture salience, in addition quality checks for the auditory stimuli (adult ratings) and chosen words (are they familiar to the child? The authors make a strong point about this in the introduction) seems advisable.

In other words, try to set yourselves up for success and maximal effect size, also to guarantee that the two manipulations yield interpretable results.

I also wonder whether 64 trials is realistic for this age group and would expect a lot of infants to not complete the study (see again above point on piloting). Even 50% of the trials (ie 32) which also fulfill the additional requirements (looks at both objects before naming, no looks for 1/3 of post-naming window, parent errors [which could be avoided with recordings, I am not aware of a strong argument for parent-produced stimuli, if I am not mistaken most cited studies rely on recordings]). Note that you cannot adjust these criteria post hoc.

Finally, the definition of fussiness is insufficient, it would be possible to use looking times to have strict exclusion criteria (eg failure to look at the screen for 5 [or what you deem reasonable])

consecutive seconds instead of “shows no interest” etc or a decline of overall looking time to the controls by more than a certain percentage (60%?).

7. Further comments on the method

- I take table 1 to contain the actual words, but then I do not see how items are matched for frequency in the corresponding condition. The differences in frequency in fact seem to be similar to the other conditions. This needs a lot more explanation or different stimuli.

- A white background (p5, l 35) is not to be recommended for eye tracking research (the pupil is too small for most eye trackers to obtain a reliable signal), grey would be a much better choice, as the authors themselves say on page 6 (l 13)

- Some pairs share onsets, how do you account for the to be expected timing difference in infant looks to target (if they know both words, the first phoneme usually disambiguates and lets infants initiate saccades towards or decide to remain fixated on target).

- Why no attention getter between trials? Will you ensure infants look to the screen before a trial starts? Right now it does not look this way, which might lead to even greater data loss.

Reviewer: 2

Comments to the Author(s)

The goals of this study are two-fold: 1. a replication of the finding of word comprehension in 6-9 month olds in the form of longer looking toward a visual display of a labeled object in preferential looking/eye-tracking experiments. 2. an examination of two alternative explanations for the infants' performance, namely a) the object is associated contextually rather than semantically with the auditory label (e.g. experiences in the kitchen) and/or b) the object is associated by frequency with the auditory label (i.e., high frequency words and high frequency labels).

The topic is an important and timely one. These findings of early word-meaning associations in young infants have generated much discussion. The overall theoretical question of what it means to have lexical knowledge of a word-meaning pairing has been a topic of discussion for a long time but is far from resolved – this study would add to that discussion. However, I have a number of concerns about the study as currently articulated. These concerns centre mainly around a) a clearer articulation of the distinction between semantic/conceptual/contextual relationships as they relate to word learning and b) concerns about the frequency analysis. I also have some questions of clarification around the method.

The authors propose to replicate the general finding of an association between the target words and the visual display. Two additional conditions will test the alternative explanations for the infants' behavior. In the context condition, infants will not hear either of the presented images labeled, but will instead hear a contextually related word to one of the images. In the frequency condition, the label will be frequency-matched to one of the two presented visual displays. In principle, the proposed hypotheses and the manner of testing are logical, and have the potential to provide some important insight into how to interpret the findings of word comprehension in infants.

However, a number of questions and concerns emerged regarding the details of the method and stimuli:

1. The testing age range is wide (6-9 month olds). Although this is consistent with widely cited research from one particular research laboratory (the “original study”, see below), and therefore

appropriate, the authors should take care in their wording in some places regarding whether this is examining lexical knowledge in 6-month-olds (which is the bottom of the testing range).

2. There was some lack of clarity in the description of the study, I think in part because the authors may have been expecting the reader to be familiar with the details of Bergelson & Swingley. For example, it took me a while to sort out (even with Table 1) how many trials of each type would be presented to each infant. I think I have it figured out, but there is still some confusing language – for example while there appear to be 16 trials repeated twice each (for a total of 32, as with B&S), in one place the authors refer to the “8 picture-pairs”. I can interpret this as meaning that “there are 8 picture-pairs for the replication” and an additional 8 for the two tests of alternative explanations (4 each), but I don’t see any matching of pairs between the context-relatedness condition and the associated comprehension condition. So will all pairs be presented to all infants? If not, how are they organized?

3. I don’t see any information for what audio target will be given in the context and frequency conditions – this seems crucial for evaluating any potential confounds in the stimuli, and also assessing subtleties regarding the extent to which “contextually-related” and “conceptually-related” are truly differentiable ideas.

4. Ideally, I would want to have access to at least some examples of the audio and videos, e.g. over OSF (I don’t see this necessarily as a requirement to evaluate the integrity of the study, but it seems like something that one might expect to see in this case, and might help to clarify some of my above questions).

5. Given that the authors are setting this up as a direct replication of Bergelson & Swingley rather than a more conceptual replication, it would be helpful to have some more explicit information about the ways in which the stimuli/design were the same or different. Note that while the authors state in a footnote that certain information is unavailable from that study, I am confident that Bergelson and Swingley would be responsive to requests for further information about their methods and results, and perhaps even access to their stimuli. (If an attempt was made and they were not, this is worth stating outright.)

6. The authors state that “the specificities of the eye-tracking machine and the apparatus will be provided at later stages of the pre-registration submission”. (p. 6, line 8) From the perspective of “Whether the authors provide a sufficiently clear and detailed description of the methods to prevent undisclosed flexibility in the experimental procedures or analysis pipeline”, these details are very important to tie down prior to data collection. It is not clear to me whether there will be an additional opportunity to submit them prior to IPA, but they should be committed to in advance. Again, it would be helpful to know if there are significant details that differ from Bergelson & Swingley in this aspect of the method, particularly in case there is a failure to replicate.

7. I am concerned about the problem identified by the authors in footnote 5. If the CHILDES frequencies are not indicative of their “true” frequencies in the child’s experience, and I think they are right that this is likely a problem, this becomes problematic both for testing Hypothesis 3 and for matching frequencies in the context-related condition. I don’t see a way around this (I didn’t parse the sentence “Therefore, frequency-matching of words within pairs of food- and bath-related objects is indicative”, so I’m not sure what is meant by this or whether it might help at all.)

8. It’s not clear to me what will be used as the “distractor” for the proportional (baseline-corrected) measure in the control trials.

9. The “cookie-belly” pairing seems problematic as separate categories, since one could imagine talking about eating a cookie and it ending up in the belly?

10. It would be helpful to have some articulation of how the authors went about frequency-matching and/or what was considered “high” vs. “low” frequency. In the Table, there is great diversity in frequency (as would be expected given the constraints of a natural language), and it’s not at all clear to me what is being considered matched/different and why. This seems critical to being able to address Hypothesis 3 (see below).

In addition, I had a number of wording concerns in the introduction. Although to a certain extent these are relatively minor wording issues, given that the goal of this study is to split some rather fine (if important) hairs regarding what it means for infants to have word-knowledge as a potential critique of prior findings, I think it is incumbent on the authors to take special care.

1. The authors need to better differentiate in some places between studies that identify word comprehension AT 6 MONTHS versus those finding evidence from 6-9 months, particularly given the historical alternative they are differentiating this from is “approximately 8 months of age” (p1, line 20), which is within the 6-9 months range.
2. It’s not clear what is meant by “Research on common words between 6 and 9 months, however, is less conclusive”. What is the comparison group here, since the same references are being cited as in the prior paragraph?
3. I feel like to a certain extent the introduction is setting up a straw man. Don’t get me wrong, I agree that it is very important to understand how fragile infants’ representations are and how and when they break down. But I don’t think anyone is arguing that infants have anything like adult-like understanding of these words. Certainly, if frequency effects are driving some of the prior results, that would put those prior findings in a very different light. But it’s not clear to me that finding a contextual effect is completely inconsistent with the field’s current interpretation of the existing literature. This comes down to needing a more nuanced articulation as noted above/below regarding what is meant by “contextual” and how it differs from other kinds of information that infants might use to form word category boundaries.
4. Relatedly, while the fact that our ability to detect the infants’ knowledge is fragile is presumably in large part due to the fragility of their knowledge, it is equally likely that some of the fragility is due to the difficulty in testing infant knowledge. While we need to be careful about over-interpreting infant knowledge, we also need to acknowledge the limits of our own measurements of that knowledge.
5. Again relatedly, if infants fail to differentiate between dog and cat, this may not mean that they don’t know the word “dog”, but rather that their category of dog is not fully differentiated (i.e. a conceptual/category issue rather than a lexical one). One might differentiate (though the authors don’t do so cleanly in their introduction if that is indeed their intention) between conceptually related categories of items like “dog” and “cat”, and contextually related items like “items encountered during mealtime”. But even in the latter case, it is possible to argue that this is part of the earliest stage of category formation – not a failure of the word learning system, but simply a consequence of an emerging conceptual repertoire of meanings to which to attach labels. (The labels are attached to the available meanings, but the meanings are still being refined.) I am not saying that the authors don’t have an important distinction to make here, but just that greater clarity in the fine-grained details about what they are actually arguing is needed.
6. The authors refer to reference 2 as “the original study”. There are two possible meanings, one that this is the first study to demonstrate word knowledge in young infants (the way I first interpreted it on page 2, and note that it is not the first such study), and two that it is the study being replicated (which I believe is the intended meaning). This should be made more clear.

With these in mind, I had some specific concerns about the extent to which the analyses can address the hypotheses:

1. Given that the infants under study live in Oslo, it is safe to say they would not be English-speaking as the infants in the original studies were? Given this, and other methodological differences, this should probably not be considered a direct replication, but more of a conceptual one. I don’t think this is a problem, but just to be clear in how any “failure to replicate” might be interpreted.
2. For Hypothesis 2, I’m not sure that testing against chance performance is right given what they are trying to demonstrate. In addition to the more conceptual concerns expressed already, it’s not clear to me that infants looking longer toward contextually-related words is necessarily a “wrong” behavior. When presented with a two-alternative forced choice, looking at the most related item by some measure makes sense as a response. One approach would be to directly

compare looking in the comprehension condition to looking in the contextually-related condition. One might hope to see greater looking in the comprehension condition if indeed infants have “established object-name associations at a semantic level” (page 4, line 28). However, it is questionable whether there would be sufficient power to pull out such a difference, so a null result would be less interpretable. If testing against chance is used, the conclusion would be that contextual relatedness influences the infants’ looking time, which is therefore a viable alternative explanation for their performance, but I would argue one could not go so far as to say that “infants rely on contextual... cues... without having established object-name associations at a semantic level”. Here is why a more nuanced discussion of the semantic-conceptual-contextual distinction is really needed.

3. For Hypothesis 3, I think the argument for testing against chance is more sound – it’s harder to come up with an explanation for the infants’ behavior in that case that is positive for building lexical knowledge, and not simply a confound. However, again a positive statistical test would demonstrate that the confound exists, but would not directly rule out the possibility of semantically-relevant word knowledge. More importantly, the concerns outline above regarding the frequency matching/mis-matching need to be addressed in order to evaluate the validity of this test.

Despite these concerns, I do see value in the enterprise and look forward to hearing the authors’ response.

Other/Smaller points:

1. “Recent large-scale corpus analysis...” (p. 2 line14) – is there a reference for this?
2. “babiness” is not word, but I understand the intended meaning. Perhaps “babyish-ness” would work? Or something more technical
3. The sentence “Note, however...” (p. 2, line 25) is difficult to parse.

Reviewer: 3

Comments to the Author(s)

The significance of the research question(s)

Good research idea- to differentiate between frequency and context or context familiarity is very useful indeed.

The logic, rationale, and plausibility of the proposed hypotheses

Overall the logic and rationale are good and very worthwhile, and the hypotheses are well worth exploring and would bring new knowledge to this area of research.

The soundness and feasibility of the methodology and analysis pipeline (including statistical power analysis where applicable)

Exclusion criteria are all appropriate.

Sample size is appropriate for the suggested power and effect size (GPower3 calculation comes to same result).

The sentence-framing is appropriate, too.

Items that I would like to hear more about are as follows:

- Attention getters: Attention-getters are usually displayed in IPI in the centre of the screen – also to avoid side-bias – why would they be displayed at the side? Why is separate analysis interesting? Do the authors worry that the children will drop out due to the repetition of the stimuli? The attention-getter stimuli are not described – if they are more interesting than the other stimuli, then they would obviously attract more attention, ie one could argue if they are made to fulfil their purpose, then they have to be more interesting – but then they cannot be compared to the other stimuli.
- Repetition: Why will stimuli be repeated? The design could also be tested between-subjects.

- Choice of stimuli: It is appreciated how very hard it is to find appropriate early word stimuli for this study, however, when it comes to the selection of stimuli, the authors need to make sure that they are not introducing accidental confounders. For example, comparing the comprehension condition (food-related and bath-related item) and the context-related conditions and frequency-related conditions with other comprehension raises some questions. Food-related and bath-related items under Comprehension condition are really food items and body parts. In the context-related condition, food-related and bath-related items are 1 food (drink “milk”) and 2 “kitchen cupboard items” and a piece of furniture (table). I wonder whether this could be problematic. While the 3 items are context-related, albeit from 2 different semantic/conceptual categories, milk is another food item. It might be useful to exchange the food item (milk) with “bowl” or “chair” which are early words, too, so at least the items are all non-foods.

The authors write: Crucially, words within each picture-pair and their frequency-matched substitute words will refer to objects from different categories, consequently minimizing any confounding bias at a semantic and/or contextual levels.

Do the authors mean different categories to food and bath-time? Water in the frequency-match is related to the bath-routine.

It will be vital to make sure the stimuli really only vary on the dimensions chosen – context and frequency. Introducing more variables – same category / different category) may make findings hard to interpret and has the potential to confound the study as the authors agree.

The authors could use the Norwegian CDI data to help with stimulus choice for the age range.

- Frequency: It is unclear how the items in the frequency-match condition are frequency matched (range from 2-255) and with which items – within the pair or between conditions?

- Sound stimuli: Pre-recorded native Norwegian female – then presented to parents over headphones and then repeat to child – why introduce this variation? What if fathers attend versus mothers? In addition, it will be impossible to match target onset with word onset between children. Why not simply use all pre-recorded stimuli and avoid this variation? Baseline and post-word onset times can then also be made identical – this helps with ease of analysis, too.

- Analysis – please explain more clearly what data is intended for comparison.

- Whether the clarity and degree of methodological detail would be sufficient to replicate exactly the proposed experimental procedures and analysis pipeline

- Procedure & Stimuli:

- Detail and clarity is not enough yet to replicate the procedure or stimuli, eg the onset of the auditory input from the parents may vary from each parent to the next – how would this be replicated?

- It is unclear whether Norwegian CDIs are collected from the children to be tested to see if their individual word knowledge fits with the stimuli selected – this would be vital as children look at stimuli they have a name for over stimuli they cannot name yet.

- Visual stimuli are not described – as especially children’s early word-object-mappings depend on how typical these objects are, visual stimuli should be assessed for typicality in Norway and described for replication (size, background, colour or B&W, etc.). Ideally, and wherever possible, animate stimuli should not be shown with inanimate stimuli as animates tend to be more interesting to children than inanimates.

- Attention-getters are not described.

- The authors say “The pictures will remain on the screen for 3500 ms (after the target-word onset) – it is unclear how this will be controlled

- Whether the authors provide a sufficiently clear and detailed description of the methods to prevent undisclosed flexibility in the experimental procedures or analysis pipeline

Not yet enough information given and also, the analysis needs to be clarified. Which data is compared to which other data in the t-tests? Also unclear why this is not tested between-subjects to avoid repetition (and drop-out)?

Whether the authors have considered sufficient outcome-neutral conditions (e.g. positive controls) for ensuring that the results obtained are able to test the stated hypotheses
Yes, but please see comments above that need to be addressed.

Other:

Introduction:

In the introduction, it may be more useful not to speak of concrete versus abstract words – but use the appropriate linguistic terms instead (eg verbs, etc.). “Abstract word” can be easily confused with abstract noun (love, belief, etc.).

Babiness is not a good word. Saliency seems to be meant, or infant-relatedness?

Spoon over nose: “‘nose’, which is heard in less spatially and linguistically distinct contexts.” This is unclear - children may only hear their nose mentioned in highly-specific contexts, eg when they hearing and playing baby rhymes when brought to bed, so this needs careful consideration.

Author's Response to Decision Letter for (RSOS-172200.R0)

See Appendix A.

RSOS-180415.R0

Review form: Reviewer 1

Is the language acceptable?

Yes

Do you have any ethical concerns with this paper?

No

Have you any concerns about statistical analyses in this paper?

Yes

Recommendation?

Major revision

Comments to the Author(s)

I was very pleased to read the revised version of the registered report “Word knowledge in 6-9-month-old infants. Recognition without comprehension?” The authors have greatly increased their manuscript’s clarity and the replicability of their study.

I still have three major remarks, one concerning the frequency explanation, one regarding the hypothesis testing / result interpretation, and one about the control. I am also adding some minor remarks that mainly concern writing.

1. Explain the frequency account with an example as early as possible.

I am still not sure I can completely follow the frequency account. In the introduction (page 2 of the manuscript, middle paragraph), you write:

1. Frequent object-word-co-occurrences before test lead to more looks to the frequent object when hearing the label. (Note: this is different to simply matching in frequency).

2. Different word frequencies lead to biases, in either direction. (But don't you predict a strong familiarity preference for frequent words? How does a novelty preference factor in? – Your response to the question about novelty also doesn't address this issue.)
3. Infants use frequency to disambiguate between objects. (What does that mean? Yes, they can distinguish the two objects, but we assume they can do so without frequency? What role does the label play then?)

I still cannot follow that logic. If I go just by what I read (and summarize above), then we would see a consistent preference for one of the two objects if there is a striking difference in frequency. But that would mean that in a typical difference-score type of analysis (like BS12 do, if I am not mistaken) you would not see a difference and thus could not conclude that infants recognized the words. In other words, the frequency account as I read it in the paper does not match the data. What you write on page 3 is then in my opinion something different (and it makes a different prediction), namely infants track word and object frequencies and look to what is matching in frequency in their input (i.e. high/low frequency object will be looked at when hearing a high/low frequency label OR they look to the high frequency object when hearing a frequent word and don't show a preference otherwise). I think it would be important to make this clear as early as possible (and maybe clarify what you think happens with the less frequent word/object pair and how your account "survives" the paired analyses in BS12).

The example on page 3 just shows that there are frequency differences in the stimuli of BS12, but it doesn't really clarify what you think is happening. You nicely explain the context-related alternative in the following paragraph with a concrete example, a similar explanation here might help.

A visual of the alternative explanations might also help, showing both what infants 'think' and how they behave in the different scenarios, but that is entirely up to the authors. I still consider the study to be an important contribution, it would just be great if the authors' reasoning would be as clear as possible.

I don't think it's required for the paper but because it would be useful to support their argument, I'd like to explain my comment regarding effect sizes further: if frequency differences are an important additional cue, then you would predict larger looking time differences for those very mismatched pairs than more balanced pairs, correct? You could extract differences and variances from Figure 2 of BS12 (among others) and then plot effect sizes against frequency ratios or differences. It's only 6 pairs, but for each pair you have two estimates. Note for example that hand-yogurt is not showing a difference in the older age group, but more in line with your alternative explanation hair-banana seems to work well.

2. Interpret the Bayes factors and effect sizes and do not interpret the difference between significant and non-significant tests

The statistical analyses have been improved substantially as well, and the authors mention Bayes factors and effect sizes, which is great if you want to argue from non-significant effects. What is missing though is how you will use Bayes factors to interpret your results, particularly on pages 12-13 you should integrate Bayes factors, because those sections still only rely on significant results of t-tests and I am missing ways to assess graded predictions. A (non)significant difference does not assess graded predictions, even if you change the comparison.

You further explicitly plan to compare whether a t-test was significant and another was not, without looking at effect sizes, Bayes factors, or interactions (page 12, bottom and 13, 3rd paragraph). This does not work, and I must say I got lost in this paragraph overall. If you want to make statements about differences between conditions (matching versus related), you must test these in one test and not compare (non)significant outcomes (which might yield a Bayes factor near 1, what would you do then?)

Make sure to mention whether you will correct for multiple comparisons (I think you should), and if not why not.

3. Add a distractor to control trials

The authors have added a control to check whether infants are on task, which I consider a great improvement over the typical design. However, since there is currently no distractor, what infants do in the control trials is very different from what the authors claim it assesses. As with attention getters, infants will look at a single object if it's the only thing on the screen. There is no assessment whether they react to the label and I would expect a ceiling effect that makes such an assessment actually impossible (if you were to compare looks before to looks after naming, and you have to introduce a baseline correction to really get at the naming effect as you claim on page 11).

Minor remarks:

- Throughout the paper: Make sure you cite all R packages (including pwr). This link will explain how to get citation data for R packages: <https://stat.ethz.ch/R-manual/R-devel/library/utils/html/citation.html>
- Throughout the paper: "Related" is a very unclear label for a condition, do you mean context or frequency or both?
- P1: What is the target image for abstract words (night-night)? Is this mention relevant here?
- P1: The example in paragraph 3 is inconsistent: banana-hand is introduced, then hair is mentioned later as label.
- P1, end of 3rd paragraph: It might be useful to mention that we're quite sure that infants can distinguish all objects visually.
- P2, 2nd paragraph: Is shape part of semantics? Shape doesn't come back, and I don't think of ball and bowl to be semantically related but visually similar. It might be good to be as clear as possible here.
- P2, 2nd paragraph: The goals of this paper come quite sudden and before you introduce your alternative explanations, maybe move this to the end of the introduction or rephrase?
- The subheadline on p2 is a bit unclear, as you already mention factors modulating word comprehension in the preceding section, how about something like: Two alternative account for infants' behavior in word recognition studies?
- P2, 3rd paragraph, 1st sentence: either "a recent..." or "analysEs" (pl)
- Idem: Not all studies use concrete words as you reviewed on page 1
- P2, bottom paragraph: Could you either briefly introduce the units or use another measure of contextuality?
- Idem: Why the " around predicted?
- Idem: Can you elaborate on the less contextually distinct words as you do on the more distinct words?
- P3, 3rd paragraph: missing "explanations" after "alternative" in the last sentence.
- P3, bottom: I think it's good to mention early on that you also replicate the study in a new language, to me that is a strength of your work and a much needed expansion.
- P4, top paragraph: What do you mean with "knowledge/performance"?
- Idem: "as" or "compared to" (I would say)
- P4, 3rd paragraph, 1st sentence: "two additional typeS of trials"
- P5, Participants: "Shall"-> "Should"
- Idem: Add information about whether you obtain informed consent.
- Idem: Add preterm birth and developmental delays to the exclusion criteria.
- P6, figure 1: Isn't couch also contextually related to carpet?
- Figures 2,3: Increase font sizes, please
- Figure 2: Related? I assume you mean contextually/frequency related? Could you label which is which? I also am not sure you need to show the same images 4 times, you can mark the respective targets and/or have one condition be above and the other below the respective image
- P8: Can you make clear whether BS12 used different pictures or not?
- P9, Procedure, 1st sentence: "explain the study (to them)" (I'd think)

- P9, 5th paragraph: Can you refer back to the respective figure? (In general you can rely more on the figures in the text).
- P9 and elsewhere: Try to avoid “)(“ and join text in parentheses, separated by a semi-colon.
- P 10, bottom: A failure to fixate in ALL control trials? 50%?
- Idem: “In this case” (not conditions, those are experimental).
- Idem: Ideally streamline your exclusion. First exclude all with failed calibration, etc, THEN perform the quality check. Also make sure to re-apply your child-level criteria after excluding single trials.
- P12: I don’t follow where 100 diff_prop scores come from.
- P12, (3); P13, (3) Related vs matching trials: Add “testS” in the headline to make clear that you want to conduct multiple tests.
- P14: “Also” is not the best start of a new section.

Review form: Reviewer 2

Is the language acceptable?

Yes

Do you have any ethical concerns with this paper?

No

Have you any concerns about statistical analyses in this paper?

No

Recommendation?

Accept with minor revision

Comments to the Author(s)

The authors have done a great job of addressing my concerns and those of the other reviewers. I’ll limit my comments to my remaining (relatively minor) concerns:

Page 2 and elsewhere: The authors have overall addressed my concern regarding semantic/conceptual relationships versus context. I still feel they could further differentiate between category membership that is conceptual (i.e. non-linguistic) versus semantic (i.e. labeling-based). Their study doesn’t really differentiate these two possibilities – I don’t see this as a fatal flaw, but worth acknowledging somewhere.

Analysis section: Overall I find the analyses to be very clear and appropriate. My one remaining concern is somewhat nitpicky, but I still feel important for an RR. In the post hoc power analysis, the authors discuss carefully what they “expect” to find, but do not explicitly describe what they would do/how they would interpret their finding should their power expectations not be met. This leaves open a small window of “researcher degrees of freedom” that should be squared away.

Wording:

In a few places in the manuscript (e.g. page 5), the authors use the term “the latter”. I’d suggest avoiding this in each case, as they are all places where the reader is trying to follow a complex line of reasoning, and it adds to the reader’s burden.

“Big data” – I would avoid using this term as it means very different things to different people.

Decision letter (RSOS-180415.R0)

12-Apr-2018

Dear Dr Kartushina,

The Editors assigned to your revised Stage 1 Registered Report ("Word knowledge in 6-9-month-old infants. Recognition without comprehension?") have now received comments from reviewers. We would like you to revise your paper in accordance with the referee and editors suggestions which can be found below (not including confidential reports to the Editor). Please note this decision does not guarantee eventual acceptance.

Please submit a copy of your revised paper within three weeks (i.e. by the 04-May-2018). If we do not hear from you within this time then it will be assumed that the paper has been withdrawn. In exceptional circumstances, extensions may be possible if agreed with the Editorial Office in advance. We do not allow multiple rounds of revision so we urge you to make every effort to fully address all of the comments at this stage. If deemed necessary by the Editors, your manuscript will be sent back to one or more of the original reviewers for assessment. If the original reviewers are not available we may invite new reviewers.

Kind regards,
Andrew Dunn
Royal Society Open Science
openscience@royalsociety.org

on behalf of Professor Chris Chambers (Registered Reports Editor, Royal Society Open Science)
openscience@royalsociety.org

Associate Editor Comments to Author (Professor Chris Chambers):

Associate Editor: 1

Comments to the Author:

Two of the three original reviewers have reappraised the manuscript. Both agree that the submission is greatly improved and has moved significantly closer to IPA. However, a number of significant issues remain to be addressed before we can proceed further. Reviewer 1 continues to be concerned about the rationale of the frequency manipulation, and how NHST and Bayesian

inferences will be combined in interpretation of results (please make clear which will dominate in the interpretation of outcomes). Greater clarity is required in both cases. The reviewer also suggests a methodological improvement for the control condition and offers a host of detailed edits for including clarity throughout. Reviewer 2 is mostly satisfied but also questions whether the statistical plan is sufficiently detailed to prevent researcher degrees of freedom. Given the extent of improvement achieved in the first round, the authors are offered one last opportunity to address these issues before a final Stage 1 decision is reached.

Comments to Author:

Reviewer: 1

Comments to the Author(s)

I was very pleased to read the revised version of the registered report "Word knowledge in 6-9-month-old infants. Recognition without comprehension?" The authors have greatly increased their manuscript's clarity and the replicability of their study.

I still have three major remarks, one concerning the frequency explanation, one regarding the hypothesis testing / result interpretation, and one about the control. I am also adding some minor remarks that mainly concern writing.

1. Explain the frequency account with an example as early as possible.

I am still not sure I can completely follow the frequency account. In the introduction (page 2 of the manuscript, middle paragraph), you write:

1. Frequent object-word-co-occurrences before test lead to more looks to the frequent object when hearing the label. (Note: this is different to simply matching in frequency).
2. Different word frequencies lead to biases, in either direction. (But don't you predict a strong familiarity preference for frequent words? How does a novelty preference factor in? - Your response to the question about novelty also doesn't address this issue.)
3. Infants use frequency to disambiguate between objects. (What does that mean? Yes, they can distinguish the two objects, but we assume they can do so without frequency? What role does the label play then?)

I still cannot follow that logic. If I go just by what I read (and summarize above), then we would see a consistent preference for one of the two objects if there is a striking difference in frequency. But that would mean that in a typical difference-score type of analysis (like BS12 do, if I am not mistaken) you would not see a difference and thus could not conclude that infants recognized the words. In other words, the frequency account as I read it in the paper does not match the data. What you write on page 3 is then in my opinion something different (and it makes a different prediction), namely infants track word and object frequencies and look to what is matching in frequency in their input (i.e. high/low frequency object will be looked at when hearing a high/low frequency label OR they look to the high frequency object when hearing a frequent word and don't show a preference otherwise). I think it would be important to make this clear as early as possible (and maybe clarify what you think happens with the less frequent word/object pair and how your account "survives" the paired analyses in BS12).

The example on page 3 just shows that there are frequency differences in the stimuli of BS12, but it doesn't really clarify what you think is happening. You nicely explain the context-related alternative in the following paragraph with a concrete example, a similar explanation here might help.

A visual of the alternative explanations might also help, showing both what infants 'think' and how they behave in the different scenarios, but that is entirely up to the authors. I still consider the study to be an important contribution, it would just be great if the authors' reasoning would be as clear as possible.

I don't think it's required for the paper but because it would be useful to support their argument, I'd like to explain my comment regarding effect sizes further: if frequency differences are an

important additional cue, then you would predict larger looking time differences for those very mismatched pairs than more balanced pairs, correct? You could extract differences and variances from Figure 2 of BS12 (among others) and then plot effect sizes against frequency ratios or differences. It's only 6 pairs, but for each pair you have two estimates. Note for example that hand-yogurt is not showing a difference in the older age group, but more in line with your alternative explanation hair-banana seems to work well.

2. Interpret the Bayes factors and effect sizes and do not interpret the difference between significant and non-significant tests

The statistical analyses have been improved substantially as well, and the authors mention Bayes factors and effect sizes, which is great if you want to argue from non-significant effects. What is missing though is how you will use Bayes factors to interpret your results, particularly on pages 12-13 you should integrate Bayes factors, because those sections still only rely on significant results of t-tests and I am missing ways to assess graded predictions. A (non)significant difference does not assess graded predictions, even if you change the comparison.

You further explicitly plan to compare whether a t-test was significant and another was not, without looking at effect sizes, Bayes factors, or interactions (page 12, bottom and 13, 3rd paragraph). This does not work, and I must say I got lost in this paragraph overall. If you want to make statements about differences between conditions (matching versus related), you must test these in one test and not compare (non)significant outcomes (which might yield a Bayes factor near 1, what would you do then?)

Make sure to mention whether you will correct for multiple comparisons (I think you should), and if not why not.

3. Add a distractor to control trials

The authors have added a control to check whether infants are on task, which I consider a great improvement over the typical design. However, since there is currently no distractor, what infants do in the control trials is very different from what the authors claim it assesses. As with attention getters, infants will look at a single object if it's the only thing on the screen. There is no assessment whether they react to the label and I would expect a ceiling effect that makes such an assessment actually impossible (if you were to compare looks before to looks after naming, and you have to introduce a baseline correction to really get at the naming effect as you claim on page 11).

Minor remarks:

- Throughout the paper: Make sure you cite all R packages (including pwr). This link will explain how to get citation data for R packages: <https://stat.ethz.ch/R-manual/R-devel/library/utils/html/citation.html>
- Throughout the paper: "Related" is a very unclear label for a condition, do you mean context or frequency or both?
- P1: What is the target image for abstract words (night-night)? Is this mention relevant here?
- P1: The example in paragraph 3 is inconsistent: banana-hand is introduced, then hair is mentioned later as label.
- P1, end of 3rd paragraph: It might be useful to mention that we're quite sure that infants can distinguish all objects visually.
- P2, 2nd paragraph: Is shape part of semantics? Shape doesn't come back, and I don't think of ball and bowl to be semantically related but visually similar. It might be good to be as clear as possible here.
- P2, 2nd paragraph: The goals of this paper come quite sudden and before you introduce your alternative explanations, maybe move this to the end of the introduction or rephrase?

- The subheadline on p2 is a bit unclear, as you already mention factors modulating word comprehension in the preceding section, how about something like: Two alternative account for infants' behavior in word recognition studies?
- P2, 3rd paragraph, 1st sentence: either "a recent..." or "analyses" (pl)
- Idem: Not all studies use concrete words as you reviewed on page 1
- P2, bottom paragraph: Could you either briefly introduce the units or use another measure of contextuality?
- Idem: Why the " around predicted?
- Idem: Can you elaborate on the less contextually distinct words as you do on the more distinct words?
- P3, 3rd paragraph: missing "explanations" after "alternative" in the last sentence.
- P3, bottom: I think it's good to mention early on that you also replicate the study in a new language, to me that is a strength of your work and a much needed expansion.
- P4, top paragraph: What do you mean with "knowledge/performance"?
- Idem: "as" or "compared to" (I would say)
- P4, 3rd paragraph, 1st sentence: "two additional types of trials"
- P5, Participants: "Shall"-> "Should"
- Idem: Add information about whether you obtain informed consent.
- Idem: Add preterm birth and developmental delays to the exclusion criteria.
- P6, figure 1: Isn't couch also contextually related to carpet?
- Figures 2,3: Increase font sizes, please
- Figure 2: Related? I assume you mean contextually/frequency related? Could you label which is which? I also am not sure you need to show the same images 4 times, you can mark the respective targets and/or have one condition be above and the other below the respective image
- P8: Can you make clear whether BS12 used different pictures or not?
- P9, Procedure, 1st sentence: "explain the study (to them)" (I'd think)
- P9, 5th paragraph: Can you refer back to the respective figure? (In general you can rely more on the figures in the text).
- P9 and elsewhere: Try to avoid ")((" and join text in parentheses, separated by a semi-colon.
- P 10, bottom: A failure to fixate in ALL control trials? 50%?
- Idem: "In this case" (not conditions, those are experimental.
- Idem: Ideally streamline your exclusion. First exclude all with failed calibration, etc, THEN perform the quality check. Also make sure to re-apply your child-level criteria after excluding single trials.
- P12: I don't follow where 100 diff_prop scores come from.
- P12, (3); P13, (3) Related vs matching trials: Add "tests" in the headline to make clear that you want to conduct multiple tests.
- P14: "Also" is not the best start of a new section.

Reviewer: 2

Comments to the Author(s)

The authors have done a great job of addressing my concerns and those of the other reviewers. I'll limit my comments to my remaining (relatively minor) concerns:

Page 2 and elsewhere: The authors have overall addressed my concern regarding semantic/conceptual relationships versus context. I still feel they could further differentiate between category membership that is conceptual (i.e. non-linguistic) versus semantic (i.e. labeling-based). Their study doesn't really differentiate these two possibilities – I don't see this as a fatal flaw, but worth acknowledging somewhere.

Analysis section: Overall I find the analyses to be very clear and appropriate. My one remaining concern is somewhat nitpicky, but I still feel important for an RR. In the post hoc power analysis,

the authors discuss carefully what they “expect” to find, but do not explicitly describe what they would do/how they would interpret their finding should their power expectations not be met. This leaves open a small window of “researcher degrees of freedom” that should be squared away.

Wording:

In a few places in the manuscript (e.g. page 5), the authors use the term “the latter”. I’d suggest avoiding this in each case, as they are all places where the reader is trying to follow a complex line of reasoning, and it adds to the reader’s burden.

“Big data” - I would avoid using this term as it means very different things to different people.

Author's Response to Decision Letter for (RSOS-180415.R0)

See Appendix B.

RSOS-180711.R0

Review form: Reviewer 1

Is the language acceptable?

Yes

Do you have any ethical concerns with this paper?

No

Have you any concerns about statistical analyses in this paper?

No

Recommendation?

Accept in principle

Comments to the Author(s)

I enjoyed reading this revision of the paper a great deal and find it substantially improved. The introduction makes a clear case, and offers interesting predictions as well as making clear where this study deviated from BS12.

I am still not fully convinced by the authors' argument for their control trial (I expect a ceiling effect so that the naming portion will not induce a change in looking behavior, so it doesn't really do what you say), but another manipulation check is built into the study with their direct replication of BS12, so it should be possible to discern whether the study "worked" as we expect some recognition response for the matching trials as well.

(I wonder whether you need the control trials at all, then, also because they constitute a deviation from the original study, but I do not consider this a major issue either way.)

Minor comments:

on page 4, line 27, maybe mention that BS12 also used the IPL and this is where your studies are closely matched?

Review form: Reviewer 2

Is the language acceptable?

Yes

Do you have any ethical concerns with this paper?

No

Have you any concerns about statistical analyses in this paper?

I do not feel qualified to assess the statistics

Recommendation?

Accept in principle

Comments to the Author(s)

Given the minor nature of my comments on the last round of revisions, I have not done a full re-review, but focused on those specific points.

1. Regarding my first point, I'm not sure that the authors have truly addressed it, though it may be because I did not express myself very well. You could replace all the use of the word "semantic" in the introduction with "conceptual" and everything would still hold. Perhaps another way of saying it is that the discussion does not address whether the categories in question are linguistic categories or general cognitive ones (infants can, in some theories, group/classify objects meaningfully and not just perceptually without that knowledge being linguistic knowledge). Again I don't see this as a fundamental flaw, but I would like to see it mentioned somewhere for clarity given the other hairs that are being split.
2. Again, I don't think the authors directly addressed my analytic concern regarding power checks. However, the shift in focus away from NHST toward interpretation of effect size and Bayesian analysis as requested by Reviewer 1 makes this concern less important.

If another round of revisions is undertaken, I think it would strengthen the paper to take another stab at these two concerns. However, they are sufficiently minor that I would not want acceptance to be held up only on their account. In my opinion the manuscript acceptable in its current form and I look forward to seeing the results of the study and Stage 2 manuscript.

Decision letter (RSOS-180711.R0)

23-May-2018

Dear Dr Kartushina

On behalf of the Editor, I am pleased to inform you that your Manuscript RSOS-180711 entitled

"Word knowledge in 6-9-month-old infants. Recognition without comprehension?" has been accepted in principle for publication in Royal Society Open Science. The reviewers' and editors' comments are included at the end of this email.

You may now progress to Stage 2 and complete the study as approved. Before commencing data collection we ask that you:

- 1) Update the journal office as to the anticipated completion date of your study.
- 2) Register your approved protocol on the Open Science Framework (<https://osf.io/>) or other recognised repository, either publicly or privately under embargo until submission of the Stage 2 manuscript. Please note that a time-stamped, independent registration of the protocol is mandatory under journal policy, and manuscripts that do not conform to this requirement cannot be considered at Stage 2. The protocol should be registered unchanged from its current approved state, with the time-stamp preceding implementation of the approved study design. We recommend using the dedicated portal for registering Stage 1 RRs at the Open Science Framework: <https://osf.io/rr/>

Following completion of your study, we invite you to resubmit your paper for peer review as a Stage 2 Registered Report. Please note that your manuscript can still be rejected for publication at Stage 2 if the Editors consider any of the following conditions to be met:

- The results were unable to test the authors' proposed hypotheses by failing to meet the approved outcome-neutral criteria.
- The authors altered the Introduction, rationale, or hypotheses, as approved in the Stage 1 submission.
- The authors failed to adhere closely to the registered experimental procedures. Please note that any deviations from the approved experimental procedures must be communicated to the editor immediately for approval, and prior to the completion of data collection. Failure to do so can result in revocation of in-principle acceptance and rejection at Stage 2 (see complete guidelines for further information).
- Any post-hoc (unregistered) analyses were either unjustified, insufficiently caveated, or overly dominant in shaping the authors' conclusions.
- The authors' conclusions were not justified given the data obtained.

We encourage you to read the complete guidelines for authors concerning Stage 2 submissions at <http://rsos.royalsocietypublishing.org/content/registered-reports>. Please especially note the requirements for data sharing, reporting the URL of the independently registered protocol, and that withdrawing your manuscript will result in publication of a Withdrawn Registration.

Once again, thank you for submitting your manuscript to Royal Society Open Science and we look forward to receiving your Stage 2 submission. If you have any questions at all, please do not hesitate to get in touch. We look forward to hearing from you shortly with the anticipated submission date for your stage two manuscript.

Kind regards,

Andrew Dunn
Royal Society Open Science
openscience@royalsociety.org

on behalf of Professor Chris Chambers (Registered Reports Editor, Royal Society Open Science)
openscience@royalsociety.org

Associate Editor Comments to Author (Professor Chris Chambers):

The reviewers' concerns are now sufficiently addressed to permit progression to in principle acceptance.. If the authors wish to implement Reviewer 1's recommended minor revision (re p4 line 27), please do before registering the approved protocol on the OSF.

Reviewers' comments to Author:

Reviewer: 2

Comments to the Author(s)

Given the minor nature of my comments on the last round of revisions, I have not done a full review, but focused on those specific points.

1. Regarding my first point, I'm not sure that the authors have truly addressed it, though it may be because I did not express myself very well. You could replace all the use of the word "semantic" in the introduction with "conceptual" and everything would still hold. Perhaps another way of saying it is that the discussion does not address whether the categories in question are linguistic categories or general cognitive ones (infants can, in some theories, group/classify objects meaningfully and not just perceptually without that knowledge being linguistic knowledge). Again I don't see this as a fundamental flaw, but I would like to see it mentioned somewhere for clarity given the other hairs that are being split.
2. Again, I don't think the authors directly addressed my analytic concern regarding power checks. However, the shift in focus away from NHST toward interpretation of effect size and Bayesian analysis as requested by Reviewer 1 makes this concern less important.

If another round of revisions is undertaken, I think it would strengthen the paper to take another stab at these two concerns. However, they are sufficiently minor that I would not want acceptance to be held up only on their account. In my opinion the manuscript acceptable in its current form and I look forward to seeing the results of the study and Stage 2 manuscript.

Reviewer: 1

Comments to the Author(s)

I enjoyed reading this revision of the paper a great deal and find it substantially improved. The introduction makes a clear case, and offers interesting predictions as well as making clear where this study deviated from BS12.

I am still not fully convinced by the authors' argument for their control trial (I expect a ceiling effect so that the naming portion will not induce a change in looking behavior, so it doesn't really do what you say), but another manipulation check is built into the study with their direct replication of BS12, so it should be possible to discern whether the study "worked" as we expect some recognition response for the matching trials as well.

(I wonder whether you need the control trials at all, then, also because they constitute a deviation from the original study, but I do not consider this a major issue either way.)

Minor comments:

on page 4, line 27, maybe mention that BS12 also used the IPL and this is where your studies are closely matched?

Author's Response to Decision Letter for (RSOS-180711.R0)

See Appendix C.

RSOS-180711.R1 (Revision)

Review form: Reviewer 1

Is the language acceptable?

Yes

Do you have any ethical concerns with this paper?

No

Have you any concerns about statistical analyses in this paper?

Yes

Recommendation?

Major revision

Comments to the Author(s)

The present manuscript is a Stage 2 Registered Report, where the authors completed their study on 6-9 month old Norwegian infants' early word comprehension.

The introduction and relevant parts of the methods section seem identical, and the authors have added analyses (confirmatory and exploratory), and a discussion section contextualizing their findings.

I have a few remarks to further improve the manuscript and have it aligned with the Registered Report format. I will first list major issues that I consider necessary to address and then minor recommendations that the authors can take under advice / reviewer preferences. (Note: I am using the page numbers at the bottom of the page).

First, I am missing the analysis of the control, I found no section or data on it. In the registered report (I accessed it via the OSF repository, thanks for putting it there!) it is the first planned analysis on page 11.

In general, I would not treat those two documents as separately as done in the manuscript. While the reader can certainly access the stage 1 version, the manuscript would be easier to follow if it were self-contained and details like preprocessing were discussed transparently and in depth in this manuscript, and any deviations from the plans were highlighted (i.e. simply stating "as preregistered, we did ...[section from the stage 1 manuscript]").

Second, I am missing Bayes Factors, effect sizes, and sometimes all statistics for non-significant analyses. All this information is crucial, particularly given (1) your testing of an additional sample and (2) your interpreting null results as evidence for the null hypothesis (difficult without effect sizes and BF) in the discussion section in particular, but also in the results (e.g. page 14).

I also see an interpretation of condition differences (8-9 month olds exploratory analyses), without having tested whether this difference is significant in a reduced model or referring to the full linear mixed model that showed no significant effect (page 18). I would caution against overstating any findings in this case, particularly since those analyses are strictly exploratory.

Third, I would recommend explicitly labelling the exploratory analyses (unplanned statistical tests, additional participants) as such, and maybe adding an interim discussion before such a section that discusses the implications of all the planned analyses in isolation plus motivates the

additional analyses. With such an interim discussion, the implications of all results of your planned analyses become clearer and it would be easier to separate them from the exploratory (and I must say very interesting) findings, which require follow-up confirmatory work.

Fourth, I could not find the conditional part of your planned analyses you refer to at the bottom of page 14. In other words, and I might have missed it, I cannot find the part of the stage 1 registered report where you state that you will not include analyses if some results are not significant.

I personally would prefer to strictly follow the plan (again, also to have everything line up).

Fifth, I disagree with your interpretation of power when discussing your results in relation to not replicating BS12 (line 12, page 23). A single study does not provide a reliable effect size estimate, and we do not know the true underlying effect and its distribution over infant age and language. (Which also leads me to wonder about your age distribution compared to the one in BS12). You powered your study for an effect of a certain size with 50 participants (although note that some analyses had less than that number of participants, so the effect size you can observe with reasonable power is necessarily larger). This effect might very well be smaller in reality (see regression to the mean: https://en.wikipedia.org/wiki/Regression_toward_the_mean).

I also recommend to be clear in the discussion which points refer to which sample. (This might be helped with the interim discussion I mention above somewhat).

Through sequential analyses when adding 20 participant, you cannot interpret your p-values in the same way for those analyses (had you preregistered sequential sampling, I would have recommended to simply correct p accordingly, but here I would recommend to largely disregard them), so we need to focus on the Bayes factors, confidence intervals, and effect sizes and consider those for that sample. (See also my point above on providing this information in the manuscript for *all* results). It is still interesting to compare outcomes, I agree, and I appreciate your discussion of possible causes of the difference (although do not forget regression to the mean and simple measurement variance), but the basis for this is not very solid when using power to reason.

Sixth, regarding the materials shared on OSF, I would recommend (1) sharing the code in an .R or preferably .Rmd (RMarkdown) script. Particularly the latter is very convenient, because you can share the code and the output with your data very easily and transparently.

Right now, everything is only shared as pdf, which shows your analysis pipeline but makes it difficult to reproduce your results

(https://osf.io/54gs9/?view_only=a4f61a751c4b478a814db3a54ac51ead).

Here is more information on RMarkdown: <https://rmarkdown.rstudio.com/>

As minor point on the supplementary materials, I would also recommend adding headlines that clarify which analyses were preregistered, and to which parts of the manuscript they correspond. The current text is already very useful, I simply want to recommend to align it more closely with the stage 1 manuscript.

It would also be great to add descriptive information (age distribution, sex, overall looking times, CDI scores by age, etc) to the analysis document, simply to have it in one place and to be able to compare the two samples more easily.

Finally, on that topic, I recommend including at the end some information about R and package versions, a good way to do this is to include the following command:

https://devtools.r-lib.org/reference/session_info.html

Minor issues

Can you explicitly mention both N at the end of the top paragraph of page 12?

Some instances of "observed" seem a bit unconventional, maybe "inspected" fits better? (e.g. footnote 4 page 12).

Likewise, this expression was difficult to follow for me: "if a child did not validate a trial" (page 13) - Do you mean if a child did not provide sufficient looking time during a trial?

Same page: "an important data loss in our sample" - substantial?

I assume "in total, we obtained 49 average scores for the matching trials and 45 for the related trials" (page 13) means scores for 49 and 45 participants, but it might be nice to state that explicitly.

What is the difference between "difference in target looking" and "average increase in target looking" and how is the latter computed. Both are introduced on page 13. (This might be an easy fix if you follow my major recommendations and take over more information from the stage 1 manuscript so this paper becomes self-contained).

You mention ManyBabies (page 18, footnote 7), and refer to the preprint, but wouldn't it be more appropriate to include a citation to it instead of the link? Otherwise the reader might not know that the link is a manuscript (which might be updated or the link might change).

You mention on page 23 that there is a higher number of vowel cues in Norwegian that infants need to acquire to master the vowel system and that this might delay the growth of their lexicon, but previously cited data from Wordbank led me to believe that at least according to parental reports, Norwegian and American English seem to be relatively similar. Was your sample different from the expected scores? Could you add some words to reconcile your proposal with CDI in general?

And are there studies on lexicon growth / word recognition in other languages with a more high-dimensional phonology that support your proposal?

Your figures are very nice but note that the colors are indistinguishable in black-white print and might not be colorblind-friendly. I recommend using e.g. circles and triangles (or some other shape pair) to make the distinction easier.

Review form: Reviewer 2

Is the language acceptable?

Yes

Do you have any ethical concerns with this paper?

No

Have you any concerns about statistical analyses in this paper?

No

Recommendation?

Accept with minor revision

Comments to the Author(s)

Please note: I am assuming the authors would have highlighted any changes from the Stage 1 manuscript and have not carefully compared it with the current document.

Norwegian 6-9 month old infants were tested in a word learning task. In the pre-registered analyses, no evidence of word learning was found. However, in exploratory unregistered analyses, evidence at 8-9 months was found, specifically when there was a mismatch in the frequency of the paired items.

Overall, I think the manuscript is in good shape and I see no substantive issues to prevent publication. However, there are some things that I would recommend changing before publication, particularly given the emphasis in the final manuscript on analyses that were not part of the original preregistration. This is not problematic in itself, but it does lead to the need for closer scrutiny and care in wording.

1. Given the heavy salience (and impact on the conclusions) of the unregistered analyses within the “registered” report, it would be helpful to make it a bit more explicit when the transition is made. Perhaps the section “Additional Statistical Analyses” could be retitled as “Exploratory” or “Unregistered” Analyses?
2. I am somewhat unsure how to interpret the finding illustrated in Figure 11. While it is true that infants whose mother reported the word was “understood” showed longer looking in the frequency condition, the converse appears to be true in the context condition. The authors state that “looking times were significantly different from chance only for words reported as ‘understood’ in the frequency condition” – however, based on Figure 11, the negative effect in the context condition seems stronger than the positive effect in the frequency condition. I am aware that overinterpreting the visuals is dangerous, but in this case some additional comment seems warranted, even if the context-yes condition didn’t “reach significance” (was the test two-tailed?). Relatedly, it’s not clear to me what the “dots” represent in this figure.
3. I would be careful about the wording on Page 24 “These differences.... cannot be attributed to a lack of power in the current study”. I get the point, but there is going to be random variation across studies – just because one study has more participants than another that “got a significant effect” does not mean overall that the study is not underpowered.
4. In the same paragraph as [3], I was not terribly convinced by the argument that Norwegian is a simpler language to learn – there are just too many ways to compare languages. If the authors want to make this very specific claim, I would suggest highlighting that it is speculative.
5. I am still a bit uncomfortable with the framing of the results with English as establishing word learning AT 6 months (particularly comparatively with the current finding, e.g. in the last paragraph on page 25, “In sum...”) – the only study (Tincoff et al. X2) that found evidence specifically at 6 months (rather than over a range) tested a very small number of specific words. Bergelson typically frames her result as suggesting the emergence of knowledge of concrete words at “6-9 months”, which encompasses the 8 month old age range tested in this study. I don’t have a problem with noting the lack of finding with 6-7 month olds in this study, but I think there could be a bit more acknowledgement that this difference across the two (sets of) studies may or may not indicate a meaningful developmental difference across the populations. It is suggestive, but far from confirmatory.

Below I make some additional comments on the wording of the introduction/abstract. Given that this was already accepted through peer-review, it may well be the journal’s policy not to make further changes – however, I provide them anyway in case it is appropriate to make further edits.

1. Abstract: “explosion of claims” – this seems a bit over the top. We’re talking about a relatively small number of study.
2. I’m not sure it’s accurate to refer to name recognition as “word comprehension” (page 1 at the bottom)
3. It is perhaps noteworthy that Tincoff et al. found evidence for recognition of “hand” at 6 months given the contrast made between hand and spoon on page 4.

4. On page 5, an N should be provided for the Pearson correlation.
5. Also on page 5, I did not understand in what way book and ball are semantically congruent.

Review form: Reviewer 3

Is the language acceptable?

Yes

Do you have any ethical concerns with this paper?

No

Have you any concerns about statistical analyses in this paper?

Yes

Recommendation?

Accept with minor revision

Comments to the Author(s)

Whether the data are able to test the authors' proposed hypotheses by passing the approved outcome-neutral criteria (such as absence of floor and ceiling effects or success of positive controls)

Not sure about this – see comments below. Some further information is needed.

Whether the Introduction, rationale and stated hypotheses are the same as the approved Stage 1 submission

Seems the case

Whether the authors adhered precisely to the registered experimental procedures

It seems stimuli and attention getter plus parts of procedure are changed from initial pilot to main study data, but in line with reviewers' comments – however, in light of the first 12 children being included, it is unclear if children in the data set have undergone different testing procedures and stimuli.

Where applicable, whether any unregistered exploratory statistical analyses are justified, methodologically sound, and informative

There is additional exploration, but it is worthwhile – it would be more elegant and economic to include all children in one analysis straight away.

Whether the authors' conclusions are justified given the data

Please see comments below

Please note that editorial decisions will not be based on the perceived importance, novelty, or clarity of the results.

Comments

Abstract

- grammatical errors – an English native speaker should read it (has witnessed – not witnesses, article missing, etc., spelling of 24-month-olds – hyphenation errors; numbers (38,000))

Introduction

- Line 9 should say 3 months later – from 9 (not 10) months as ref 4 from which presumably the examples sleeping and kissing are taken, tested 9-month-olds.

- Furthermore, sleeping and to kiss are not abstract words, but verbs. An “abstract” word could be to “think”, or “thought”. Sleeping and kissing are visible routines (ref 4 actually used “night night”, not “sleeping” – is that what the authors refer to and mistakenly call it sleeping?). In the case of kissing, this is even a highly actional and volitional event with a results and an effect on

others, ie a highly prototypical action verb. The authors made this error already in a previous version – please correct or omit the word “abstract”.

- The reporting is incorrect line 19-22: Ref 4 has shown that infants failed to show a significant increase in looking towards the target words if the standard word list was tested with items children were expected to know, but which parents had not confirmed as understood. In contrast, children did understand words that parents had confirmed as understood.
- It is curious that understanding of words as judged by parents is assessed in the result section, but not mentioned in the introduction at all. As there is a result chapter on this, this should be included in the introduction as well, and also in aims and hypotheses.
- The authors should report results in their complexity, e.g. the Ref 4 paper showed:
 - Nine-month-olds display word knowledge independent of context and without repetitions of words.
 - First words encompass not only nouns, but a range of other word classes (e.g. verbs like sleeping and kissing).
 - Parents are good at indicating which words their infants do and do not understand.

The authors need to integrate this correctly into the manuscript.

- Line 23: I would advise to phrase more carefully: “TAKEN TOGETHER, these results MAY suggest (...)”
- Line 25-26: the authors write: “however, if the objects belong to the same semantic category (e.g., a cat and a dog both represent animals), infants fail to disambiguate between them, even though they discriminate both objects visually”

Please amend to the more correct version:

However, if the objects belong to the same semantic category (e.g., a cat and a dog both represent animals), infants fail to disambiguate between them IF PARENTS HAVE NOT INDICATED THEM AS KNOWN WORDS (4), even though they discriminate both objects visually.

- L. 30-33: The authors do not portray the full story here as the conceptual level is omitted: There is ample evidence of typicality effects in early word learning (Meints et al., 1999 for nouns, see also Poulin-Dubois & Sissons; for other early word learning see Meints et al., 2002, 2004 and 2008) showing that 12-month-olds do look to a named target, even when part of same category (e.g. cat and dog) if the named items are typical. They do not link the name with atypical targets (regardless of the category being related or not). This conceptual effect gets lost, despite it having been shown to contribute significantly to early word learning. I.e. infants’ word-referent mappings are also conceptually limited to typical mappings at 12-months for nouns, and more refined at 18 months and more fully formed at 24 months. This is an important omission and should be added and integrated for a more complete picture of early word learning– especially also as the authors then go on to assess their stimuli for typicality (information on how ratings were done is not provided).
- L. 47: To sum up, early word categories may also be conceptually limited, not just semantically or perceptually limited (coarse).
- Here and in the discussion, please bear in mind limitations of corpus-analysis as this usually does not consider typicality or saliency information.
- L. 33-37 – contradiction: the authors argue here from corpus data of 1 child why “hand” must be a later word, but they have already shown evidence in the introduction that from 6 months “ infants start showing comprehension for some concrete objects, for example, body parts ‘hand’ and ‘feet’ (5)” - resolve this contradiction in argumentation, please.
- While is useful to look at frequency and the frequency argument is well made, salience and typicality and familiarity are also factors - mention in discussion at least.

Method section

- Curiously, it emerges in the method section under “stimuli” that the authors have after all considered typicality and have assessed all items for typicality, however, still omit to refer to the research that gives the reasons to do so (Meints et al., 1999, and Poulin-Dubois and Sissons) – can

this omission in references be remedied, please, in the method section and in the introduction above.

- p. 8: counterbalancing incomplete: counterbalancing of naming seems forgotten – when pairs are presented, e.g. apple on left, foot on right, in one set of trials the apple should be named, and in a different trial (different child) the foot should be named. Complete counterbalancing looks like this:

- apple foot, apple named
- apple foot, foot named
- foot apple, foot named
- foot apple, apple named

- In relation to this, was it ensured that also items on the right were named first? This is important as children have a left gaze bias, also for objects (Guo et al., 2008).

- Attention-getter: use of tense - was used not “will be used”

- Figure 2 is misleading as it leads the reader initially to wonder if 2 pairs of images were presented – please adapt acc. to what was really shown (I assume only 1 apple and 1 foot were shown at a time from descriptions below).

- p. 9: the authors wrote: “(...) minimize the number of errors due to parental mistakes – I don’t understand this point if the video is shown and parents are behind the child and “muted” – there is no space for parental mistake anyway – this seems to have been part of an earlier version and of initial pilot testing of the first children - clarify please – and clarify especially if all children taking part and counted into this data set have undergone the same procedure and seen the same stimuli, i.e. why can the 12 initially tested children be included if these had initially other stimuli and a somewhat different procedure? (parents involved in stimulus presentation – see note on this left in manuscript)

- p. 12 of 26: the authors wrote: “The word familiarity questionnaire will ask, for each of the 32 words used in the experiment, how frequently parents have used it (on a scale from 0-never to 5-very frequently) while interacting with the child or in his presence, since their baby was born.” Should read: “asked” in past tense.

- Also, why was this not asked before the study to control for frequency effects?

- Also, asking about familiarity in advance would avoid parents coming in vain as the authors report this in the exclusions: “In addition, two infants did not pass the trial inclusion criteria, because they were unfamiliar with the words depicted in 6 out of 8 picture pairs; their parents reported to never have used them with the child (or in his/her presence), since the child was born.”

- Give mean and age range in participant information, not in data processing, also provide it in months only.

- Drop-out rate is 31% (20 plus 3 kids did not finish while 50 kids finished the study) - this is high – does the procedure need improvement? Or could it be due to the impossibility of the non-match task? Time of testing? Tiredness or hunger? The manuscript does not inform about this.

- The authors wrote: “All infants in the final sample attended to the targets on the control trials, suggesting that they were engaged in the task.” Why was it not checked that infant attended also to test trials? This can easily be checked, esp. when an eye-tracker is used: attention to both stimuli in pre-naming phase should be the criterion per trial, and also in post-naming.

- Frequency data on words not given, please add in table.

Results

- It is somewhat unclear whether the pre-naming phase included the 1500ms of exposure to the 2 images, or only the 0-367ms. If the latter is the case, this is the time a child needs to change their gaze, i.e. the authors then only measure latency instead of a proper pre-naming phase. Instead, the whole 1500ms where the objects are visible should be calculated as pre-naming phase. Ideally, pre-naming would be equal to post-naming. Please clarify in all relevant parts of the manuscript.

- Animacy was not controlled for in stimulus pairs – that could potentially have destroyed any effects if children look more at animate items – and 6 out of 8 picture pairs show animals or body

parts of humans versus inanimate objects – this is far from ideal, given the salient nature of animacy in children’s early categorisation (see all the research by Jean Mandler, Quinn, Rakison, etc.). Was this checked/analysed?

- Also, could the effects be carried by car, cat, keys and banana as these are more frequent items in early language (at least in English “key” is frequent, too), and as these seem significant for older children (in figure) (car-couch / cat-keys and banana-hair).
- Potential confounding issue: Due to the way the results are calculated using only the post-naming phase, it is not possible to see potential bias in images. As items were of differing frequency for the first set within each image pair, this is a potential confounder and it would have been important to disentangle the naming effect from a frequency effect by looking at pre- and post-naming phases separately.

I.e. generally, in pre-naming children should look at both images (equally if equally salient) and the post-naming phase then shows if the name drives the looking. Thus, I am not sure why targets looking time versus distracter looking was not compared for baseline pre-naming phase versus post-naming phase (t/t+d for each)? That would have given a clear additional measure if any of the two displayed items was preferred to the other – that is, another possible reason for exclusion if items are not equally salient – albeit, the frequency condition may actually be confounded in this study as it is likely that children will prefer the more frequent item once it is named – but also potentially before it is named – only analysis of pre- versus post-naming data can show this.

- The rationale for analysis should be to use the best and most useful analysis for the data, not necessarily to replicate other researchers’ analysis.

Discussion

- Include more critical evaluation of results, including points raised above and thorough discussion of frequency differences in matching trials, and saliency / familiarity discussion that could help explain the results better and derive a more complete picture and include suggestions for future research.

Decision letter (RSOS-180711.R1)

16-Jul-2019

Dear Dr Kartushina,

The editors assigned to your Stage 2 Registered Report ("Word knowledge in 6-9-month-old infants? Not without additional frequency cues.") has now received comments from reviewers. We would like you to revise your paper in accordance with the referee and Subject Editor suggestions which can be found below (not including confidential reports to the Editor). Please note this decision does not guarantee eventual acceptance.

When submitting your revised manuscript, you must respond to the comments made by the referees and upload a file "Response to Referees" in "Section 6 - File Upload". Please use this to document how you have responded to the comments, and the adjustments you have made. In

order to expedite the processing of the revised manuscript, please be as specific as possible in your response.

- Data accessibility

If you wish to submit your supporting data or code to Dryad (<http://datadryad.org/>), or modify your current submission to dryad, please use the following link:
<http://datadryad.org/submit?journalID=RSOS&manu=RSOS-180711.R1>

- Competing interests

- Authors' contributions

- Acknowledgements

- Funding statement

Kind regards,
Royal Society Open Science Editorial Office
Royal Society Open Science

on behalf of Chris Chambers
Subject Editor, Royal Society Open Science
openscience@royalsociety.org

Associate Editor's comments (Professor Chris Chambers):

Associate Editor: 1

Comments to the Author:

The three expert reviewers who assessed the Stage 1 manuscript have now reviewed the completed Stage 2 submission. In general, the manuscript is in good shape but some revision is required to improve clarity and transparency. Both Reviewers 1 and 2 note that the conclusions must be based on the preregistered outcomes (and it appears that one preregistered analysis is missing), and that the preregistered and post hoc analyses (and any conclusions drawn thereof) must be more clearly distinguished -- including in the Abstract, Results and Discussion. Reviewers 2 and 3 also offer a range of detailed comments on the Introduction and Methods. In considering these comments, please note the overarching policy requirement for a Stage 2 RR that no changes can be made to the rationale, aims or hypotheses, even where it may seem sensible to do so in light of the results. Therefore, some of the comments raised by the reviewers will not be possible to address directly and you should respond accordingly. In revising the Introduction or Method, please do so ONLY where such changes are very minor, or to correct errors of fact or provide necessary clarifications that would otherwise mislead the reader. Where any changes (beyond the most minor) are made please also footnote the change in the manuscript to note the deviation. The Discussion can be used to provide more extensive clarifications. Note that critical assessment of the preregistered design is not a review criterion at Stage 2 as these elements have already been approved. If you have any questions about editorial requirements in addressing specific points, please email me (chambersc1@cardiff.ac.uk).

Comments to Author:

Reviewers' Comments to Author:

Reviewer: 1

Comments to the Author(s)

The present manuscript is a Stage 2 Registered Report, where the authors completed their study on 6-9 month old Norwegian infants' early word comprehension.

The introduction and relevant parts of the methods section seem identical, and the authors have added analyses (confirmatory and exploratory), and a discussion section contextualizing their findings.

I have a few remarks to further improve the manuscript and have it aligned with the Registered Report format. I will first list major issues that I consider necessary to address and then minor

recommendations that the authors can take under advice / reviewer preferences. (Note: I am using the page numbers at the bottom of the page).

First, I am missing the analysis of the control, I found no section or data on it. In the registered report (I accessed it via the OSF repository, thanks for putting it there!) it is the first planned analysis on page 11.

In general, I would not treat those two documents as separately as done in the manuscript. While the reader can certainly access the stage 1 version, the manuscript would be easier to follow if it were self-contained and details like preprocessing were discussed transparently and in depth in this manuscript, and any deviations from the plans were highlighted (i.e. simply stating "as preregistered, we did ...[section from the stage 1 manuscript]").

Second, I am missing Bayes Factors, effect sizes, and sometimes all statistics for non-significant analyses. All this information is crucial, particularly given (1) your testing of an additional sample and (2) your interpreting null results as evidence for the null hypothesis (difficult without effect sizes and BF) in the discussion section in particular, but also in the results (e.g. page 14).

I also see an interpretation of condition differences (8-9 month olds exploratory analyses), without having tested whether this difference is significant in a reduced model or referring to the full linear mixed model that showed no significant effect (page 18). I would caution against overstating any findings in this case, particularly since those analyses are strictly exploratory.

Third, I would recommend explicitly labelling the exploratory analyses (unplanned statistical tests, additional participants) as such, and maybe adding an interim discussion before such a section that discusses the implications of all the planned analyses in isolation plus motivates the additional analyses. With such an interim discussion, the implications of all results of your planned analyses become clearer and it would be easier to separate them from the exploratory (and I must say very interesting) findings, which require follow-up confirmatory work.

Fourth, I could not find the conditional part of your planned analyses you refer to at the bottom of page 14. In other words, and I might have missed it, I cannot find the part of the stage 1 registered report where you state that you will not include analyses if some results are not significant.

I personally would prefer to strictly follow the plan (again, also to have everything line up).

Fifth, I disagree with your interpretation of power when discussing your results in relation to not replicating BS12 (line 12, page 23). A single study does not provide a reliable effect size estimate, and we do not know the true underlying effect and its distribution over infant age and language. (Which also leads me to wonder about your age distribution compared to the one in BS12). You powered your study for an effect of a certain size with 50 participants (although note that some analyses had less than that number of participants, so the effect size you can observe with reasonable power is necessarily larger). This effect might very well be smaller in reality (see regression to the mean: https://en.wikipedia.org/wiki/Regression_toward_the_mean).

I also recommend to be clear in the discussion which points refer to which sample. (This might be helped with the interim discussion I mention above somewhat).

Through sequential analyses when adding 20 participant, you cannot interpret your p-values in the same way for those analyses (had you preregistered sequential sampling, I would have recommended to simply correct p accordingly, but here I would recommend to largely disregard them), so we need to focus on the Bayes factors, confidence intervals, and effect sizes and consider those for that sample. (See also my point above on providing this information in the manuscript for *all* results). It is still interesting to compare outcomes, I agree, and I appreciate your discussion of possible causes of the difference (although do not forget regression to the

mean and simple measurement variance), but the basis for this is not very solid when using power to reason.

Sixth, regarding the materials shared on OSF, I would recommend (1) sharing the code in an .R or preferably .Rmd (RMarkdown) script. Particularly the latter is very convenient, because you can share the code and the output with your data very easily and transparently.

Right now, everything is only shared as pdf, which shows your analysis pipeline but makes it difficult to reproduce your results

(https://osf.io/54gs9/?view_only=a4f61a751c4b478a814db3a54ac51ead).

Here is more information on RMarkdown: <https://rmarkdown.rstudio.com/>

As minor point on the supplementary materials, I would also recommend adding headlines that clarify which analyses were preregistered, and to which parts of the manuscript they correspond. The current text is already very useful, I simply want to recommend to align it more closely with the stage 1 manuscript.

It would also be great to add descriptive information (age distribution, sex, overall looking times, CDI scores by age, etc) to the analysis document, simply to have it in one place and to be able to compare the two samples more easily.

Finally, on that topic, I recommend including at the end some information about R and package versions, a good way to do this is to include the following command:

https://devtools.r-lib.org/reference/session_info.html

Minor issues

Can you explicitly mention both N at the end of the top paragraph of page 12?

Some instances of "observed" seem a bit unconventional, maybe "inspected" fits better? (e.g. footnote 4 page 12).

Likewise, this expression was difficult to follow for me: "if a child did not validate a trial" (page 13) - Do you mean if a child did not provide sufficient looking time during a trial?

Same page: "an important data loss in our sample" - substantial?

I assume "in total, we obtained 49 average scores for the matching trials and 45 for the related trials" (page 13) means scores for 49 and 45 participants, but it might be nice to state that explicitly.

What is the difference between "difference in target looking" and "average increase in target looking" and how is the latter computed. Both are introduced on page 13. (This might be an easy fix if you follow my major recommendations and take over more information from the stage 1 manuscript so this paper becomes self-contained).

You mention ManyBabies (page 18, footnote 7), and refer to the preprint, but wouldn't it be more appropriate to include a citation to it instead of the link? Otherwise the reader might not know that the link is a manuscript (which might be updated or the link might change).

You mention on page 23 that there is a higher number of vowel cues in Norwegian that infants need to acquire to master the vowel system and that this might delay the growth of their lexicon, but previously cited data from Wordbank led me to believe that at least according to parental reports, Norwegian and American English seem to be relatively similar. Was your sample different from the expected scores? Could you add some words to reconcile your proposal with CDI in general?

And are there studies on lexicon growth / word recognition in other languages with a more high-dimensional phonology that support your proposal?

Your figures are very nice but note that the colors are indistinguishable in black-white print and might not be colorblind-friendly. I recommend using e.g. circles and triangles (or some other shape pair) to make the distinction easier.

Reviewer: 2

Comments to the Author(s)

Please note: I am assuming the authors would have highlighted any changes from the Stage 1 manuscript and have not carefully compared it with the current document.

Norwegian 6-9 month old infants were tested in a word learning task. In the pre-registered analyses, no evidence of word learning was found. However, in exploratory unregistered analyses, evidence at 8-9 months was found, specifically when there was a mismatch in the frequency of the paired items.

Overall, I think the manuscript is in good shape and I see no substantive issues to prevent publication. However, there are some things that I would recommend changing before publication, particularly given the emphasis in the final manuscript on analyses that were not part of the original preregistration. This is not problematic in itself, but it does lead to the need for closer scrutiny and care in wording.

1. Given the heavy salience (and impact on the conclusions) of the unregistered analyses within the “registered” report, it would be helpful to make it a bit more explicit when the transition is made. Perhaps the section “Additional Statistical Analyses” could be retitled as “Exploratory” or “Unregistered” Analyses?
2. I am somewhat unsure how to interpret the finding illustrated in Figure 11. While it is true that infants whose mother reported the word was “understood” showed longer looking in the frequency condition, the converse appears to be true in the context condition. The authors state that “looking times were significantly different from chance only for words reported as ‘understood’ in the frequency condition” – however, based on Figure 11, the negative effect in the context condition seems stronger than the positive effect in the frequency condition. I am aware that overinterpreting the visuals is dangerous, but in this case some additional comment seems warranted, even if the context-yes condition didn’t “reach significance” (was the test two-tailed?). Relatedly, it’s not clear to me what the “dots” represent in this figure.
3. I would be careful about the wording on Page 24 “These differences.... cannot be attributed to a lack of power in the current study”. I get the point, but there is going to be random variation across studies – just because one study has more participants than another that “got a significant effect” does not mean overall that the study is not underpowered.
4. In the same paragraph as [3], I was not terribly convinced by the argument that Norwegian is a simpler language to learn – there are just too many ways to compare languages. If the authors want to make this very specific claim, I would suggest highlighting that it is speculative.
5. I am still a bit uncomfortable with the framing of the results with English as establishing word learning AT 6 months (particularly comparatively with the current finding, e.g. in the last paragraph on page 25, “In sum...”) – the only study (Tincoff et al. X2) that found evidence specifically at 6 months (rather than over a range) tested a very small number of specific words. Bergelson typically frames her result as suggesting the emergence of knowledge of concrete words at “6-9 months”, which encompasses the 8 month old age range tested in this study. I don’t have a problem with noting the lack of finding with 6-7 month olds in this study, but I think there could be a bit more acknowledgement that this difference across the two (sets of) studies may or

may not indicate a meaningful developmental difference across the populations. It is suggestive, but far from confirmatory.

Below I make some additional comments on the wording of the introduction/abstract. Given that this was already accepted through peer-review, it may well be the journal's policy not to make further changes – however, I provide them anyway in case it is appropriate to make further edits.

1. Abstract: “explosion of claims” – this seems a bit over the top. We’re talking about a relatively small number of study.
2. I’m not sure it’s accurate to refer to name recognition as “word comprehension” (page 1 at the bottom)
3. It is perhaps noteworthy that Tincoff et al. found evidence for recognition of “hand” at 6 months given the contrast made between hand and spoon on page 4.
4. On page 5, an N should be provided for the Pearson correlation.
5. Also on page 5, I did not understand in what way book and ball are semantically congruent.

Reviewer: 3

Comments to the Author(s)

Whether the data are able to test the authors’ proposed hypotheses by passing the approved outcome-neutral criteria (such as absence of floor and ceiling effects or success of positive controls)

Not sure about this – see comments below. Some further information is needed.

Whether the Introduction, rationale and stated hypotheses are the same as the approved Stage 1 submission

Seems the case

Whether the authors adhered precisely to the registered experimental procedures

It seems stimuli and attention getter plus parts of procedure are changed from initial pilot to main study data, but in line with reviewers’ comments – however, in light of the first 12 children being included, it is unclear if children in the data set have undergone different testing procedures and stimuli.

Where applicable, whether any unregistered exploratory statistical analyses are justified, methodologically sound, and informative

There is additional exploration, but it is worthwhile – it would be more elegant and economic to include all children in one analysis straight away.

Whether the authors’ conclusions are justified given the data

Please see comments below

Please note that editorial decisions will not be based on the perceived importance, novelty, or clarity of the results.

Comments

Abstract

- grammatical errors – an English native speaker should read it (has witnessed – not witnesses, article missing, etc., spelling of 24-month-olds – hyphenation errors; numbers (38,000))

Introduction

- Line 9 should say 3 months later – from 9 (not 10) months as ref 4 from which presumably the examples sleeping and kissing are taken, tested 9-month-olds.

- Furthermore, sleeping and to kiss are not abstract words, but verbs. An “abstract” word could be to “think”, or “thought”. Sleeping and kissing are visible routines (ref 4 actually used “night night”, not “sleeping” – is that what the authors refer to and mistakenly call it sleeping?). In the case of kissing, this is even a highly actional and volitional event with a results and an effect on

others, ie a highly prototypical action verb. The authors made this error already in a previous version – please correct or omit the word “abstract”.

- The reporting is incorrect line 19-22: Ref 4 has shown that infants failed to show a significant increase in looking towards the target words if the standard word list was tested with items children were expected to know, but which parents had not confirmed as understood. In contrast, children did understand words that parents had confirmed as understood.

- It is curious that understanding of words as judged by parents is assessed in the result section, but not mentioned in the introduction at all. As there is a result chapter on this, this should be included in the introduction as well, and also in aims and hypotheses.

- The authors should report results in their complexity, e.g. the Ref 4 paper showed:

- Nine-month-olds display word knowledge independent of context and without repetitions of words.

- First words encompass not only nouns, but a range of other word classes (e.g. verbs like sleeping and kissing).

- Parents are good at indicating which words their infants do and do not understand.

The authors need to integrate this correctly into the manuscript.

- Line 23: I would advise to phrase more carefully: “TAKEN TOGETHER, these results MAY suggest (...)”

- Line 25-26: the authors write: “however, if the objects belong to the same semantic category (e.g., a cat and a dog both represent animals), infants fail to disambiguate between them, even though they discriminate both objects visually”

Please amend to the more correct version:

However, if the objects belong to the same semantic category (e.g., a cat and a dog both represent animals), infants fail to disambiguate between them IF PARENTS HAVE NOT INDICATED THEM AS KNOWN WORDS (4), even though they discriminate both objects visually.

- L. 30-33: The authors do not portray the full story here as the conceptual level is omitted: There is ample evidence of typicality effects in early word learning (Meints et al., 1999 for nouns, see also Poulin-Dubois & Sissons; for other early word learning see Meints et al., 2002, 2004 and 2008) showing that 12-month-olds do look to a named target, even when part of same category (e.g. cat and dog) if the named items are typical. They do not link the name with atypical targets (regardless of the category being related or not). This conceptual effect gets lost, despite it having been shown to contribute significantly to early word learning. I.e. infants’ word-referent mappings are also conceptually limited to typical mappings at 12-months for nouns, and more refined at 18 months and more fully formed at 24 months. This is an important omission and should be added and integrated for a more complete picture of early word learning– especially also as the authors then go on to assess their stimuli for typicality (information on how ratings were done is not provided).

- L. 47: To sum up, early word categories may also be conceptually limited, not just semantically or perceptually limited (coarse).

- Here and in the discussion, please bear in mind limitations of corpus-analysis as this usually does not consider typicality or saliency information.

- L. 33-37 – contradiction: the authors argue here from corpus data of 1 child why “hand” must be a later word, but they have already shown evidence in the introduction that from 6 months “ infants start showing comprehension for some concrete objects, for example, body parts ‘hand’ and ‘feet’ (5)” - resolve this contradiction in argumentation, please.

- While is useful to look at frequency and the frequency argument is well made, salience and typicality and familiarity are also factors - mention in discussion at least.

Method section

- Curiously, it emerges in the method section under “stimuli” that the authors have after all considered typicality and have assessed all items for typicality, however, still omit to refer to the research that gives the reasons to do so (Meints et al., 1999, and Poulin-Dubois and Sissons) – can

this omission in references be remedied, please, in the method section and in the introduction above.

- p. 8: counterbalancing incomplete: counterbalancing of naming seems forgotten – when pairs are presented, e.g. apple on left, foot on right, in one set of trials the apple should be named, and in a different trial (different child) the foot should be named. Complete counterbalancing looks like this:
 - apple foot, apple named
 - apple foot, foot named
 - foot apple, foot named
 - foot apple, apple named
- In relation to this, was it ensured that also items on the right were named first? This is important as children have a left gaze bias, also for objects (Guo et al., 2008).
- Attention-getter: use of tense - was used not “will be used”
- Figure 2 is misleading as it leads the reader initially to wonder if 2 pairs of images were presented – please adapt acc. to what was really shown (I assume only 1 apple and 1 foot were shown at a time from descriptions below).
- p. 9: the authors wrote: “(...) minimize the number of errors due to parental mistakes – I don’t understand this point if the video is shown and parents are behind the child and “muted” – there is no space for parental mistake anyway – this seems to have been part of an earlier version and of initial pilot testing of the first children - clarify please – and clarify especially if all children taking part and counted into this data set have undergone the same procedure and seen the same stimuli, i.e. why can the 12 initially tested children be included if these had initially other stimuli and a somewhat different procedure? (parents involved in stimulus presentation – see note on this left in manuscript)
- p. 12 of 26: the authors wrote: “The word familiarity questionnaire will ask, for each of the 32 words used in the experiment, how frequently parents have used it (on a scale from 0-never to 5-very frequently) while interacting with the child or in his presence, since their baby was born.” Should read: “asked” in past tense.
- Also, why was this not asked before the study to control for frequency effects?
- Also, asking about familiarity in advance would avoid parents coming in vain as the authors report this in the exclusions: “In addition, two infants did not pass the trial inclusion criteria, because they were unfamiliar with the words depicted in 6 out of 8 picture pairs; their parents reported to never have used them with the child (or in his/her presence), since the child was born.”
- Give mean and age range in participant information, not in data processing, also provide it in months only.
- Drop-out rate is 31% (20 plus 3 kids did not finish while 50 kids finished the study) - this is high – does the procedure need improvement? Or could it be due to the impossibility of the non-match task? Time of testing? Tiredness or hunger? The manuscript does not inform about this.
- The authors wrote: “All infants in the final sample attended to the targets on the control trials, suggesting that they were engaged in the task.” Why was it not checked that infant attended also to test trials? This can easily be checked, esp. when an eye-tracker is used: attention to both stimuli in pre-naming phase should be the criterion per trial, and also in post-naming.
- Frequency data on words not given, please add in table.

Results

- It is somewhat unclear whether the pre-naming phase included the 1500ms of exposure to the 2 images, or only the 0-367ms. If the latter is the case, this is the time a child needs to change their gaze, i.e. the authors then only measure latency instead of a proper pre-naming phase. Instead, the whole 1500ms where the objects are visible should be calculated as pre-naming phase. Ideally, pre-naming would be equal to post-naming. Please clarify in all relevant parts of the manuscript.
- Animacy was not controlled for in stimulus pairs – that could potentially have destroyed any effects if children look more at animate items – and 6 out of 8 picture pairs show animals or body

parts of humans versus inanimate objects – this is far from ideal, given the salient nature of animacy in children’s early categorisation (see all the research by Jean Mandler, Quinn, Rakison, etc.). Was this checked/analysed?

- Also, could the effects be carried by car, cat, keys and banana as these are more frequent items in early language (at least in English “key” is frequent, too), and as these seem significant for older children (in figure) (car-couch / cat-keys and banana-hair).
- Potential confounding issue: Due to the way the results are calculated using only the post-naming phase, it is not possible to see potential bias in images. As items were of differing frequency for the first set within each image pair, this is a potential confounder and it would have been important to disentangle the naming effect from a frequency effect by looking at pre- and post-naming phases separately.

I.e. generally, in pre-naming children should look at both images (equally if equally salient) and the post-naming phase then shows if the name drives the looking. Thus, I am not sure why targets looking time versus distracter looking was not compared for baseline pre-naming phase versus post-naming phase (t/t+d for each)? That would have given a clear additional measure if any of the two displayed items was preferred to the other – that is, another possible reason for exclusion if items are not equally salient – albeit, the frequency condition may actually be confounded in this study as it is likely that children will prefer the more frequent item once it is named – but also potentially before it is named – only analysis of pre- versus post-naming data can show this.

- The rationale for analysis should be to use the best and most useful analysis for the data, not necessarily to replicate other researchers’ analysis.

Discussion

- Include more critical evaluation of results, including points raised above and thorough discussion of frequency differences in matching trials, and saliency / familiarity discussion that could help explain the results better and derive a more complete picture and include suggestions for future research.

Author's Response to Decision Letter for (RSOS-180711.R1)

See Appendix D.

Decision letter (RSOS-180711.R2)

23-Aug-2019

Dear Dr Kartushina:

It is a pleasure to accept your manuscript entitled "Word knowledge in 6-9-month-old Norwegian infants? Not without additional frequency cues." in its current form for publication in Royal Society Open Science.

on behalf of Professor Chris Chambers (Subject Editor)
openscience@royalsociety.org

Dear Editor and reviewers,

We would like to express our sincere gratitude for your thorough reading and your insightful comments and suggestions. It was a pleasure to read them. We carefully addressed them all in a revised version of the report (please find attached). As a result, the original version of the report underwent substantial modifications, which we believe have significantly improved the clarity of the paper and highlighted its impact.

Below, we outline the most important modifications applied to the report.

1. Theoretical framing. First, we adjusted the theoretical framework by including the most recent studies on word comprehension in young infants and by discussing more in depth the role of semantic relatedness in early word learning. Second, we changed the narrative of the study as to provide coherent theoretical background: We better articulated the existing knowledge on early word comprehension (suggesting that infants use linguistic [semantic] cues to disambiguate between items) and the contribution of our study, which aims to test whether infants use other, e.g., extra-linguistic, cues to recognize words. Third, to sharpen the focus of the introduction, we removed the discussion of studies that were not directly related to our research question. Finally, we clarified our hypotheses, by carefully explaining the differences between each condition, and nuanced them by integrating the results of recent studies (Bergelson & Aslin, 2017a, 2017b).

2. Experimental design. We disambiguated between the two key notions used in the study, i.e., contextual and semantic relatedness, which were intertwined in the original report. In addition, we performed quality pre-tests of the stimuli with adults by assessing: (a) the contextual relatedness of the words within each picture-pair and (b) the typicality of the pictures. Finally, we explain further the rationale for selecting words according to their frequency of occurrence in child-directed speech (i.e., using ratios).

3. Methods. First, we carefully specified our criteria to exclude data at infant and trial levels. In particular, we included objective measures (e.g., minimal looking times as recorded by the eye-tracker), quality check trials (control picture-pairs), experimenter's notes, among others. Second, we addressed the familiarity issue: words that infant do not hear at home (as reported by parents in a new questionnaire) will be removed from his/her analyses (at a trial basis). Third, as suggested by two reviewers, we optimized the design and, thus, decreased the likelihood of data loss and drop-out, by (a) using recordings to prompt infants' search, rather than parental speech, and (b) reducing the number of repetition trials. Fourth, we clarified the type and the role of attention-getter and control trials. Finally, we included exhaustive information about the stimuli creation (step-by-step), the experimental procedure and, importantly, we provided the images of the pictures that will be used in the study. To sum up, we included all details needed to replicate the study (and to evaluate the quality of the stimuli/procedures before running it).

4. Statistical analyses. We significantly revised our statistical analysis section. In the revised version of the report, we provided clearer rationale for the statistical analyses (and, thus, we related them better to the hypotheses) and strong arguments in favour of the proposed new tests. Crucially, the new analyses include more graded predictions (in line with the results of recent studies) and follow, as strictly as possible, the analysis pipeline (including the computation of the dependent measures) of the recent studies that the current study aims to replicate. However, in contrast to previous studies, we will provide effect sizes and Bayes factors for each type of analysis. Finally, to address the issue of the "loss of rich eye-tracking timing data", we included a new type of analysis, i.e., a cluster permutation analysis. Cluster permutation analyses will allow to take into account the time course of the looking pattern

and to identify the differences in the dynamics of looking pattern between types of trials and conditions.

5. Interpretation of outcomes. We polished our possible interpretations of the outcomes: we removed over-interpretations (e.g., absence of word knowledge), tuned some claims (e.g., under-specified word categories) and reframed our speculations about the mechanisms of word learning (i.e., use of multiple cues to disambiguate between objects).

We hope that these changes fully address the most important concerns. Point-by-point answers to the specific concerns raised by individual reviewers can be found below.

We are looking forward to hearing from you and thank you again for your feedback that has significantly contributed to the quality of our report.

Sincerely,

Natalia Kartushina and Julien Mayor

Point-by-point answers

Our answers are in blue, 'p' refers to page in the manuscript. The reviewers' text was edited so to include only questions, clarifications and suggestions.

Reviewer: 1

1. Framing

1.1.... I am not sure I follow the reasoning for the two added conditions; in other words: the logic of the hypotheses was not clear to me.

We clarified the logic of the hypotheses by providing specific examples of how infants could use extra-linguistic, i.e., context and frequency, cues to disambiguate between the objects (similarly to what you have suggested). Importantly, we related the hypotheses to the results of previous studies showing the role of contextual and frequency cues in language acquisition. Please see p3.

1.2. My first question is now, how this explains that 6-month-olds seem to do fine when dealing with hands and feet (ref 5).

In fact, we do not deny that infants might build associations between words and referents at the semantic level. We want to test whether they use other/different cues (e.g., extra-linguistic) to disambiguate between two objects. We better articulated this idea in the revised version of the report (p2-3) and hope that our interpretation is not ambiguous any more.

It is also possible that infants may know the meaning of some words (have specified word-referent mappings), but they use/need additional cues to disambiguate some other words. It is also worth noting that in Bergelson and Aslin (2017b) infants failed to show preference for semantically related items, and the items 'foot' and 'hand' were among these words. A look at the item data suggests that infants exhibited a looking preference for 'hand' on both 'hand' and 'foot' trials.

1.3. I also wonder: Would it also be possible to make predictions about effect sizes based on the difference in frequency / context or both? Bergelson & Swingley provide very interesting item-pair analyses

Indeed, in the revised version of the report, we propose to use exactly the same item-pair analysis as in the Bergelson and Swingley (2012) study. However, we were unable to compute

the effect size for this study, due to insufficient report of the information in the paper. Nevertheless, we used the results of follow-up (similar) studies (Bergelson & Aslin, 2017a, 2017b) and a similar study (Tincoff & Jusczyk, 2012) to calculate the effect size for the frequency and context conditions, see p13-14.

1.4. Further, I am not sure I agree with the presentation and interpretation of reference (1), especially in light of reference (15), which did look at semantic relatedness versus nonce words in the labels.

In light of the results of a recent study (Bergelson & Aslin, 2017a) and as you suggested, we elaborated our hypotheses by including an additional outcome: infants show preference for the labels, but also, yet, to lesser extent, for related words. This alternative outcome aligns better with the existing literature and is, therefore, important to consider. Please see p4-5.

1.5. The home-lab links found in (1) also point towards a frequency effect, though. Observing words and their referents together becomes more likely with higher frequency. This could be discussed in more detail in the paper.

Thank you for this relevant suggestion. We discuss home-lab links while talking about the effects of frequency on word learning (p2).

1.6. Finally, (1) provides a very useful discussion of the key results that this paper seems to be built on and offers an interesting alternative that the authors here do not bring up as far as I can see, namely competition effects - and a lot of evidence points towards semantic relatedness predicting word acquisition in typically developing children. The present paper is not able to disentangle the two options, if I see correctly. It doesn't have to, but it should acknowledge the competing possibilities that end up making the same coarse predictions, i.e. less looking when two semantically related objects are present.

Indeed, this paper provides a very interesting alternative. However, in our study we aim at disentangling between context and semantics. Also, our experimental design is different as the babies will not be faced with semantically ambiguous pictures (as in the Bergelson & Aslin 2017 study). However, we would like to discuss the competition effects in the discussion.

1.7. In footnote 3, the authors mention mutual exclusivity, but as far as I know a corresponding ability has not been attested before late in infants' second year (see eg here: https://figshare.com/articles/Disambiguation_Meta_Analysis/1348836), I am not sure how this point applies to infants before their first birthday - see also results in (15). (...) The possibilities are that (1) infants look more to target when it is very frequent and equally to both items when the same pair is presented with the less frequent label or (2) infants look to the "correct" label independent of frequency as long as there is a frequency mismatch.

I think disentangling those possibilities would really push knowledge on infants' early lexical abilities much further.

To sharpen the focus of the paper, we removed the mention to 'mutual exclusivity' from the text. Nevertheless, we believe that your idea is very interesting and do agree with you that a separate study should be performed in order to test the two possibilities.

1.8. I would not conclude from the discussion in paragraph 2 of the introduction that research is inconclusive, but that under different conditions infants do different things - which is to be expected and actually studied in this paper.

We modified this paragraph, so this sentence was removed.

1.9. On page 2, the authors end the top paragraph with a rather general statement, it would have been useful to reiterate how to choose stimuli and why this is (not) surprising.

We modified the text in the introduction, so this sentence was removed. Please see new version of the Introduction section.

1.10. The authors discuss babiness in quite some detail, but I see no definition nor does this come back in the actual experiment.

We removed any mention to babiness, since we do not use it in the experiment.

1.11. On page 2, also carefully distinguish production and comprehension, both require different skills both in the infant and in the parent filling in a questionnaire.

Thank you. We aimed to present those studies that looked at the factors that predicted word learning in infants (both production and comprehension). We do agree that these are two different domains and competence. In our study, we want to test whether different cues that infants use to learn words (independently of the domains) are used to recognize the objects. Actually, if we find that infants rely on contextual cues to disambiguate between words, this would suggest a relationship between early comprehension and production systems, as the same cues that infants use to recognize words in comprehension are used to predict the emergence of words in production.

1.12. Why are familiarity and novelty effect introduced as terms on page 2 if they do not come back?

In the revised version of the report, we included a test in which parents will have to report whether they used the words used in the experiment in a conversation with their child or in his/her presence from their child's birth. The words that parents do not use will be considered as 'unfamiliar'/'novel', and will be removed from the analyses (similar to Tincoff & Jusczyk, 2012). See p10 for detail about the test.

1.13. Finally, do you expect any differences since you are (finally!) testing infants learning a language that is not North American English? Looking at <http://wordbank.stanford.edu> the overall trajectories look similar enough. It would be good to at least mention the language difference and whether this influences the predictions or not.

Upon your suggestion, we added a small comparison of the language development trajectories in Norwegian and English speaking 8-9-month-old infants. The comparison suggests that the trajectories are similar. Please see p3-4.

2. Statistical analysis

2.1. The authors propose to conduct three t-tests to assess what are essentially interaction effects (and I would much prefer to see more graded predictions). This analysis is not able to address the research questions, as null results are essentially uninterpretable in the way authors wish to do so (see below). In any case, to address the (now implicit) question whether infants' responses differ across conditions, a regression-type analysis is necessary. Simply comparing p-values as the authors plan to do now is not sufficient, they cannot even speak to a possible difference between conditions.

In order to assess the differences in looking behaviour between conditions, we will perform paired t-test analyses, similar to a key recent study (Bergelson & Aslin, 2017b). Please see p11. Also, as mentioned above, we modified our analysis pipeline to include more graded predictions.

Although we are aware of the limitation of this type of analysis, we are constrained by the procedures applied in the studies we attempt to replicate. However, in addition to the t-test analyses, we will provide the effect size and the Bayes factor for each of the performed analyses. In addition, we will also perform time-course analyses (please see below). We hope that this solution addresses your concern.

2.2. In addition, I was surprised to see two-tailed tests, while the predictions are clearly directional (looking more at target than distractor upon labelling). A one-tailed test would overall be more useful.

As suggested, we changed the type of the analysis to one-tailed. Previous choice (two-tailed) was motivated by the will to keep the analyses as close as possible to one of the main referenced studies (Bergelson & Aslin, 2017b).

2.3. Finally, in accordance with the journal's guidelines for good statistical practice, include assumption checks before conducting any statistical analyses. For possible data transformations, see also:

Csibra, G., Hernik, M., Mascaro, O., Tatone, D., & Lengyel, M. (2016). Statistical treatment of looking-time data. *Developmental Psychology*, 52(4), 521-536.

As suggested, we added an assumption check before conducting the analyses. Please see p12.

2.4. But even this analysis does not do the rich eye tracking data justice - I would like to point the authors to this excellent R package: <http://www.eyetracking-r.com/>.

We highly appreciate your suggestion and the link to the R package. For this study, in order to examine the rich eye-tracking data we opted for cluster permutation analyses (Maris & Oostenveld, 2007) that was previously used in the analysis of eye-tracking infant data (Dautriche, Fibla, Fievet, & Christophe, submitted). Cluster permutation analyses allow to take into account the time course of the looking pattern and to identify the differences in the dynamics of looking pattern between types of trials and conditions. Please see p14 for a full description of the method we planned on using.

2.6. An updated manuscript should specify the according analyses and move away from p-values as sole basis for any conclusions.

Please see our detailed new section on the statistical analyses and the bases for the conclusions (p12-14).

3. Power / the meaning of null results

This does not mean that the authors should drastically increase their sample size, but procedures to be better able to interpret null results should be implemented to distinguish low-powered results from support for the null hypothesis: the authors should report Bayes Factors and effect sizes (as required by the APA since the 5th edition). Both are straightforward for t-tests and not impossible to obtain for ANOVAs:

<https://sites.google.com/site/lakens2/effect-sizes> and <https://jasp-stats.org/>. Any conclusions should be based on a consideration of all of those. It would of course also be possible to completely move to Bayes Factors, I simply assume the authors might be more comfortable with null hypothesis significance tests.

We are thankful to your suggestions and the links. In order to address the above issue, we, first, calculated the effect sizes and reported the results of power tests in three studies having similar experimental design and the age of the tested infants. Then, we calculated the Bayes factor for the study that has most similar experimental design to ours (Bergelson & Aslin, 2017b). Finally, for each effect size, we reported the results of power test given our sample size.

In addition, as mentioned previously, in our statistical analyses we will report both, the effects sizes and the Bayes factors.

4. Degree of detail needed to replicate the study

4.1. Provide a lab visit walk-through video.

As committed during the submission of our registered report, we will upload a lab visit walk-through video, as well as all the experimental stimuli (audio/video files, questionnaires) used in the experiment to an OSF database.

4.2. List all demographic data you are collecting (infant sex, age; parental education or similar)

We included the data we are collecting in the report, i.e., their linguistic and general background (sex, maternal education, age, number of siblings, language exposure).

4.3. Detail whether (and if yes, how) you plan to share your protocols and materials

We included in the report that the experimental protocols and the materials will be uploaded to the Open Science Framework depository www.osf.io.

4.4. Add a mock screen of what the infant sees and hears to better illustrate the conditions. Provide a list of all picture-pairs and replacement words per conditions (I must admit I was only confused by Table 1).

We improved the clarity of the Table 1 and included three pictures that illustrate the conditions and show what the child will see during the experiment.

5. Piloting

I strongly recommend piloting, especially when working with a new method. I see no specification of pilot participants, and thus there should be no pilots planned. It might be useful to reconsider this, if only to get the procedure straight and ensure that the target images are not dramatically different in terms of salience (this is possible through analyzing pre-naming periods).

We are grateful to your suggestion. We have run successfully two pilots with babies. Both completed the task, were engaged into the task, and fixated the control trials.

6. "Quality checks" / item/trial/participant exclusion criteria

6.1. The quality check does not ensure that the stimuli were chosen to ensure a successful replication (see above point about picture salience, in addition quality checks for the auditory stimuli (adult ratings) and chosen words (are they familiar to the child? The authors make a strong point about this in the introduction) seems advisable.

Thank you. Upon these concerns we (a) assessed the pictures for typicality by five native Norwegian speakers: all sixteen pictures were unambiguously recognized as representing the words they were meant to represent (please note that one object 'milk' was removed); (b) assessed infants' familiarity with the words used in the experiment by asking parents to fill in a questionnaire where, for each of the 32 words used in the experiment, they have to report whether they used it while interacting with the child or in his presence, since their baby was born. The results of this questionnaire will be used in the analyses of the gaze data (see Analyses section for detail). These two points were added to the report.

6.2. I also wonder whether 64 trials is realistic for this age group and would expect a lot of infants to not complete the study.

Upon your suggestion and by having a closer look to the recent similar study (Bergelson & Aslin, 2017b), we opted for less trials, i.e., 32, as in the ditto study.

6.3. I am not aware of a strong argument for parent-produced stimuli.

As suggested, we decreased the likelihood of data loss and drop-out, by using recordings to prompt infants' search, rather than parental speech, as in previous studies.

6.4. Finally, the definition of fussiness is insufficient

We carefully specified our criteria to exclude data at infant and trial levels. In particular, we included objective measures (e.g., minimal looking times as recorded by the eye-tracker), quality check trials (control picture-pairs), experimenter's notes, among others. Please see Data Exclusion section for more detail.

Also, please refer to our general answer in 3. Methods for more detail with regard to the points 6.2-6.4.

7. Further comments on the method

7.1. I take table 1 to contain the actual words, but then I do not see how items are matched for frequency in the corresponding condition. The differences in frequency in fact seem to be similar to the other conditions. This needs a lot more explanation or different stimuli.

Indeed, this issue was one of the most important. In the main text, we explained the rationale for selecting words according to their frequency of occurrence in child-directed speech (i.e., using ratios). Please see footnote 1. In particular, in the frequency condition, words referring to items (and the related words) within each picture-pair will be contrasted on their frequency of occurrence in Norwegian child-directed speech (Braginsky, Yurovsky, Marchman, & Frank, 2017; Larsen, 2014; Simonsen, 1990), with a minimal frequency ratio between the two words of 1:7. Therefore, within each pair, the high frequency word is heard at least seven times more frequently than the low frequency word. In the context condition, the items will be selected in such a way that infants will not be able to use frequency to consistently disambiguate between the objects (the ratio is below 1:4). Please see Word Section for details.

7.2. A white background (p5, l 35) is not to be recommended for eye tracking research (the pupil is too small for most eye trackers to obtain a reliable signal), grey would be a much better choice, as the authors themselves say on page 6 (l 13)

It was an error, we will use grey background. Thank you.

7.3. Some pairs share onsets, how do you account for the to be expected timing difference in infant looks to target (if they know both words, the first phoneme usually disambiguates and lets infants initiate saccades towards or decide to remain fixated on target).

Thank you. For the 'bread-leg' picture pair, whose matching and related words share the same onset (/brø:/, /bæjn/, /bu:r/, /blæjə/), the word onset will be fixed to the onset of the second (disambiguating) phoneme. We included this in the report.

7.4. Why no attention getter between trials? Will you ensure infants look to the screen before a trial starts? Right now it does not look this way, which might lead to even greater data loss.

This is an important issue. Thank you. A video of a spinning colourful flower displayed in the center of a light-grey screen (the same shade as for the experimental stimuli) and accompanied by a tinkling attractive sound will be used to get infants' attention and to center their gaze between each trial.

Reviewer: 2

However, I have a number of concerns about the study as currently articulated. These concerns centre mainly around a) a clearer articulation of the distinction between semantic/conceptual/contextual relationships as they relate to word learning and b) concerns about the frequency analysis. I also have some questions of clarification around the method.

Thank you. As you may see from the letter, the issues that you have raised were one of the most important and we thoroughly addressed them in a revised version of the manuscript.

In particular, we disambiguated between the two key notions used in the study, i.e., contextual and semantic relatedness, which were intertwined in the original report. Please see p3-4. In addition, we performed quality pre-tests of the stimuli with adults by assessing: (a) the contextual relatedness of the words within each picture-pair and (b) the typicality of the pictures. Please see p7. Also, we explain the rationale for selecting words according to their frequency of occurrence in child-directed speech (i.e., using ratios). Please see footnote 1. In particular, in the frequency condition, words referring to items (and the related words) within each picture-pair will be contrasted on their frequency of occurrence in Norwegian child-directed speech (Braginsky et al., 2017; Larsen, 2014; Simonsen, 1990), with a minimal frequency ratio between the two words of 1:7. Therefore, within each pair, the high frequency

word is heard at least seven times more frequently than the low frequency word. In the context condition, the items will be selected in such a way that infants will not be able to use frequency to consistently disambiguate between the objects (the ratio is below 1:4). Please see Word Section for details.

However, a number of questions and concerns emerged regarding the details of the method and stimuli:

1. The testing age range is wide (6-9 month olds). Although this is consistent with widely cited research from one particular research laboratory (the "original study", see below), and therefore appropriate, the authors should take care in their wording in some places regarding whether this is examining lexical knowledge in 6-month-olds (which is the bottom of the testing range).

Thank you. We carefully checked the wording in the report.

2. There was some lack of clarity in the description of the study, I think in part because the authors may have been expecting the reader to be familiar with the details of Bergelson & Swingley. For example, it took me a while to sort out (even with Table 1) how many trials of each type would be presented to each infant. I think I have it figured out, but there is still some confusing language – for example while there appear to be 16 trials repeated twice each (for a total of 32, as with B&S), in one place the authors refer to the "8 picture-pairs". I can interpret this as meaning that "there are 8 picture-pairs for the replication" and an additional 8 for the two tests of alternative explanations (4 each), but I don't see any matching of pairs between the context-relatedness condition and the associated comprehension condition. So will all pairs be presented to all infants? If not, how are they organized?

We apologize for the confusion. We substantially improved the methodological part of the study. Please see our answer to the general issue 3. Methods. Also, importantly, we improved the clarity of the Table 1 and included three pictures that illustrate the conditions and show what the child will see during the experiment. Please see the pictures in the revised version of the manuscript. To briefly answer your questions: all eight picture-pairs will be presented to all infants. Each picture will be seen twice on matching trials and twice on related trials. Please see new sections Pictures on p5 and Procedure on p9. We hope that the revised version of the Methods section is clearer now.

3. I don't see any information for what audio target will be given in the context and frequency conditions – this seems crucial for evaluating any potential confounds in the stimuli, and also assessing subtleties regarding the extent to which "contextually-related" and "conceptually-related" are truly differentiable ideas.

To address this issue we created a separate section Audio Stimuli. Please see p7. Also, we clarified the Procedure by providing specific examples on what infants will see and hear. With regard to the confounds, we carefully explained the two concepts in the Introduction section and assessed our stimuli for contextual and semantic/conceptual relatedness. Please see details on p7. We believe that now that we have substantially reorganized and improved the methods section, this issue should be solved.

4. Ideally, I would want to have access to at least some examples of the audio and videos, e.g. over OSF (I don't see this necessarily as a requirement to evaluate the integrity of the study, but it seems like something that one might expect to see in this case, and might help to clarify some of my above questions).

As committed during the submission of our registered report, we will upload a lab visit walk-through video, as well as all the experimental stimuli (audio/video files, questionnaires) used in the experiment to an OSF database. We included in the report that the experimental protocols and the materials will be uploaded to the Open Science Framework depository [www.osf.io](http://www.osf.io).

5. Given that the authors are setting this up as a direct replication of Bergelson & Swingley rather than a more conceptual replication, it would be helpful to have some more explicit information about the ways in which the stimuli/design were the same or different. Note that while the authors state in a footnote that certain information is unavailable from that study, I am confident that Bergelson and Swingley would be responsive to requests for further information about their methods and results, and perhaps even access to their stimuli. (If an attempt was made and they were not, this is worth stating outright.)

Thank you. We added at the beginning of the Current study section that due to some methodological differences between our experiment and the BS12 study (e.g., the present study will investigate word comprehension in Norwegian infants, see Methods for further details), the current study should be considered as a conceptual replication of Bergelson and Swingley's study, rather than a direct replication. As you suggested, we contacted the first author. However, unfortunately, she could not provide the effect size in short notice. She kindly provided us with a link to a data repository where they stocked the data for another study (Bergelson & Swingley, 2017), that compared the results to the original study (Bergelson & Swingley, 2012) we attempt to replicate. However, the data were processed in a different way, because different trial and subject exclusion criteria were applied. For example, there were 20 infants of 6-7-months old in (Bergelson & Swingley, 2012) study; only 17 of them were retained for the analyses in (Bergelson & Swingley, 2017). These differences made it impossible to recover from these data the effects sizes for the original study (Bergelson & Swingley, 2012). Although the first author had kindly proposed to help, we did not want her to prioritize our needs over the work on such a short notice.

6. The authors state that "the specificities of the eye-tracking machine and the apparatus will be provided at later stages of the pre-registration submission". (p. 6, line 8) From the perspective of "Whether the authors provide a sufficiently clear and detailed description of the methods to prevent undisclosed flexibility in the experimental procedures or analysis pipeline", these details are very important to tie down prior to data collection. It is not clear to me whether there will be an additional opportunity to submit them prior to IPA, but they should be committed to in advance. Again, it would be helpful to know if there are significant details that differ from Bergelson & Swingley in this aspect of the method, particularly in case there is a failure to replicate.

This concern has also drew all our attention. In the revised version of the report, we report the specificities of the eye-tracker. Please note that we substantially improved the Methods section by including exhaustive information about the stimuli creation (step-by-step), the experimental procedure and, importantly, by providing the images of the pictures that will be used in the study. Also, when relevant, we mentioned the methodological differences between our study and the original study we attempt to replicate (Bergelson & Swingley, 2012). To sum up, we included all details needed to replicate the study (and to evaluate the quality of the stimuli/procedures before running it). Please see the revised version of the Methods section.

7. I am concerned about the problem identified by the authors in footnote 5. If the CHILDES frequencies are not indicative of their "true" frequencies in the child's experience, and I think they are right that this is likely a problem, this becomes problematic both for testing Hypothesis 3 and for matching frequencies in the context-related condition. I don't see a way around this (I didn't parse the sentence "Therefore, frequency-matching of words within pairs of food- and bath-related objects is indicative", so I'm not sure what is meant by this or whether it might help at all.)

Indeed, this issue was one of the most important. In the main text, we explained the rationale for selecting words according to their frequency of occurrence in child-directed speech (i.e., using ratios). Please see footnote 1. In particular, in the frequency condition, words referring to items (and the related words) within each picture-pair will be contrasted on their frequency of occurrence in Norwegian child-directed speech, with a minimal frequency ratio between

the two words of 1:7. Therefore, within each pair, the high frequency word is heard at least seven times more frequently than the low frequency word. In the context condition, the items will be selected in such a way that infants will not be able to use frequency to consistently disambiguate between the objects (the ratio is below 1:4). Please see Word Section for details. In addition, in the revised version of the report, we included a test in which parents will have to report whether they used the words used in the experiment in a conversation with their child or in his/her presence from their child's birth. The words that parents do not use will be considered as 'unfamiliar'/'novel', and will be removed from the analyses (similar to Tincoff & Jusczyk, 2012). By doing this, we maximize the chances that infants are assessed on the words they are familiar with.

8. It's not clear to me what will be used as the "distractor" for the proportional (baseline-corrected) measure in the control trials.

We clarified the analyses. In fact, there is no distractor in the control trials. We will measure the proportion of looks at the target object upon hearing its name (at post-naming window).

9. The "cookie-belly" pairing seems problematic as separate categories, since one could imagine talking about eating a cookie and it ending up in the belly?

In order to make sure that the stimuli do not share contextual and/or semantic cues, we assessed the contextual relatedness of the words within each picture-pair. Fourteen parents, having children of ages from 6 months to 6 years, filled in a 5-force-choice questionnaire in which they had to choose, for each word (n = 16), the context in which they believed a 0 to 12-month-old infant was most likely to encounter it. The results confirmed our initial assignments for both contexts, with 96% and 83% of parents assigning the test words to the expected respective kitchen/food and bathroom/cleaning-related routines contexts. Crucially, none of the tested words from the kitchen-related set of items was assigned to the bath/cleaning-related routines, and vice versa. Also, we assessed their semantic relatedness (please see p7). Please note that we changed one related pair 'banana-milk' and replaced it with 'banana-bottle'.

10. It would be helpful to have some articulation of how the authors went about frequency-matching and/or what was considered "high" vs. "low" frequency. In the Table, there is great diversity in frequency (as would be expected given the constraints of a natural language), and it's not at all clear to me what is being considered matched/different and why. This seems critical to being able to address Hypothesis 3 (see below).

Thank you. Please see our answer #7 on page 9.

In addition, I had a number of wording concerns in the introduction.

1. The authors need to better differentiate in some places between studies that identify word comprehension AT 6 MONTHS versus those finding evidence from 6-9 months, particularly given the historical alternative they are differentiating this from is "approximately 8 months of age" (p1, line 20), which is within the 6-9 months range.

Thank you, we carefully checked the wording in the report.

2. It's not clear what is meant by "Research on common words between 6 and 9 months, however, is less conclusive". What is the comparison group here, since the same references are being cited as in the prior paragraph?

The revised version of the report does not contain this sentence any more.

3. I feel like to a certain extent the introduction is setting up a straw man. Don't get me wrong, I agree that it is very important to understand how fragile infants' representations are and how and when they break down. But I don't think anyone is arguing that infants have anything like adult-like understanding of these words. Certainly, if frequency effects are driving some of the prior results, that would put those prior findings in a very different light.

But it's not clear to me that finding a contextual effect is completely inconsistent with the field's current interpretation of the existing literature. This comes down to needing a more nuanced articulation as noted above/below regarding what is meant by "contextual" and how it differs from other kinds of information that infants might use to form word category boundaries.

Thank you. In light of your comment, we significantly reorganized the narrative of the introduction. In particular, we adjusted the theoretical framework by including the most recent studies on word comprehension in young infants and by discussing more in depth the role of semantic relatedness in early word learning. Importantly, we changed the narrative of the study as to provide coherent theoretical background: We better articulated the existing knowledge on early word comprehension (suggesting that infants use linguistic [semantic] cues to disambiguate between items) and the contribution of our study, which aims to test whether infants use other, e.g., extra-linguistic, cues to recognize words. Please see the Introduction section for detail. We hope that this version fully addresses the raised issue.

4. Relatedly, while the fact that our ability to detect the infants' knowledge is fragile is presumably in large part due to the fragility of their knowledge, it is equally likely that some of the fragility is due to the difficulty in testing infant knowledge. While we need to be careful about over-interpreting infant knowledge, we also need to acknowledge the limits of our own measurements of that knowledge.

We agree with you. We will come back to this point in the Discussion of our results.

5. Again relatedly, if infants fail to differentiate between dog and cat, this may not mean that they don't know the word "dog", but rather that their category of dog is not fully differentiated (i.e. a conceptual/category issue rather than a lexical one). One might differentiate (though the authors don't do so cleanly in their introduction if that is indeed their intention) between conceptually related categories of items like "dog" and "cat", and contextually related items like "items encountered during mealtime". But even in the latter case, it is possible to argue that this is part of the earliest stage of category formation – not a failure of the word learning system, but simply a consequence of an emerging conceptual repertoire of meanings to which to attach labels. (The labels are attached to the available meanings, but the meanings are still being refined.) I am not saying that the authors don't have an important distinction to make here, but just that greater clarity in the fine-grained details about what they are actually arguing is needed.

Thank you, we do agree with you. In line with your comments, we significantly refocused the main narrative of the Introduction and, importantly, we emphasize that our study aims to test whether infants use extra-linguistic cues to recognize words, or, in other words, whether their word categories are under-specified enough as to include contextually and frequency-related words.

6. The authors refer to reference 2 as "the original study". There are two possible meanings, one that this is the first study to demonstrate word knowledge in young infants (the way I first interpreted it on page 2, and note that it is not the first such study), and two that it is the study being replicated (which I believe is the intended meaning). This should be made more clear.

Thank you. We changed 'original' to BS12 for Bergelson & Swingley 2012.

With these in mind, I had some specific concerns about the extent to which the analyses can address the hypotheses:

1. Given that the infants under study live in Oslo, it is safe to say they would not be English-speaking as the infants in the original studies were? Given this, and other methodological differences, this should probably not be considered a direct replication, but more of a conceptual one. I don't think this is a problem, but just to be clear in how any "failure to replicate" might be interpreted.

Thank you. Indeed, it is important to state all important differences between our study and the BS12 study. With regard to infants' native language, we added a small comparison of the language development trajectories in Norwegian and English speaking 8-9-month-old infants. The comparison suggests that the trajectories are similar. Please see p5.

2. For Hypothesis 2, I'm not sure that testing against chance performance is right given what they are trying to demonstrate. In addition to the more conceptual concerns expressed already, it's not clear to me that infants looking longer toward contextually-related words is necessarily a "wrong" behavior.

This is an important concern, which was also addressed among the general issues at the beginning of the letter. As you may see, we significantly revised our statistical analysis section. In the revised version of the report, we included more graded predictions (in line with the results of recent studies) for both context and frequency conditions. In particular, we will compare looking behaviour on matching vs related trials. Please see our specific hypotheses and analyses on p4 and p12, respectively.

3. For Hypothesis 3, I think the argument for testing against chance is more sound – it's harder to come up with an explanation for the infants' behavior in that case that is positive for building lexical knowledge, and not simply a confound. However, again a positive statistical test would demonstrate that the confound exists, but would not directly rule out the possibility of semantically-relevant word knowledge. More importantly, the concerns outline above regarding the frequency matching/mis-matching need to be addressed in order to evaluate the validity of this test.

We aligned our analyses for both conditions, so in each condition we will test against chance and also against matching trials. Please see our above answers with regard to the frequency balance in our stimuli pairs.

Other/Smaller points:

1. "Recent large-scale corpus analysis..." (p. 2 line14) – is there a reference for this?
We added accordingly.

2. "babiness" is not word, but I understand the intended meaning. Perhaps "babyish-ness" would work? Or something more technical

We removed any mention to babiness, since we do not use it in the experiment.

3. The sentence "Note, however..." (p. 2, line 25) is difficult to parse.

We changed it.

Reviewer: 3

Items that I would like to hear more about are as follows:

- Attention getters: Attention-getters are usually displayed in IPL in the centre of the screen – also to avoid side-bias – why would they be displayed at the side? Why is separate analysis interesting? Do the authors worry that the children will drop out due to the repetition of the stimuli? The attention-getter stimuli are not described – if they are more interesting than the other stimuli, then they would obviously attract more attention, ie one could argue if they are made to fulfil their purpose, then they have to be more interesting – but then they cannot be compared to the other stimuli.

Thank you. We created appropriate attention-getters for our experiment. A video of a spinning colourful flower displayed in the center of a light-grey screen (the same shade as for the experimental stimuli) and accompanied by a tinkling attractive sound will be used to get

infants' attention and to center their gaze between each trial. The length of the video is ten seconds.

In addition, In order to make sure that infants are fully engaged in the task and that their performance arises from the experimental manipulation(s), we will include four control trials, in which infants will see familiar objects (e.g., a house) which will appear either on the left or on the right sides of the screen, preceded by a prompting phrase produced in a child-directed register. Please see p10 for more information.

- Repetition: Why will stimuli be repeated? The design could also be tested between-subjects.

Thank you. We optimized the design and, thus, decreased the likelihood of data loss and drop-out, by (a) using recordings to prompt infants' search, rather than parental speech, and (b) reducing the number of repetition trials (now it is similar to Bergelson & Aslin, 2017b).

- Choice of stimuli: It is appreciated how very hard it is to find appropriate early word stimuli for this study, however, when it comes to the selection of stimuli, the authors need to make sure that they are not introducing accidental confounders. For example, comparing the comprehension condition (food-related and bath-related item) and the context-related conditions and frequency-related conditions with other comprehension raises some questions. Food-related and bath-related items under Comprehension condition are really food items and body parts. In the context-related condition, food-related and bath-related items are 1 food (drink "milk") and 2 "kitchen cupboard items" and a piece of furniture (table). I wonder whether this could be problematic. While the 3 items are context-related, albeit from 2 different semantic/conceptual categories, milk is another food item. It might be useful to exchange the food item (milk) with "bowl" or "chair" which are early words, too, so at least the items are all non-foods.

Thank you. This is an excellent suggestion. We changed one related pair 'banana-milk' and replaced it with 'banana-bottle'. In addition, in order to validate our context assignments for final stimuli, fourteen parents, having children of ages from 6 months to 6 years, were asked to fill in a 5-force-choice questionnaire in which they had to choose, for each word (n = 16), the context in which they believed a 0 to 12-month-old infant was most likely to encounter it. The results confirmed our initial assignments for both contexts, with 96% and 83% of parents assigning the test words to the expected respective kitchen/food and bathroom/cleaning-related routines contexts. Please see p7 for detail.

The authors write: Crucially, words within each picture-pair and their frequency-matched substitute words will refer to objects from different categories, consequently minimizing any confounding bias at a semantic and/or contextual levels.

Do the authors mean different categories to food and bath-time? Water in the frequency-match is related to the bath-routine.

Since there will be two separate blocks (and statistical analyses) to test for reliance on frequency and contextual cues, matching contexts across conditions should not be a problem. Please note that we controlled, in particular, for the fact that, in the context-related trials, the items would not be disambiguated based on differences in frequency between the objects, whereas in the frequency-related trials, they would not be disambiguated based on differences in the context of use. Crucially, in both frequency- and context-related trials, infants will not be able to rely on semantic cues to disambiguate between objects, since the matching labels and the related to them words will belong to different semantic categories (e.g., 'spoon' and 'cookie' belong to 'utensils' and 'food' categories, yet both are contextually related to the spatial location 'kitchen'; similarly, 'belly' and 'bathtub' belong to different categories, but are encountered in the spatial location of bathroom during bath/cleaning-routines). Therefore, the related item 'water', which will be tested in frequency condition, should not be a problem.

It will be vital to make sure the stimuli really only vary on the dimensions chosen – context and frequency. Introducing more variables – same category / different category) may make findings hard to interpret and has the potential to confound the study as the authors agree.

The authors could use the Norwegian CDI data to help with stimulus choice for the age range. Indeed, the Norwegian CDI data (Kristoffersen & Simonsen, 2012; Simonsen, Kristoffersen, Bleses, Wehberg, & Jørgensen, 2014) was used to choose the stimuli and data from Norwegian child-directed speech was used to balance items for frequency (retrieved from Braginsky et al., 2017; Larsen, 2014; Simonsen, 1990). Please see p4 for the specificities of our trials in both conditions.

- Frequency: It is unclear how the items in the frequency-match condition are frequency matched (range from 2-255) and with which items – within the pair or between conditions?

As you may see from the letter, this issue was one of the most important and we thoroughly addressed it in a revised version of the manuscript. In particular, we explained the rationale for selecting words according to their frequency of occurrence in child-directed speech (i.e., using ratios). Please see footnote 1. In the frequency condition, words referring to items (and the related words) within each picture-pair will be contrasted on their frequency of occurrence in Norwegian child-directed speech, with a minimal frequency ratio between the two words of 1:7. Therefore, within each pair, the high frequency word is heard at least seven times more frequently than the low frequency word. In the context condition, the items will be selected in such a way that infants will not be able to use frequency to consistently disambiguate between the objects (the ratio is below 1:4). Please see Word Section for details. The items were matched for frequency within each pair of items.

- Sound stimuli: Pre-recorded native Norwegian female – then presented to parents over headphones and then repeat to child – why introduce this variation? What if fathers attend versus mothers? In addition, it will be impossible to match target onset with word onset between children. Why not simply use all pre-recorded stimuli and avoid this variation? Baseline and post-word onset times can then also be made identical – this helps with ease of analysis, too.

Thank you. This is an excellent suggestion and (as also mentioned by another reviewer) this helps to decreased the likelihood of data loss and drop-out. So, we recorded a native Norwegian speaker to create our stimuli (in contrast to previous studies that used parental input, e.g., Bergelson & Aslin, 2017b; Bergelson & Swingley, 2012). In addition, as suggested, we controlled for the baseline and post-word onset times in the audio stimuli. Please see our Audio stimuli section on p7.

- Analysis – please explain more clearly what data is intended for comparison.

As stated at the beginning of the letter, we significantly revised our statistical analysis section. In the revised version of the report, we provided clearer rationale for the statistical analyses (and, thus, we related them better to the hypotheses) and strong arguments in favour of the proposed new tests. Crucially, the new analyses include more graded predictions (in line with the results of recent studies) and follow, as strictly as possible, the analysis pipeline (including the computation of the dependent measures) of the recent studies that the current study aims to replicate. However, in contrast to previous studies, we will provide effect sizes and Bayes factors for each type of analysis. Finally, to address the issue of the “loss of rich eye-tracking timing data”, we included a new type of analysis, i.e., a cluster permutation analysis. Cluster permutation analyses will allow to take into account the time course of the looking pattern and to identify the differences in the dynamics of looking pattern between types of trials and conditions.

- Procedure & Stimuli:

- Detail and clarity is not enough yet to replicate the procedure or stimuli, eg the onset of the auditory input from the parents may vary from each parent to the next – how would this be replicated?

Thank you. We substantially improved the Methods section by including exhaustive information about the stimuli creation (step-by-step), the experimental procedure and, importantly, by providing the images of the pictures that will be used in the study. To sum up, we included all details needed to replicate the study (and to evaluate the quality of the stimuli/procedures before running it). Please see the revised version of the Methods section. Please see the answer to your question in the comment above.

- It is unclear whether Norwegian CDIs are collected from the children to be tested to see if their individual word knowledge fits with the stimuli selected – this would be vital as children look at stimuli they have a name for over stimuli they cannot name yet.

After the experiment, parents will be asked to fill in the word-familiarity questionnaire. This questionnaire will ask, for each of the 32 words used in the experiment, whether parents have used it while interacting with the child or in his presence, since their baby was born. The results of this questionnaire will be used in the analyses of the gaze data. In particular, similar to (Tincoff & Jusczyk, 2012), individual picture-pair trials will be removed from individual child data, if the parents reported that they did not use one (or both) of the produced words with the child (or in his presence), since the child was born. For example, if a parent did not use the word 'apple' then all 'apple-foot' picture trials will be removed from his/her child analyses.

- Visual stimuli are not described – as especially children's early word-object-mappings depend on how typical these objects are, visual stimuli should be assessed for typicality in Norway and described for replication (size, background, colour or B&W, etc.). Ideally, and wherever possible, animate stimuli should not be shown with inanimate stimuli as animates tend to be more interesting to children than inanimates.

Thank you for the suggestions and comments. We assessed the pictures for typicality by five native Norwegian speakers: all sixteen pictures were unambiguously recognized as representing the words they were meant to represent. We elaborated on a description of the pictures in a revised version of the report. Please see Pictures section.

- Attention-getters are not described.

We created appropriate attention-getters for our experiment. A video of a spinning colourful flower displayed in the center of a light-grey screen (the same shade as for the experimental stimuli) and accompanied by a tinkling attractive sound will be used to get infants' attention and to center their gaze between each trial. The length of the video is ten seconds. They will be displayed before each trial.

- The authors say "The pictures will remain on the screen for 3500 ms (after the target-word onset) – it is unclear how this will be controlled

In order to control for the picture length, we created video files which align on the length of the audio files. Please see Audio stimuli section and an illustration in Figure 3 for further detail.

□ Whether the authors provide a sufficiently clear and detailed description of the methods to prevent undisclosed flexibility in the experimental procedures or analysis pipeline Not yet enough information given and also, the analysis needs to be clarified. Which data is compared to which other data in the t-tests? Also unclear why this is not tested between-subjects to avoid repetition (and drop-out)?

This is an important concern that we addressed in the general issues section at the beginning of the letter. As you may see, we significantly revised our statistical analysis section. In the revised version of the report, we included more graded predictions (in line with the results of recent studies) for both context and frequency conditions. Please see our specific hypotheses

and analyses on p4 and p12, respectively. Also, we decreased the likelihood of data loss and drop-out, by reducing the number of repetition trials (32 trials as in Bergelson & Aslin, 2017b).

Whether the authors have considered sufficient outcome-neutral conditions (e.g. positive controls) for ensuring that the results obtained are able to test the stated hypotheses
Yes, but please see comments above that need to be addressed.

We substantially revised our control stimuli and the analyses. Please see Control stimuli and Control check analyses sections.

Other:

Introduction:

In the introduction, it may be more useful not to speak of concrete versus abstract words – but use the appropriate linguistic terms instead (eg verbs, etc.). “Abstract word” can be easily confused with abstract noun (love, belief, etc.).

Thank you. We removed abstract words from the Introduction and refer to concrete words only (as these will be tested in our study).

Babiness is not a good word. Saliency seems to be meant, or infant-relatedness?

We removed any mention to babiness, since we do not use it in the experiment.

Spoon over nose: “‘nose’, which is heard in less spatially and linguistically distinct contexts.” This is unclear - children may only hear their nose mentioned in highly-specific contexts, eg when they hearing and playing baby rhymes when brought to bed, so this needs careful consideration.

Thank you. We removed the word ‘nose’ from the text.

References

Bergelson, E., & Aslin, R. (2017a). Semantic Specificity in One-Year-Olds’ Word

Comprehension. *Language Learning and Development*, *0*(0), 1–21.

<https://doi.org/10.1080/15475441.2017.1324308>

Bergelson, E., & Aslin, R. N. (2017b). Nature and origins of the lexicon in 6-mo-olds.

Proceedings of the National Academy of Sciences, 201712966.

<https://doi.org/10.1073/pnas.1712966114>

Bergelson, E., & Swingley, D. (2012). At 6–9 months, human infants know the meanings of

many common nouns. *Proceedings of the National Academy of Sciences of the United States of America*, *109*(9), 3253–3258.

<https://doi.org/10.1073/pnas.1113380109>

Bergelson, E., & Swingley, D. (2017). Young Infants’ Word Comprehension Given An

Unfamiliar Talker or Altered Pronunciations. *Child Development*.

<https://doi.org/10.1111/cdev.12888>

- Braginsky, M., Yurovsky, D., Marchman, V., & Frank, M. C. (2017). Consistency and variability in word learning across languages. *Proceedings of the National Academy of Sciences*. Retrieved from 10.17605/osf.io/cg6ah
- Dautriche, I., Fibla, L., Fievet, A.-C., & Christophe, A. (submitted). Learning homophones in context: Easy cases are favoured in the lexicon of natural languages.
- Kristoffersen, K. E., & Simonsen, H. G. (2012). *Tidlig språkutvikling hos norske barn. MacArthur-Bates foreldrerapport for kommunikatív utvikling*. Oslo: Novus.
- Larsen, T. L. (2014). *Byggekløssar i barnespråk. Om tre norske born si tileigning av funksjonelle kategoriar*. the Norwegian University of Science and Technology (NTNU), Trondheim.
- Maris, E., & Oostenveld, R. (2007). Nonparametric statistical testing of EEG- and MEG-data. *Journal of Neuroscience Methods*, *164*(1), 177–190. <https://doi.org/10.1016/j.jneumeth.2007.03.024>
- Simonsen, H. G. (1990). *Barns fonologi: system og variasjon hos tre norske og et samoisk barn. [Children's phonology: system and variation in three Norwegian children and one Samoan child]*. University of Oslo.
- Simonsen, H. G., Kristoffersen, K. E., Bleses, D., Wehberg, S., & Jørgensen, R. N. (2014). The Norwegian Communicative Development Inventories: Reliability, main developmental trends and gender differences. *First Language*, *34*(1), 3–23.
- Tincoff, R., & Jusczyk, P. W. (2012). Six-Month-Olds Comprehend Words That Refer to Parts of the Body. *Infancy*, *17*(4), 432–444. <https://doi.org/10.1111/j.1532-7078.2011.00084.x>

Appendix B

Dear editor and reviewers,

We are very impressed by the depth and the quality of the feedback and are deeply grateful to reviewers' comments and suggestions. We have carefully addressed them in a revised version of the manuscript (please find attached). First, we clearly explained the rationale of the frequency manipulation and gave concrete examples, early into the text. Importantly, as suggested by reviewer 1, we included the results of a by-pair effect size re-analysis performed for the BS12 study that provides support for the frequency manipulation in the planned study; the positive correlation between effect sizes and frequency imbalance between items in BS12 suggests that infants capitalise on frequency imbalance to discriminate between items. Second, we improved the statistical section of the manuscript as follows: (a) we detailed the statistical plan further (concern raised by reviewer 1); (b) we included a comparison of the effect sizes obtained in the current study to those reported in previous research; (c) we specify the role of Bayes Factor analyses and, finally, (d) we included multi-level (mixed-effects) regression analyses to examine graded predictions and variability in random effects (as suggested by reviewer 1). Third, we highlighted that the study aims to examine whether 6-9-month-old infants rely on non-linguistic cues (frequency and context of use) when disambiguating between items (concern raised by reviewer 2). Fourth, we discuss below the advantage of using the present design for the control trials, i.e., without any distractor (concern raised by reviewer 1). Finally, we modified the manuscript to take into an account all suggested edits.

We believe that the manuscript has significantly gained in clarity and quality, and hope that it will meet all the requirements for a successful pre-registered report. Please find below/attached point-by-point answers to the specific concerns raised by each reviewer.

Sincerely,

Natalia Kartushina and Julien Mayor

Reviewer: 1

1. Explain the frequency account with an example as early as possible.

Thank you. The current version of the manuscript clearly explains how an imbalance in frequency might help infants to disambiguate between items. We provide concrete examples and emphasise the role of frequency in terms of a mapping between high/low frequently heard labels and high/low frequency encountered objects: e.g., when hearing the high frequency label 'hand', infants might look at the high frequency encountered object, 'hand', but also to other high frequency encountered objects such as, for example, 'hair' or 'milk' (Page 3).

2. You could extract differences and variances from Figure 2 of BS12 (among others) and then plot effect sizes against frequency ratios or differences. It's only 6 pairs, but for each pair you have two estimates. Note for example that hand-yogurt is not showing a difference in the older age group, but more in line with your alternative explanation hair-banana seems to work well.

This is an excellent suggestion. We performed this analysis; it revealed a strong correlation between the effect-size and the frequency mismatch between the two words in a pair in BS12 (Pearson $r=.71$, $p=.048$), suggesting that frequency differences might be an important cue that infants use to disambiguate between items (Page 3).

3. Interpret the Bayes factors and effect sizes and do not interpret the difference between significant and non-significant tests

Thank you. The current version of the manuscript announces, specifically, that Bayes factors and effect sizes (and, importantly, their comparison with those reported in previous research) will be used to interpret the results of the study. In order to replicate and compare our results to those obtained in previous research (BS12, BA17), we kept simple one-tailed and two-tailed paired analyses. However, and crucially, in order to perform graded comparisons, we will also perform multi-level (mixed-effects)

regression analyses with subjects and picture pairs being included as random factors. Please see Page 13. We hope that these changes fully address this concern.

4. Add a distractor to control trials

The authors have added a control to check whether infants are on task, which I consider a great improvement over the typical design. However, since there is currently no distractor, what infants do in the control trials is very different from what the authors claim it assesses. As with attention getters, infants will look at a single object if it's the only thing on the screen. There is no assessment whether they react to the label and I would expect a ceiling effect that makes such an assessment actually impossible (if you were to compare looks before to looks after naming, and you have to introduce a baseline correction to really get at the naming effect as you claim on page 11).

We do agree with you, if infants are engaged in the task and look at the visual stimuli that appear on the screen, then we might obtain ceiling results on the control trials. However, and importantly, this is exactly what we want to measure: infants' gaze re-orientation (after being centered with the attention getter) to the visual stimuli appearing on either screen side that attract their attention. We believe that if we include a distractor, then the stimuli in the control condition will not differ from those used in the experimental condition. The aim of the control trials is to examine whether, in absence of a distractor, infants follow the instruction to look at the target and reorient their gaze to fix the named image. We believe that, despite the limitations they have, the control trials proposed in the current study accomplish this aim.

Minor remarks:

- Throughout the paper: Make sure you cite all R packages (including pwr). This link will explain how to get citation data for R packages: <https://stat.ethz.ch/R-manual/R-devel/library/utils/html/citation.html>

This has been done.

- Throughout the paper: "Related" is a very unclear label for a condition, do you mean context or frequency or both?

As we introduce this in p.4 (3rd paragraph), we use the term 'related' [words] to refer to the context and frequency related words used in context and frequency conditions, respectively. We decided to use this term for the sake of consistency: i.e., in both conditions (frequency and context) there are matching and related words. Related words in frequency condition have similar frequency, whereas related words in context condition are similar in the context of use. However, we clarified the two terms throughout the paper and, in particular, in the statistical section of the paper, which we believe was affected the most by this terminology.

- P1: What is the target image for abstract words (night-night)? Is this mention relevant here?

For the target word 'night-night', Syrnyk and Meints (2017) used a video clip. Here is its description: "With his back to the camera, a man (father) holds a young girl (daughter) in his arms, rocking her back and forth. She has her head on her father's shoulder and her eyes are closed." We replaced 'night-night' with 'sleeping'.

- P1: The example in paragraph 3 is inconsistent: banana-hand is introduced, then hair is mentioned later as label.

Many thanks for pointing this out. We have now changed 'hand' to 'hair'.

- P1, end of 3rd paragraph: It might be useful to mention that we're quite sure that infants can distinguish all objects visually.

Thank you, we added this at the end of the 3rd paragraph on P1.

- P2, 2nd paragraph: Is shape part of semantics? Shape doesn't come back, and I don't think of ball and bowl to be semantically related but visually similar. It might be good to be as clear as possible here.

In P2, 2nd paragraph, we now explicitly mention that shape is part of perceptual cues, which are not manipulated in the current study.

- P2, 2nd paragraph: The goals of this paper come quite sudden and before you introduce your alternative explanations, maybe move this to the end of the introduction or rephrase?

Thank you. We believe that mentioning the two cues that we are planning to manipulate is a nice way to make a transition to the next section, which will present these two cues in detail. Also, we believe

that introducing the goals of the study at this point focuses readers on the crucial aspects of the study, in comparison to previous research.

- The subheadline on p2 is a bit unclear, as you already mention factors modulating word comprehension in the preceding section, how about something like: Two alternative account for infants' behavior in word recognition studies?

Thank you, we changed it according to your suggestion

- P2, 3rd paragraph, 1st sentence: either "a recent..." or "analysEs" (pl)

Changed to 'A recent'.

- Idem: Not all studies use concrete words as you reviewed on page 1

Now changed to 'most experimental studies'.

- P2, bottom paragraph: Could you either briefly introduce the units or use another measure of contextuality?

For the sake of clarity, we removed the units.

- Idem: Why the " around predicted?

We removed '.

- Idem: Can you elaborate on the less contextually distinct words as you do on the more distinct words?

To illustrate less contextually distinct words, we used the word 'hand', which is used in less spatially and linguistically distinct contexts. We elaborated on the possible contexts of its use. Please see P2, 3rd paragraph.

- P3, 3rd paragraph: missing "explanations" after "alternative" in the last sentence.

Now added.

- P3, bottom: I think it's good to mention early on that you also replicate the study in a new language, to me that is a strength of your work and a much needed expansion.

Thank you, we added this in the first sentence of the 'Current study' section.

- P4, top paragraph: What do you mean with "knowledge/performance"?

We specified that we referred to 'recognition behaviour'.

- Idem: "as" or "compared to" (I would say)

Changed to 'compared to'.

- P4, 3rd paragraph, 1st sentence: "two additional typeS of trials"

Changed accordingly.

- P5, Participants: "Shall"-> "Should"

Changed accordingly.

- Idem: Add information about whether you obtain informed consent.

We added that informed consent will be obtained from the parents, see P5 'Participants'.

- Idem: Add preterm birth and developmental delays to the exclusion criteria.

Thank you, we added this information.

- P6, figure 1: Isn't couch also contextually related to carpet?

This is a very good remark. In Norwegian, the word 'carpet' has several meanings, i.e., (1) a piece of fabric to lay on the floor or over a bed, (2) a piece of fabric to hang on the wall of decorations, wallpaper and (3) a piece of fabric to hang in front of a scene. Indeed, one of them is contextually related to the word 'couch' (both could be seen in the living room and one could stand on the other). Nevertheless, given the constraints we had to select the stimuli (imageability, frequency, semantic relatedness, word knowledge and phonology), we believe it can remain included in the set of words.

- Figures 2,3: Increase font sizes, please

Thank you, we increased font sizes in both figures.

- Figure 2: Related? I assume you mean contextually/frequency related? Could you label which is which? I also am not sure you need to show the same images 4 times, you can mark the respective targets and/or have one condition be above and the other below the respective image

Thank you, we modified the figure: now it does not repeat the same images and it specifies (in line with the information given in the text) that the sentences were produced in the context condition: on matching and related trials.

- P8: Can you make clear whether BS12 used different pictures or not?

We selected a new set of pictures, compared to BS12, to be closer to culturally-relevant stimuli for Norway (as attested positively by adult ratings).

- P9, Procedure, 1st sentence: “explain the study (to them)” (I’d think)
Thank you, we changed accordingly.
- P9, 5th paragraph: Can you refer back to the respective figure? (In general you can rely more on the figures in the text).
Thank you, we now referred back to the respective figure.
- P9 and elsewhere: Try to avoid “)(“ and join text in parentheses, separated by a semi-colon.
Changed it here and at other relevant places.
- P 10, bottom: A failure to fixate in ALL control trials? 50%?
Thank you. We clarified that we referred to matching and related conditions. We also realised that our criteria were rather strict (as compared to other studies) and changed the criteria to 20%.
- Idem: “In this case” (not conditions, those are experimental).
Thank you. We changed this accordingly.
- Idem: Ideally streamline your exclusion. First exclude all with failed calibration, etc, THEN perform the quality check. Also make sure to re-apply your child-level criteria after excluding single trials.
Thank you, we changed the exclusion section based on your suggestions.
- P12: I don’t follow where 100 *diff_prop* scores come from.
We clarified this in the text as follows: ‘Therefore, for each subject, we will obtain two *average_diff_prop* scores in the context condition, one for matching and one for context-related trials; and two *average_diff_prop* scores in the frequency condition, one for matching and one for frequency-related trials (i.e., 100 *average_diff_prop* scores (50 infants * 2 *average_diff_prop* scores for each condition)’
- P12, (3); P13, (3) Related vs matching trials: Add “testS” in the headline to make clear that you want to conduct multiple tests.
We believe this is a misunderstanding, because only one two-tailed paired t-test will be performed in each condition (P12, (3); P13, (3)). However, the current version of the manuscript does not have these analyses anymore but relies on the results of multi-level regression analysis to perform graded analyses.
- P14: “Also” is not the best start of a new section.
Thank you, we changed it to ‘In addition’.

Reviewer: 2

Page 2 and elsewhere: The authors have overall addressed my concern regarding semantic/conceptual relationships versus context. I still feel they could further differentiate between category membership that is conceptual (i.e. non-linguistic) versus semantic (i.e. labeling-based). Their study doesn’t really differentiate these two possibilities – I don’t see this as a fatal flaw, but worth acknowledging somewhere.

Many thanks. We believe that the latest version of the manuscript highlights the different types of cues used by infants when recognising words, and that the planned study will aim at finding out whether infants can rely on single extra-linguistic cues (context, frequency) when disambiguating between items, despite not having established word-object associations at a semantic level.

Analysis section: Overall I find the analyses to be very clear and appropriate. My one remaining concern is somewhat nitpicky, but I still feel important for an RR. In the post hoc power analysis, the authors discuss carefully what they “expect” to find, but do not explicitly describe what they would do/how they would interpret their finding should their power expectations not be met. This leaves open a small window of “researcher degrees of freedom” that should be squared away.

Many thanks for your comment. In the revised version of the manuscript we now specify that we will calculate the effect size using the *pwr.t.test* function (18) and will compare it to the effect size(s) obtained in similar studies using the *BF_T/BF_U* function (24) in R. This will allow to evaluate whether the results (i.e., the effect size) obtained in the current study replicate the results of previous studies. A failure to replicate the results of previous studies (performed in English) would suggest that Norwegian 6-9-month-old infants do not recognise the familiar concrete words, used in the study (Page 12). In addition, we included multi-level (mixed-effects) regression analyses and discuss all possible outcomes resulting from the model.

Wording:

In a few places in the manuscript (e.g. page 5), the authors use the term “the latter”. I’d suggest avoiding this in each case, as they are all places where the reader is trying to follow a complex line of reasoning, and it adds to the reader’s burden.

Thank you. We removed the term ‘the latter’ from the manuscript.

“Big data” – I would avoid using this term as it means very different things to different people.

Thank you, we changed it to ‘large-scale corpus’ study.

Appendix C

Cover letter

Dear Editor Christopher Chambers,

We are happy to announce that we have finished data collection and analyses for our preregistered report entitled ‘Word knowledge in 6-9-month-old infants. Recognition without comprehension?’. Please find attached the results and the discussion of our study.

Our results revealed that, in contrast to previous studies with American-English learning infants, 6-7-month-old Norwegian infants did not show word comprehension for familiar words used in the study (e.g., body parts, food items, animals). 8-9-month-olds, on the other hand, looked longer at the target when they heard the target label as compared to the distractor; however, they disambiguated only those item pairs that were imbalanced in the frequency of word use. This suggests that 8-9-month-old infants have established some/a mapping between a referent word and an item; however, this mapping is weak, as infants need additional cues (such as an imbalance in frequency of word use) to reveal word recognition. 6-9-month-old Norwegian infants failed to rely exclusively on the frequency or context of word use to disambiguate between two items, though.

Given the results of our study, we would like to slightly change the title of our manuscript to: ‘Word knowledge in 6-9-month-old Norwegian infants? Not without additional frequency cues.’

The URL for the study raw and processed data, digital materials/code and the laboratory log can be found on page 1 of the Stage 2 manuscript containing, as well, the URL for the approved Stage 1 protocol on the Open Science Framework (OSF).

As agreed in the email dated 04.06.2018, we included data from infants who were tested before the IPA Stage 1 date. Twenty of them took the current study after having completed a different study and thus were not included in the main sample; their analysis is reported separately. Those of them (n=12) who took the current study first, were included in the main sample. We certify that (1) the experimental protocol was absolutely identical for the data already collected before the IPA Stage 1 date and the data that was collected after the IPA date; and (2) we had not observed any of the already-collected data in any way prior to the date of Stage 1 IPA.

We also confirm that the completed experiment(s) have been executed and analysed in the manner originally approved with any unforeseen changes in those approved methods and analyses clearly noted. The outcome of all registered analyses is included in the main document and the codes are available in the additional document on the OSF. Unregistered (*post hoc*) analyses are reported in a separate section of the results. We specified the data analysis procedures that we used, including the rules for data elimination.

We are looking forward to receiving your feedback.

Sincerely,

Natalia Kartushina & Julien Mayor

Appendix D

Dear editor and reviewers,

We are sincerely thankful to your thorough reading of our manuscript and to your insightful and valuable comments and suggestions. We have addressed all major and minor issues in the revised version of the manuscript. The main amendments include the following.

First, as suggested by Editor and Reviewer 1, we added data from the control trials both in the manuscript and on the OSF depository (we detailed the results of the analyses in the Data pre-processing section and added raw data to the Excel file 'Participants_info_70'). Second, as suggested by Reviewer 1 and 2, we added an interim discussion for the planned analyses, whereby we briefly discuss the implications of the planned analyses in isolation and motivate the exploratory analyses. Third, we included effect sizes (95% CI, Cohen d or R^2), Bayes factors and all statistics in our analyses¹, in particular for non-significant ones in the planned analyses, and provided a fully commented R code (R_code.R on the OSF) for the replication/reproduction purposes (Reviewer's 1 concern). Forth, we reworded our interpretation of the results (power over-interpretation) in relation to not replicating BS12 (Reviewer 1 and 2) and we included the research on the role of typicality in infant word recognition (as suggested by Reviewer 3). Fifth, we clarified the apparent contradiction between the studies reporting the acquisition of the word 'hand' (raised by Reviewer 2 and 3). Only minor changes (clarifications, errors) were made to the Introduction and Methods sections (pointed by Reviewer 2 and 3), though. All changes in the revised version of the manuscript are marked in a purple ink.

Point-by-point answers to specific issues can be found below.

We believe that the manuscript has significantly gained in clarity and quality, and we hope that it will meet all requirements for a successful publication.

We thank you again for your valuable input and we look forward to hearing from you.

Kind regards,

The authors

Reviewer: 1

In general, I would not treat those two documents as separately as done in the manuscript. While the reader can certainly access the stage 1 version, the manuscript would be easier to follow if it were self-contained and details like preprocessing were discussed transparently and in depth in this manuscript, and any deviations from the plans were highlighted (i.e. simply stating "as preregistered, we did ...[section from the stage 1 manuscript]").

Thank you. We re-organised and edited the Results section to improve the transparency and the clarity of the manuscript. The following changes were implemented: (1) we added a subsection Dependent measures to ease the understanding of the results; (2) we added a subsection Control quality check, where we described the analyses performed to examine infants' engagement in the task; (3) we highlighted any deviations from the plan in the Results and pre-processing sections (e.g., we stated explicitly when the analyses were preregistered and when they were not registered/exploratory); (4) importantly, we now clearly distinguish the two types of analyses performed in the manuscript, i.e., Planned statistical analyses and Exploratory statistical analyses, in two dedicated sections.

Second, I am missing Bayes Factors, effect sizes, and sometimes all statistics for non-significant analyses. All this information is crucial, particularly given (1) your testing of an additional sample and

¹ Note that for the sake of clarity (the results section is quite dense), statistical details for some (exploratory) results are not reported in the manuscript, but are provided in the additional document, which the readers are readily referred to.

(2) your interpreting null results as evidence for the null hypothesis (difficult without effect sizes and BF) in the discussion section in particular, but also in the results (e.g. page 14).

Thank you. As mentioned above, in a revised version of the manuscript, we included effect sizes, Bayes factors and all statistics for all analyses including non-significant ones; yet, for the sake of clarity (the results section is quite dense), statistical details for some (exploratory) results are not reported in the manuscript, but are provided in the additional document, which the readers are readily referred to.

I also see an interpretation of condition differences (8-9 month olds exploratory analyses), without having tested whether this difference is significant in a reduced model or referring to the full linear mixed model that showed no significant effect (page 18). I would caution against overstating any findings in this case, particularly since those analyses are strictly exploratory.

Thank you. We have added the results of the full linear mixed model showing no significant effect of condition. In the revised version of the manuscript, we highlighted that future research is needed to confirm our results, as the analyses were strictly exploratory.

Third, I would recommend explicitly labelling the exploratory analyses (unplanned statistical tests, additional participants) as such, and maybe adding an interim discussion before such a section that discusses the implications of all the planned analyses in isolation plus motivates the additional analyses. With such an interim discussion, the implications of all results of your planned analyses become clearer and it would be easier to separate them from the exploratory (and I must say very interesting) findings, which require follow-up confirmatory work.

To address this issue, as suggested by the reviewer, we implemented the following changes: first, we labelled our additional statistical analyses as '*Exploratory statistical analyses*'; and, second, before the ditto section, we added a brief discussion section entitled '*Interim discussion of the planned analyses*' to discuss the implications of all our planned analyses in isolation and to motivate the additional analyses.

Fourth, I could not find the conditional part of your planned analyses you refer to at the bottom of page 14. In other words, and I might have missed it, I cannot find the part of the stage 1 registered report where you state that you will not include analyses if some results are not significant. I personally would prefer to strictly follow the plan (again, also to have everything line up).

Thank you. It was our mistake. Given an already very dense Statistical analyses section, we opted for not reporting these non-significant results in the main manuscript. We now removed the conditional part from the manuscript and now refer readers to the exact sections in the supplementary document codes_results_analyses.pdf (Sections 2.3 and 2.4), where they can find the results of these analyses (also in the R_code script).

Fifth, I disagree with your interpretation of power when discussing your results in relation to not replicating BS12 (line 12, page 23). A single study does not provide a reliable effect size estimate, and we do not know the true underlying effect and its distribution over infant age and language. (Which also leads me to wonder about your age distribution compared to the one in BS12). You powered your study for an effect of a certain size with 50 participants (although note that some analyses had less than that number of participants, so the effect size you can observe with reasonable power is necessarily larger). This effect might very well be smaller in reality (see regression to the mean: https://en.wikipedia.org/wiki/Regression_toward_the_mean).

Thank you for the comment and for the link. We removed our interpretation of power from the discussion.

I also recommend to be clear in the discussion which points refer to which sample. (This might be helped with the interim discussion I mention above somewhat).

As suggested, we clarified in the beginning of the discussion section which results originate from which analysis. In addition, before presenting the exploratory analyses, we added an interim discussion section, where we briefly discuss the planned analyses.

Through sequential analyses when adding 20 participant, you cannot interpret your p-values in the same way for those analyses (had you preregistered sequential sampling, I would have recommended to simply correct p accordingly, but here I would recommend to largely disregard them), so we need to focus on the Bayes factors, confidence intervals, and effect sizes and consider those for that sample. (See also my point above on providing this information in the manuscript for *all* results).

As suggested, in the revised version of the manuscript, we provided confidence intervals, effect sizes and Bayes factors for all analyses and use them while interpreting our results.

Sixth, regarding the materials shared on OSF, I would recommend (1) sharing the code in an .R or preferably .Rmd (RMarkdown) script. Particularly the latter is very convenient, because you can share the code and the output with your data very easily and transparently. Right now, everything is only shared as pdf, which shows your analysis pipeline but makes it difficult to reproduce your results (https://osf.io/54gs9/?view_only=a4f61a751c4b478a814db3a54ac51ead). Here is more information on RMarkdown: <https://rmarkdown.rstudio.com/>.

Thank you for providing information about RMarkdown, it looks great; unfortunately, given the time constraints, we will not be able to generate an .Rmd script. (We will make sure we use it in our future reports.) However, as suggested, we have added a fully-commented code in an .R format so readers can reproduce our analyses and the output (we also added additional data frames to ensure reproducibility). It is now available on the OSF page of the study.

As minor point on the supplementary materials, I would also recommend adding headlines that clarify which analyses were preregistered, and to which parts of the manuscript they correspond.

Thank you. We added headlines to the supplementary materials (i.e., results_and_codes.pdf) to clarify which analyses were preregistered and to which parts of the manuscript they correspond.

It would also be great to add descriptive information (age distribution, sex, overall looking times, CDI scores by age, etc) to the analysis document, simply to have it in one place and to be able to compare the two samples more easily.

Thank you. We added descriptive information about each sample to the analysis document. In addition, we added a summary table providing the means and the confidence intervals for each group, condition and trial type (when available) for the main analyses. [Also implemented in the .R code]. Please see a revised version of the document analyses_results_codes.pdf.

Finally, on that topic, I recommend including at the end some information about R and package versions, a good way to do this is to include the following command: https://devtools.r-lib.org/reference/session_info.html

Thank you. We added at the end of the analyses document the information about R and package versions using the suggested command.

Minor issues

Can you explicitly mention both N at the end of the top paragraph of page 12?

This has been amended.

Some instances of "observed" seem a bit unconventional, maybe "inspected" fits better? (e.g. footnote 4 page 12).

Thank you. We changed "observed" to "inspected".

Likewise, this expression was difficult to follow for me: "if a child did not validate a trial" (page 13) - Do you mean if a child did not provide sufficient looking time during a trial? Same page: "an important data loss in our sample" - substantial?

Thank you. This has been amended.

I assume "in total, we obtained 49 average scores for the matching trials and 45 for the related trials" (page 13) means scores for 49 and 45 participants, but it might be nice to state that explicitly.

Thank you. We have stated this explicitly.

What is the difference between "difference in target looking" and "average increase in target looking" and how is the latter computed. Both are introduced on page 13. (This might be an easy fix if you follow my major recommendations and take over more information from the stage 1 manuscript so this paper becomes self-contained).

Thank you for pointing out this inconsistency. The "average increase in target looking" is the "difference in target looking" averaged over subjects. We now corrected this in the manuscript and use consistently (both in the text and in the figures) the same term "difference in looking proportion"; we specify when it is averaged over subjects or items. We clarified also that this measure was part of the preregistered report and distinguish it from the other measure (baseline-corrected target looking) that was examined in the exploratory analyses.

You mention ManyBabies (page 18, footnote 7), and refer to the preprint, but wouldn't it be more appropriate to include a citation to it instead of the link? Otherwise the reader might not know that the link is a manuscript (which might be updated or the link might change).

Absolutely, thank you. We added an appropriate citation.

You mention on page 23 that there is a higher number of vowel cues in Norwegian that infants need to acquire to master the vowel system and that this might delay the growth of their lexicon, but previously cited data from Wordbank led me to believe that at least according to parental reports, Norwegian and American English seem to be relatively similar. Was your sample different from the expected scores? Could you add some words to reconcile your proposal with CDI in general? And are there studies on lexicon growth / word recognition in other languages with a more high-dimensional phonology that support your proposal?

Thank you. Indeed, when we compared the median number of words reported as understood in the CDI, there were no differences between Norwegian and American-English 8 and 9-month-old infants (at least according to parental reports). Our two samples of 8 and 9-month-old infants showed similar vocabulary sizes to those reported in Wordbank for 8 and 9-month-old Norwegian infants, suggesting that our sample was representative of its population. We added the following footnote on page 25: "Note that the median vocabulary sizes of 8.5 and 15 words, reported in our study by parents of 8 and 9-month-old infants, respectively, are very similar to the median vocabulary sizes of 9 and 19 words previously reported in Norwegian 8 and 9-month-old infants (CDIs; retrieved from wordbank.stanford.edu, see 12,15,16), the vocabulary size revealed in our sample represents the vocabulary size of Norwegian 8-9-month-old infants (at least as far as the CDI reports are concerned)." As far as 6-7-month-olds are concerned, we have no CDI norms for this early age to compare to, yet, the vocabulary sizes for this age group reported in our study look coherent with the data reported in 8-9-month-old infants as they allow to draw a nice linear developmental trajectory. For instance, the median CDI scores in (70) infants tested in our study are 4.5, 5, 8.5 and 15 words for 6,7,8 and 9-month-old infants, respectively.

With regard to other studies on lexicon growth / word recognition in other languages with a more high-dimensional phonology, we have identified only one of them that appears to support our hypothesis. The following was added: "For instance, an analysis of the CDI reports revealed that the

vocabulary score in the Danish children was the lowest across 17 languages from the age one onwards (32), suggesting that the complexity of a language's sound system might influence early lexical development. Together with the fact that Danish is considered to be less intelligible than Norwegian (as judged by native Scandinavians, in 33), we can only speculate that word comprehension would be delayed in infants learning a phonologically more complex language (Danish). Future research needs to shed light on this issue."

Your figures are very nice but note that the colors are indistinguishable in black-white print and might not be colorblind-friendly. I recommend using e.g. circles and triangles (or some other shape pair) to make the distinction easier.

Thank you. We have edited all our figures: in the revised version of the manuscript, we use shapes (circles and triangles) to make distinctions.

Reviewer: 2

1. Given the heavy salience (and impact on the conclusions) of the unregistered analyses within the "registered" report, it would be helpful to make it a bit more explicit when the transition is made. Perhaps the section "Additional Statistical Analyses" could be retitled as "Exploratory" or "Unregistered" Analyses?

Thank you. We took this concern seriously (also expressed by Reviewer 1). As can be seen in the letter, we labelled our additional statistical analyses as 'Exploratory statistical analyses' and we added a brief discussion section entitled 'Interim discussion of the planned analyses' to discuss the implications of all our planned analyses in isolation and to motivate the additional analyses.

2. I am somewhat unsure how to interpret the finding illustrated in Figure 11. While it is true that infants whose mother reported the word was "understood" showed longer looking in the frequency condition, the converse appears to be true in the context condition. The authors state that "looking times were significantly different from chance only for words reported as 'understood' in the frequency condition" – however, based on Figure 11, the negative effect in the context condition seems stronger than the positive effect in the frequency condition. I am aware that overinterpreting the visuals is dangerous, but in this case some additional comment seems warranted, even if the context-yes condition didn't "reach significance" (was the test two-tailed?). Relatedly, it's not clear to me what the "dots" represent in this figure.

We clarified that the analyses were two-tailed; we added the results of the analyses in the context condition (Cohen $d = 0.33$, $BF_{10} = 1.97$, $p = 0.13$, two-tailed one-sample test) in a footnote. We added to the figure's caption that coloured shapes (red and blue) represent individual child data averaged over item pairs.

3. I would be careful about the wording on Page 24 "These differences.... cannot be attributed to a lack of power in the current study". I get the point, but there is going to be random variation across studies – just because one study has more participants than another that "got a significant effect" does not mean overall that the study is not underpowered.

Thank you. We agree with your point (also raised by Reviewer 1). We removed this sentence from the manuscript.

4. In the same paragraph as [3], I was not terribly convinced by the argument that Norwegian is a simpler language to learn – there are just too many ways to compare languages. If the authors want to make this very specific claim, I would suggest highlighting that it is speculative.

Thank you. We added that the argument is speculative. However, we added, in the following paragraph, that the complexity of the phonological system might have impact on early language

development. An analysis of the CDI reports revealed that the vocabulary size in the Danish children was the lowest across 17 languages from the age one onwards (Bleses et al., 2008), suggesting that the complexity of a language's sound system might influence early lexical development. Together with the fact that Danish is considered to be less intelligible than Norwegian (as judged by native Scandinavians, see Delsing and Lundin, 2005), we can only speculate that word comprehension would be delayed in infants learning a phonologically more complex language (Danish). Future research needs to shed light on this issue.

5. I am still a bit uncomfortable with the framing of the results with English as establishing word learning AT 6 months (particularly comparatively with the current finding, e.g. in the last paragraph on page 25, "In sum...") – the only study (Tincoff et al. X2) that found evidence specifically at 6 months (rather than over a range) tested a very small number of specific words. Bergelson typically frames her result as suggesting the emergence of knowledge of concrete words at "6-9 months", which encompasses the 8 month old age range tested in this study. I don't have a problem with noting the lack of finding with 6-7 month olds in this study, but I think there could be a bit more acknowledgement that this difference across the two (sets of) studies may or may not indicate a meaningful developmental difference across the populations. It is suggestive, but far from confirmatory.

Thank you. We checked that we use the term "age range over 6-7 months of age", rather than 6 months, in our comparative analyses and in the discussion. Please note that recently Bergelson and Aslin (2017) have shown word recognition in 6 month-old infants.

In the revised version of the manuscript, we acknowledged (in the Discussion section) that more data are required in order to be able to claim that there are developmental differences across the Norwegian and American-English learning infants. In addition, as suggested for the last paragraph, we added "Yet, more cross-language studies are needed to confirm this hypothesis."

Additional comments:

1. Abstract: "explosion of claims" – this seems a bit over the top. We're talking about a relatively small number of study.

Thank you. We agree with you. We removed the word 'explosion'.

2. I'm not sure it's accurate to refer to name recognition as "word comprehension" (page 1 at the bottom)

We changed it to 'word recognition'.

3. It is perhaps noteworthy that Tincoff et al. found evidence for recognition of "hand" at 6 months given the contrast made between hand and spoon on page 4.

Thank you for this remark. Indeed, Tincoff and Jusczyk (2012) found evidence for recognition of "hand" at 6 months; however, a comparison with the production data reported in Roy et al (2015) indicates that word recognition alone is not sufficient to trigger its production, as words 'recognized' early are not necessarily those that are produced early. We highlighted that, in this context, 'acquisition' refers to word production. We hope this change addresses your comment.

4. On page 5, an N should be provided for the Pearson correlation.

Thank you. We added $n = 8$.

5. Also on page 5, I did not understand in what way book and ball are semantically congruent. Both are toys. We added this in the text.

In light of the first 12 children being included, it is unclear if children in the data set have undergone different testing procedures and stimuli.

Thank you for this comment. The disclaimer was included in a foot note (page 12 in the original document) in the section Data pre-processing; we certified that (1) the experimental protocol was absolutely identical for the data already collected before the IPA Stage 1 date and the data that was collected after the IPA date; and (2) we had not inspected any of the already-collected data in any way prior to the date of Stage 1 IPA. To make this clear, we moved this information earlier in the manuscript, to the section 'Participants'.

Comments

Abstract

- grammatical errors – an English native speaker should read it (has witnessed – not witnesses, article missing, etc., spelling of 24-month-olds – hyphenation errors; numbers (38,000))

Thank you. We fixed the listed errors and carefully checked the new version of the manuscript.

Introduction

- Line 9 should say 3 months later – from 9 (not 10) months as ref 4 from which presumably the examples sleeping and kissing are taken, tested 9-month-olds.

Thank you. It appears that the results for 9-month-olds are contradictory between the two studies (Bergelson and Swingley, 2013 and Syrnyk and Meints, 2017). However, given the differences in stimuli, it is difficult to draw any strong conclusions. We now say that infants start understanding more abstract words at 9-10 months of age; we also now give examples of more abstract words used in one of the studies, e.g., "all-gone", "hi". We also highlighted that these are 'more' abstract words.

- Furthermore, sleeping and to kiss are not abstract words, but verbs. An "abstract" word could be to "think", or "thought". Sleeping and kissing are visible routines (ref 4 actually used "'night night", not "sleeping" – is that what the authors refer to and mistakenly call it sleeping?). In the case of kissing, this is even a highly actional and volitional event with a results and an effect on others, ie a highly prototypical action verb. The authors made this error already in a previous version – please correct or omit the word "abstract".

Thank you. We agree with you. The word "sleeping" was used in Bergelson and Swingley (2013) study. As indicated above, we now give examples of more abstract words, e.g., "all-gone", "hi" (also highlighted in the manuscript). We adopt the wording "abstract word" following Bergelson and Swingley's (2013) article "The acquisition of abstract words by young infants".

- The reporting is incorrect line 19-22: Ref 4 has shown that infants failed to show a significant increase in looking towards the target words if the standard word list was tested with items children were expected to know, but which parents had not confirmed as understood. In contrast, children did understand words that parents had confirmed as understood.

Thank you for this remark. We fully agree with you, the authors reported that infants failed to show word comprehension if they reportedly did not know the target words. However, in this sentence, we reported the results for nouns and for those of them that were used in the standard list condition (as these nouns are typically used in infant research, e.g., dog, ball, car, duck, etc.). The authors report that "while infants' looking to the target increased after naming for standard set nouns reported to be understood, this did not reach significance" (Syrnyk and Meints, 2017, p.210). For the sake of clarity, we removed this reference from the sentence.

- It is curious that understanding of words as judged by parents is assessed in the result section, but not mentioned in the introduction at all. As there is a result chapter on this, this should be included in the introduction as well, and also in aims and hypotheses. • The authors should report results in their complexity, e.g. the Ref 4 paper showed:

- Nine-month-olds display word knowledge independent of context and without repetitions of words.
- First words encompass not only nouns, but a range of other word classes (e.g. verbs like sleeping and kissing).
- Parents are good at indicating which words their infants do and do not understand. The authors need to integrate this correctly into the manuscript.

Thank you for this suggestion. Examining word understanding as judged by parents was not planned in our preregistered report and was performed in an exploratory analysis. Given the overarching policy requirement for a Stage 2 RR that no changes can be made to the rationale, aims or hypotheses, we are unfortunately unable to address this comment directly and to integrate the full results of the study into the Introduction section. However, in the revised version of the manuscript, we discuss the results of this exploratory analysis in light of the results of the above-mentioned study in the Discussion section.

- Line 23: I would advise to phrase more carefully: “TAKEN TOGETHER, these results MAY suggest (...)”

Thank you, this has been amended.

- Line 25-26: the authors write: “however, if the objects belong to the same semantic category (e.g., a cat and a dog both represent animals), infants fail to disambiguate between them, even though they discriminate both objects visually”. Please amend to the more correct version: However, if the objects belong to the same semantic category (e.g., a cat and a dog both represent animals), infants fail to disambiguate between them IF PARENTS HAVE NOT INDICATED THEM AS KNOWN WORDS (4), even though they discriminate both objects visually.

Thank you for this suggestion. Unfortunately, we are not able to amend it, because the sentence refers to the results of the study by Bergelson and Aslin (2017), where the authors manipulated the semantic similarity between the two objects in a pair with no adjustment for word knowledge.

- L. 30-33: The authors do not portray the full story here as the conceptual level is omitted: There is ample evidence of typicality effects in early word learning (Meints et al., 1999 for nouns, see also Poulin-Dubois & Sissons; for other early word learning see Meints et al., 2002, 2004 and 2008) showing that 12-month-olds do look to a named target, even when part of same category (e.g. cat and dog) if the named items are typical. They do not link the name with atypical targets (regardless of the category being related or not). This conceptual effect gets lost, despite it having been shown to contribute significantly to early word learning. I.e. infants’ word-referent mappings are also conceptually limited to typical mappings at 12-months for nouns, and more refined at 18 months and more fully formed at 24 months. This is an important omission and should be added and integrated for a more complete picture of early word learning— especially also as the authors then go on to assess their stimuli for typicality (information on how ratings were done is not provided).

Thank you for this detailed comment and your insightful suggestion. Given the limited nature of changes that we are allowed to make with respect to the sections Introductions and Methods, we added only a brief sentence on the role of typicality in both Introduction and Methods sections. In particular, we added that “infants’ word-referent mappings are limited to typical exemplars of the semantical categories until the age of 24 months” (Meints et al., 1999) and “Given previously reported effects of typicality on word recognition (Meints et al., 1999), the pictures were assessed for typicality by five native Norwegian speakers. Nevertheless, we integrated the research on typically in our discussion session. We added more information about how ratings were done (on a scale from 1-low to 5-high; all were rated 5) in the methods section.

- L. 47: To sum up, early word categories may also be conceptually limited, not just semantically or perceptually limited (coarse).

Thank you for this comment. Unfortunately, for the above-mentioned reasons, it is not possible to add this in the manuscript, as it would require the inclusion of more details on the development of conceptual knowledge in infants in the section ‘Introduction’.

- L. 33-37 – contradiction: the authors argue here from corpus data of 1 child why “hand” must be a later word, but they have already shown evidence in the introduction that from 6 months “infants start showing comprehension for some concrete objects, for example, body parts ‘hand’ and ‘feet’ (5)” - resolve this contradiction in argumentation, please.

Thank you for this remark (also pointed by Reviewer 2). Indeed, Tincoff and Jusczyk (2012) found evidence for recognition of “hand” at 6 months; however, a comparison with the production data reported in Roy et al (2015) indicates that word recognition alone is not sufficient to trigger its production, as words ‘recognized’ early are not necessarily those that are produced early. We highlighted that, in this context, ‘acquisition’ refers to word production.

- While it is useful to look at frequency and the frequency argument is well made, salience and typicality and familiarity are also factors - mention in discussion at least.

Thank you. The following has been added to the discussion section: “Other research has shown that object-related factors, such as salience and typicality may also influence word comprehension in young infants, as 12-month-old infants do not recognize atypical objects, suggesting that their early word categories are conceptually limited to typical word-referent mappings (Meints et al., 1999, 2008).”

Method section

- Curiously, it emerges in the method section under “stimuli” that the authors have after all considered typicality and have assessed all items for typicality, however, still omit to refer to the research that gives the reasons to do so (Meints et al., 1999, and Poulin-Dubois and Sissons) – can this omission in references be remedied, please, in the method section and in the introduction above.

Thank you for this helpful suggestion. We have added the reference to Meints et al. (1999) both in the methods and in the introduction sections.

- p. 8: counterbalancing incomplete: counterbalancing of naming seems forgotten – when pairs are presented, e.g. apple on left, foot on right, in one set of trials the apple should be named, and in a different trial (different child) the foot should be named. Complete counterbalancing looks like this:
 - apple foot, apple named
 - apple foot, foot named
 - foot apple, foot named
 - foot apple, apple named

As mentioned in the Procedure section, to counterbalance the side of the object presentation across trials and conditions, two presentation lists were created (List 1 and 2). In each list, the target object appeared twice on each side (once on matching and once on related trial). Given the nature of our design (the same picture pair was used four times, two on matching and two on related trials), having complete counterbalancing within each infant would require eight presentations of the same picture pair, which would make the task very repetitive and very long. The two presentation lists were created to address this limitation.

- In relation to this, was it ensured that also items on the right were named first? This is important as children have a left gaze bias, also for objects (Guo et al., 2008).

Thank you. This is an interesting suggestion and a very interesting study. Guo et al., 2008 showed that infants had left gaze bias while looking at one-object images centered on the screen. It would be interesting to know whether this bias operates when infants are presented with two spatially separated objects on the screen. Other research suggests that when 6-month-old infants are faced

with several objects (six in the cited study), they fixate first those objects that appear most salient to them, as, for example, faces (Gliga, Elsabbagh, Andravizou & Johnson, 2009), independently of their spatial location on the screen, suggesting that object salience/preference might overwrite the left gaze bias when several objects are present on the screen. To avoid salience/preference bias, in our study, the side of the target object was pseudo-randomised: half infants heard the right object named first and the other half heard the left object named first.

- Attention-getter: use of tense - was used not “will be used”

Thank you. We amended this.

- Figure 2 is misleading as it leads the reader initially to wonder if 2 pairs of images were presented – please adapt acc. to what was really shown (I assume only 1 apple and 1 foot were shown at a time from descriptions below).

Thank you for pointing this out. We clarified in the Figure’s caption, that “on each trial, only one picture-pair was shown on the screen”.

- p. 9: the authors wrote: “(...) minimize the number of errors due to parental mistakes – I don’t understand this point if the video is shown and parents are behind the child and “muted” – there is no space for parental mistake anyway – this seems to have been part of an earlier version and of initial pilot testing of the first children - clarify please – and clarify especially if all children taking part and counted into this data set have undergone the same procedure and seen the same stimuli, i.e. why can the 12 initially tested children be included if these had initially other stimuli and a somewhat different procedure? (parents involved in stimulus presentation – see note on this left in manuscript)

Thank you for pointing out this issue. The phrase “minimize the number of errors due to parental mistakes”, written in the section ‘Audio stimuli’, was used to motivate our choice for the use of audio pre-recorded stimuli, as opposed to parental prompts used in BS12. To make this clear, we moved this sentence to the beginning of the section ‘Audio stimuli’. In a footnote on page 12, we certified that (1) the experimental protocol was absolutely identical for the data (twelve participants) already collected before the IPA Stage 1 date and the data that was collected after the IPA date; and (2) we had not inspected any of the already-collected data in any way prior to the date of Stage 1 IPA.

- p. 12 of 26: the authors wrote: “The word familiarity questionnaire will ask, for each of the 32 words used in the experiment, how frequently parents have used it (on a scale from 0-never to 5-very frequently) while interacting with the child or in his presence, since their baby was born.” Should read: “asked” in past tense.

Thank you. This has been amended. We carefully checked the manuscript for misuse of the future tense (used in the preregistered report) instead of the past tense.

- Also, why was this not asked before the study to control for frequency effects?

Our first aim was to replicate conceptually the results of the BS12 study, where the authors did not manipulate the frequency effects.

- Also, asking about familiarity in advance would avoid parents coming in in vain as the authors report this in the exclusions: “In addition, two infants did not pass the trial inclusion criteria, because they were unfamiliar with the words depicted in 6 out of 8 picture pairs; their parents reported to never have used them with the child (or in his/her presence), since the child was born.”

We did not want to ask about familiarity in advance in order to not influence parental behaviour before the test. Some parents might feel anxious about “not using some words that we asked about” and might start using them before coming to the lab, making then the questionnaire data invalid.

- Give mean and age range in participant information, not in data processing, also provide it in months only.

Participants' mean and age range are indicated in both sections, Participants (Methods) and Pre-processing (Results). To describe participants, we now provide mean and age range in months only.

- Drop-out rate is 31% (20 plus 3 kids did not finish while 50 kids finished the study) - this is high – does the procedure need improvement? Or could it be due to the impossibility of the non-match task? Time of testing? Tiredness or hunger? The manuscript does not inform about this.

On page 12, we explained in details the breakdown of our drop-out rate, largely driven by the strict application of the exclusion criteria laid-out in the pre-registered report. In particular, we mentioned that “Twenty other infants did not pass the trial inclusion criteria, i.e., they were not able to contribute to at least 20% (6/32) of the experimental trials, either because they were not looking at either image in the pre-naming period or they were not looking at either image for at least 0.5 sec in the post-naming period, as in (Bergelson and Aslin, 2017)”.

- The authors wrote: “All infants in the final sample attended to the targets on the control trials, suggesting that they were engaged in the task.” Why was it not checked that infant attended also to test trials? This can easily be checked, esp. when an eye-tracker is used: attention to both stimuli in pre-naming phase should be the criterion per trial, and also in post-naming.

In the revised version of the manuscript, we provided more information about the analyses of the control trials. We do agree with the reviewer that it is important to make sure that participants attended to test trials. That is why, as mentioned in the Data exclusion criteria (page 11 in the original document), we carefully checked participants' looking behaviour on the test trials. In particular, “after the quality check, the following criteria were used to exclude single trials: (a) no looking at either image for at least 0.5s in the post-naming period, (b) no looking was recorded in the pre-naming period (as in Bergelson and Aslin, 2017)”.

- Frequency data on words not given, please add in table.

Thank you for pointing this out. The frequency Table 1 was not added in the submitted version of the manuscript; we do apologize for this. We have included the table in the revised version of the manuscript.

Results

- It is somewhat unclear whether the pre-naming phase included the 1500ms of exposure to the 2 images, or only the 0-367ms. If the latter is the case, this is the time a child needs to change their gaze, i.e. the authors then only measure latency instead of a proper pre-naming phase. Instead, the whole 1500ms where the objects are visible should be calculated as pre-naming phase. Ideally, pre-naming would be equal to post-naming. Please clarify in all relevant parts of the manuscript.

The pre-naming exposure included 1500ms of silence plus the time before the word onset, which was around 400ms, giving, in total around 2000ms of pre-naming exposure before the word onset, as indicated in Figure 3 (Section ‘Stimuli’). To clarify this in the analyses, we added, in the new section ‘Dependent measures’ (Planned statistical analyses) that the pre-naming window lasted around 2000ms.

- Animacy was not controlled for in stimulus pairs – that could potentially have destroyed any effects if children look more at animate items – and 6 out of 8 picture pairs show animals or body parts of humans versus inanimate objects – this is far from ideal, given the salient nature of animacy in children's early categorisation (see all the research by Jean Mandler, Quinn, Rakison, etc.). Was this checked/analysed?

Thank you for pointing out this important issue. Yet, as our main goal was to replicate the results of BS12 study, we, by design, were constrained by the use of body parts in one of our conditions (four pictures in the context condition). Our choice to use animals was partly dictated by the constraints of our design in the frequency condition, where high-frequency items had to be combined with low-frequency items. As infant early receptive vocabulary is already very limited, we were left with little

choice for the frequently used items that would be known by infants in our study. Animals are typically well known by 6-9-month-old infants and are relatively frequent in infant-directed-speech. To address this issue, we performed exploratory analyses with the baseline-corrected measure of looking time (corrected for initial preference). These analyses revealed that, analogously to our previous analyses with the dependent measure 'difference in looking proportion', 8-9-month-old infants showed word comprehension in the frequency condition only ($p = .041$, Cohen $d = 0.47$, $BF_{10} = 2.89$). We added this additional analysis to our revised version of the manuscript (please see page 23). Please note that our planned cluster permutation analyses examined infants' looking behaviour over the whole duration of the trial and, therefore, accounted for a potential looking preference/saliency at pre-naming window.

- Also, could the effects be carried by car, cat, keys and banana as these are more frequent items in early language (at least in English "key" is frequent, too), and as these seem significant for older children (in figure) (car-couch / cat-keys and banana-hair).

Indeed, and as expected from our design, the frequency imbalance between items triggered word comprehension; importantly, it is not the absolute frequency, but the imbalance in frequencies between the two items who was driving the effect. The frequency alone (as for example 'book' that is highly frequent in early input) did not induce word comprehension.

- Potential confounding issue: Due to the way the results are calculated using only the post-naming phase, it is not possible to see potential bias in images. As items were of differing frequency for the first set within each image pair, this is a potential confounder and it would have been important to disentangle the naming effect from a frequency effect by looking at pre- and post-naming phases separately.

Thank you. As mentioned above and now in the revised version of the Results section, we have performed additional exploratory analyses on a baseline-corrected measure. The results were very similar to those obtained with a measure over the post-naming phase only. Please see page 23. In addition, we performed cluster-permutation analyses that examined infants' looking behaviour over the whole duration of the trial and, therefore, accounted for infants' looking preference at pre-naming window.

I.e. generally, in pre-naming children should look at both images (equally if equally salient) and the post-naming phase then shows if the name drives the looking. Thus, I am not sure why targets looking time versus distracter looking was not compared for baseline pre-naming phase versus post-naming phase (t/t+d for each)? That would have given a clear additional measure if any of the two displayed items was preferred to the other – that is, another possible reason for exclusion if items are not equally salient – albeit, the frequency condition may actually be confounded in this study as it is likely that children will prefer the more frequent item once it is named – but also potentially before it is named – only analysis of pre- versus post-naming data can show this.

Thank you for this comment and for the suggestion to use a baseline-corrected measure. Please see the three answers above that address the issue raised in this comment.

- The rationale for analysis should be to use the best and most useful analysis for the data, not necessarily to replicate other researchers' analysis.

We fully agree with you; in our study, we used three different looking measures and three different types of analyses (paired, mixed-effects and cluster permutation analysis) to examine word comprehension in young infants. The results are very similar across the analyses and measures.

Discussion

- Include more critical evaluation of results, including points raised above and thorough discussion of frequency differences in matching trials, and saliency / familiarity discussion that could help explain the results better and derive a more complete picture and include suggestions for future research.

Thank you. In addition to improvements suggested by Reviewer 1 and 2 (described in the cover letter), in the revised version of the discussion section, we included the results of previous research on typicality, on parental knowledge of infants' word comprehension and on the role of phonological complexity in early language development. We included suggestions for future research as well. We hope they address the comment.